

# Response of water temperatures and stratification to changing climate in three lakes with different morphometry

Madeline R. Magee[1], Chin H. Wu[1]

[1]Department of Civil and Environmental Engineering, University of Wisconsin-Madison, Madison, WI 53706, U

*Correspondence to*: Chin H. Wu (chinwu@engr.wisc.edu)

**Abstract.** Water temperatures in three morphometrically different lakes are simulated using a one-dimensional hydrodynamic lake model over the century (1911-2014) to elucidate the effects of increasing air temperature and decreasing wind speed on lake thermal variables (water temperature, stratification dates, strength of stratification, and surface heat fluxes). During the study period, epilimnetic temperatures increased, hypolimnetic temperatures decreased, and the length of the stratified season
increased for the study lakes due to earlier stratification onset and later fall overturn. Additionally, there was an abrupt change in epilimnion temperature after 1930 in both Lake Mendota and Lake Wingra, and three changes, after 1934, 1995, and 2008 for Fish Lake. There was a significant change in the slope of trend of stratification duration after 1940 in Lake Mendota and a significant change in trend after 1981 for Fish Lake. Schmidt stability showed a statistically significant increasing trend for both deep lakes, with the larger trend and greater variability in the larger surface area lake. Sensible heat flux in all three lakes
increases over the simulation period while longwave heat flux decreases. The shallow study lake had a greater change in latent heat flux and net heat flux, illustrating the role of lake depth to surface heat fluxes. Sensible heat flux in all three lakes had similar timing of abrupt changes, but the magnitude of the change increased with increasing depth. Abrupt changes in latent heat flux appear to be independent of lake morphometry, indicating that the timing of change may be primarily driven by climate. Perturbing drivers showed that increasing air temperature and decreasing wind speed caused earlier stratification onset
and later fall overturn. For hypolimnetic water temperature, however, increasing air temperature warmed bottom waters while decreasing wind speed cooled bottom waters, indicating that the change of hypolimnetic temperatures globally may be influenced by local changes in wind speed. Overall, lake depth impacts the presence of stratification and magnitude of Schmidt stability, while lake surface area drives differences in hypolimnion temperature, hypolimnetic heating, variability of Schmidt stability, and stratification onset and fall overturn dates.



# 1 Introduction

Climate over the past century has changed. Globally averaged land and ocean surface temperature anomalies have increased over the period from 1850-2012 (IPCC, 2013). In the Northern Hemisphere, 1983-2012 was likely the warmest 30-year period of the last 1400 years (IPCC, 2013). Studies suggest that more intense and longer lasting heat waves will appear in the future

(Meehl and Tebaldi, 2004), and there has been a trend of increasing mean temperature anomaly across the continental United States (Hansen et al., 2010). In Wisconsin, the air temperature increased by 0.61°C from 1950 to 2006 (Wisconsin Initiative on Climate Change Impacts (WICCI), 2011). Furthermore, changes in wind speeds across the globe have been observed. For example, wintertime wind energy increased in Northern Europe (Pryor et al., 2005), while modest declines in mean wind speeds were observed in the United States (Breslow and Sailor, 2002). Similarly, on the regional scale, Klink (2002) reported

a decreasing trend in annual wind speed at five of seven stations in and around Minnesota from 1959 to 1995 and Magee et al. (2016) showed a decrease in wind speeds occurring in Madison, Wisconsin after 1994. In contrast, increasing wind speeds were observed in Lake Superior, North America (Austin and Colman, 2007). Generally, it is recognized that air temperature and wind speed have significantly changed over the last century and will likely continue to change in the future.

Lake water temperature is closely related to the meteorological variables of air temperature and wind speed. Previous studies show that warming air temperatures have caused increasing epilimnetic water temperatures (Dobiesz and Lester, 2009; Shimoda et al., 2011), increased the strength of stratification (Rempfer et al., 2010), prolonged the stratified period (Livingstone, 2003; Robertson and Ragotzkie, 1990), and altered thermocline depth (Schindler et al., 1990). For instance, Lakes Superior, Michigan, and Huron exhibited increasing water temperature and increased stratification duration during

between 1979 and 2006 (Austin and Colman, 2007). In contrast, hypolimnetic temperatures have undergone both warming or cooling trends depending on season (Robertson and Ragotzkie, 1990). Changes in wind speed also strongly affect lake mixing (Boehrer and Schultze, 2008), lake heat transfer (Boehrer and Schultze, 2008; Read et al., 2012), and temperature structure (Desai et al., 2009; Schindler et al., 1990). Stefan *et al.* (1996) found that decreasing wind speeds resulted in increased stratification and increased epilimnetic temperatures in inland lakes. In Lake Superior, increased wind speeds caused by the

decreasing air-water temperature differences (Desai et al., 2009) should have resulted in water temperature decreases, but observations show instead increasing water temperatures due to complex nonlinear interactions among air temperature, ice cover, and water temperature (Austin and Allen, 2011). In recent years, we have improved understanding of changing air temperature and wind speed on alterations of water temperature and stratification (Magee et al., 2016). Nevertheless, there still remains uncertainty in the response to isolated and combined changes in lakes.



The lake ecosystem is significantly impacted by changes in lake water temperature (MacKay et al., 2009). For example, increasing water temperatures led to changing plankton community composition and abundance (Rice et al., 2015), altered fish populations (Lynch et al., 2015), and enhanced the dominance of cyanobacteria (Jöhnk et al., 2008). Changes in these populations affect the biodiversity of freshwater ecosystems (Mantyka-Pringle et al., 2014). Furthermore, increased thermal stratification of lakes can intensify lake anoxia (Palmer et al., 2014), enhance the growth of bloom-forming cyanobacteria (Paerl and Paul, 2012), and induce changes to internal nutrient loading and lake productivity (Verburg and Hecky, 2009). Variations in water temperature impact the distribution, behavior, community composition, reproduction, and evolutionary adaptations of organisms (Thomas et al., 2004). Further assessment of the response of lake water temperature to changes in air temperature and wind speed will improve our understanding of ecosystem response, which can better prepare management, adaptation, and mitigation efforts for a range of different size of lakes.

Lake morphometry can complicate the response of lake water temperatures to air temperature and wind speed changes by altering physical processes of wind mixing, water circulation, and heat storage (Adrian et al., 2009). Basin morphometric characteristics such as mean depth, surface area, and volume can strongly affect lake stratification (Butcher et al., 2015; Kraemer et al., 2015). Large surface areas increase the effects of vertical wind mixing, an important mechanism for transferring heat to the lake bottom (Rueda and Schladow, 2009) and thermocline shifts may be dampened in large lakes where the depth of the thermocline is constrained by the lake's fetch (Boehrer and Schultze, 2008; MacIntyre and Melack, 2010). Winslow *et al.*, (2015) showed that differences in wind-driven mixing may explain the inconsistent response of hypolimnetic temperatures between small and large lakes. While previous research efforts have investigated the response of individual lakes (Austin and Colman, 2007; Voutilainen et al., 2014) and the bulk response of lakes in a geographic region to changing climate (Kirillin, 2010; Magnuson et al., 1990), few studies have focused on elucidating the effects of morphometry, specifically lake depth and surface area, on changes in lake water temperature in response to long-term changes in air temperature and wind speed.

The purpose of this paper is to investigate the response of water temperatures and stratification in lakes with different morphometry (water depth and surface area) to changing air temperature and wind speed. A one-dimensional hydrodynamic lake-ice model, allowing for additional investigation into quantities that are not available in limnological records, was employed to run continuous long-term simulations of water temperature during open water and ice covered seasons of three lakes with different morphometry. These lakes vary in surface area and depth and were close enough to each other (<30 km distance) to experience similar daily average temperature, wind speed, solar radiation, cloud cover, and precipitation over the period 1911-2014. Long-term changes in water temperature (epilimnetic and hypolimnetic temperatures), stratification



variables (stratification onset, overturn, and duration), heat fluxes, and stability from both observations and model outputs were used to reveal how lake depth and surface area influence and alter thermal structure among the three study lakes.

## 2 Methods

### 2.1 Study sites

Three morphometrically different lakes, Lake Mendota, Fish Lake, and Lake Wingra, located near Madison, Wisconsin, United States of America (USA), were selected for this study. These lakes are chosen for (i) their morphometry differences, (ii) their close proximity to one another, and (iii) the availability of long-term limnological records for model calibration.

Lake Mendota (43°6' N; 89°24'W; Figure 1a; Table 1), is a dimictic, eutrophic, drainage lake in an urbanizing agricultural
watershed (Carpenter and Lathrop, 2008). The lake stratifies during the summer, and typical stratification periods lasts from May to September. During the summer months (1 June - 31 August), the mean surface water temperature is 22.4 °C, and hypolimnetic temperatures range in value from 11°C to 15 °C. Normal secchi depth during the summer is 3.0 meters (Lathrop et al., 1996). Fish Lake (43°17'N; 89°39'W; Figure 1b; Table 1) is a dimictic, eutrophic, shallow seepage lake located in northwestern Dane County. From 1966 to 2001, the water level of the lake rose by 2.75 meters (Krohelski et al., 2002). The
lake experiences summer stratification lasting from the beginning of May to mid-September. Mean surface water temperature 23.9°C and hypolimnetic temperatures are normally near 8°C during summer months; however, some years do not experience complete mixing in the spring and reach temperatures of only 4−5°C in the bottom waters by the end of the summer. The average Secchi depth during the summer months is 2.4 m. Lake Wingra (43°3' N; 89°26' W; Figure 1c; Table 1) is a very shallow, eutrophic, drainage lake. Due to its shallow depth, Lake Wingra does not experience thermal stratification in the
summer. During the summer, the mean water temperature is 23.9°C and mean secchi depth is 0.7 meters.

### 2.2 Data

  Meteorological data used in the model input consisted of daily solar radiation, air temperature, vapor pressure, wind speed, cloud cover, rainfall, and snowfall over a period of 104 years from 1911 to 2014. Air temperature, wind speed, vapor pressure, and cloud cover were computed as an average of the whole day, while solar radiation, rainfall, and snowfall were the daily
totals. Meteorological data was gathered from Robertson (1989), who compiled a continuous daily meteorological dataset for Madison Wisconsin from 1884 to 1988 by adjusting for changes in site location. Appended to this dataset is data from the National Climate Data Center weather station at the Dane County Regional Airport. All data other than solar radiation can be obtained from http://www.ncdc.noaa.gov/, for Madison (MSN), and solar radiation can be obtained from



http://www.sws.uiuc.edu/warm/weather/. Adjustments to wind speed were made based on changes in observational techniques occurring in 1996 (McKee et al., 2000) by comparing data from Dane County Airport with that collected from the Atmospheric and Oceanic Science Building instrumentation tower at the University of Wisconsin-Madison (http://ginsea.aos.wisc.edu/labs/mendota/index.htm). Detail of this adjustment can be found in Magee et al. (2016).

Seasonal Secchi depths were used to determine the light extinction coefficients. Lathrop et al. (1996) compiled Secchi depth data for Lake Mendota between 1900 and 1993 (1701 daily Secchi depth readings from 70 calendar years), and summarized the data for six seasonal periods: winter (ice-on to ice-out), spring turnover (ice-out to 10 May), early stratification (11 May to 29 June), summer (30 June to 2 September), destratification (3 September to 12 October), and fall turnover (13 October to

10 ice-on). After 1993, Secchi depths are obtained from the North Temperate Lake Long Term Ecological Research (NTL-LTER) program (https://portal.lternet.edu/nis/home.jsp#). For Fish Lake and Lake Wingra, Secchi depths were compiled for 1995 to the present from the NTL-LTER program. For years with no Secchi data, the long-term mean seasonal Secchi depths were used. Light extinction coefficients were estimated from Secchi depth using the equation from Williams *et al.,* (1980):

$$k = 1.1/z_s^{0.73} \qquad\qquad (1)$$

where $k$ is the light extinction coefficient and $z_s$ is the Secchi depth (m).

Inflow and outflow measurements were collected from gauging stations (http://waterdata.usgs.gov/wi/nwis/sw/), and complied to calculate daily totals. In cases where inflow and outflow measurements were not available, inflow and outflow were estimated as the residual unknown term of the water budget balancing precipitation, evaporation, and lake level. The residual

term was distributed evenly across the number of days between water level measurements. For Lake Mendota, water level was recorded since 1916 (http://waterdata.usgs.gov/wi/nwis/dv). Water level at Fish Lake was recorded almost daily from 1966-2003 (http://waterdata.usgs.gov/wi/nwis/dv/?site_no=05406050&agency_cd=USGS&referred_module=sw). For Lake Wingra, water level was recorded sporadically during the period of interest. When lake level information was unavailable, the long-term mean lake level was assumed for water budget calculations. Only Lake Mendota has inflowing surface water

streams. Inflow temperatures were estimated following the method in (Magee et al., 2016). Groundwater temperature measurements near each lake were used to estimate the temperature of groundwater fluxes.

Observation data used for model calibration came from a variety of sources. For Lake Mendota, long term water temperature records for Lake Mendota were collected from Robertson (1989) and the NTL-LTER (2012b). Ice thickness data were gathered

from E. Birge, University of Wisconsin (unpublished); D. Lathrop, Wisconsin Department of Natural Resources (unpublished);



Stewart (1965); and the NTL-LTER program (2012a). Frequency of temperature data varied from one or two profiles per year to several profiles for a given week. Additionally, the vertical resolution of the water profiles varied greatly. For Fish Lake and Lake Wingra, water temperature data were collected from NTL-LTER only from 1996-2014 (2012b).

### 2.3 Model description

The DYRESM-WQ (DYnamic REservoir Simulation Model-Water Quality) model (Hamilton and Schladow, 1997) employs discrete horizontal Lagrangian layers to simulate vertical water temperature, salinity, and density with input including inflows, outflows, and mixing (Imberger et al., 1978). A one-dimensional layer structure is adopted based on the vertical density stratification over horizontal density variations and destabilizing forces such as wind stress and surface cooling abbreviated to ensure a one dimensional structure (Antenucci and Imerito, 2003). Mixing and surface layer dynamics depend on a turbulent
kinetic energy budget and potential energy required for mixing (Hamilton and Schladow, 1997; Sherman et al., 1978). Hypolimnetic mixing is parameterized through a vertical eddy diffusion coefficient, which accounts for turbulence created by the damping of basin-scale internal waves on the bottom boundary and lake interior (Yeates and Imberger, 2003). More information on the simulation of water temperature and mixing can be found in Imberger and Patterson (1981) and Yeates and Imberger (2003).

The ice model added into the DYRESM-WQ model and called DYRESM-ICE model is based on the MLI model of Rogers et al., (1995), with the additions of two-way coupling of the hydrodynamic and ice models and time-dependent sediment heat flux for all horizontal layers. Details of the ice model can be found in Magee et al., (2016). The model assumes that the time scale for heat conduction through the ice is short relative to the time scale of meteorological forcing (Patterson and Hamblin,
1988; Rogers et al., 1995), an assumption which is valid with a Stefan number less than 0.1 (Hill and Kucera, 1983). Model inputs include lake hypsography, initial vertical profiles for water temperature and salinity, Secchi depth, meteorological variables, and inflows/outflows. The model calculates the surface heat fluxes using meteorological variables: total daily shortwave radiation, daily cloud cover, air vapor pressure, daily average wind speed, air temperature, and precipitation. During the entire simulation period, all parameters and coefficients are kept constant. The time step in the model
for calculating water temperature, water budget, and ice thickness is 1 hr. Snow ice compaction, snowfall and rainfall components are updated at a daily time step, corresponding to the frequency of meteorological data input. Cloud cover, air pressure, wind speed, and temperature are assumed constant throughout the day, and precipitation is assumed uniformly distributed. Shortwave radiation distribution throughout the day was computed based on the lake latitude and the Julian day. Model parameters and simulation specifications are identical for all three study lakes and can be found in Table 1 of Magee et



al. (2016). After calibrating the model, we run the simulation period for all three lakes over 104 years, starting on 7 April 1911 and ending on 31 October 2014 without termination.

## 2.4 Model calibration and evaluation

Using known inflows, outflow, and water elevation, the water balance was closed in the method described in Section 2.2 to match measured water levels where known and long term average water levels when elevation information was unknown. We assumed that evaporative water flux and heat flux were properly parameterized by the DYRESM-WQ-I model, although we did not validate model evaporation rates. Parameters used in the model were derived from literature values (Table 1; Magee et al., 2016) with the exception of estimation of a variable light extinction coefficient calculated from observed Secchi depth (see Sect. 2.2) and adjustment of the minimum layer thickness. To calibrate water temperature, minimum layer thickness was varied from 0.05 to 0.5 m in intervals of 0.025 m for the period 1995-2000 for all three lakes, similar to the method in Tanentzap et al. (2007) and Weinburger and Vetter (2012). One minimum layer thickness was chosen for all three lakes, and the final thickness was chosen to be 0.125 m as it minimized the overall deviation between simulated and observed temperature values for the three lakes.

Three statistical measures were used to evaluate model output against observational data (Table 2): absolute mean error (AME), root mean square error (RMSE), and Nash-Suttcliffe efficiencies (NS) were used to compare simulated and observed temperature values for volumetrically-averaged epilimnion temperature, volumetrically-averaged hypolimnion temperature, and all individual water temperature measurements for unique depth and sampling time combinations. Simulated and observed values are compared directly, with the exception of aggregation of water temperature measurements to daily intervals where sub-daily intervals are available. Water temperatures were evaluated for the full range of available data on each lake.

## 2.5 Analysis

In this study, the surface mixed layer depth was determined by using LakeAnalyzer analysis (Read et al., 2011). We quantified the resistance to mechanical mixing due to the potential energy in the stratified water column as the average summer (15 July to 15 August) Schmidt number for each lake based on Idso's version of Schmidt Stability (Idso, 1973). Linear regression was used to determine the trend of long-term changes in lake variables. Breakpoints in variables over the study period were determined using a piecewise linear regression (Magee et al., 2016; Ying et al., 2015). A sequential t-test (Rodionov, 2004; Rodionov and Overland, 2005) was used to detect abrupt changes in the mean value of lake variables. The variables were tested on data with trends removed using a threshold significance level of $p = 0.05$, a Huber weight parameter of $h = 2$, and a





cut-off length L = 10 years. Finally, the coherence of lake variables (Magnuson et al., 1990) between lake pairs was determined with a Pearson correlation coefficient (Baron and Caine, 2000).

## 3 Results

### 3.1 Changes in air temperature and wind speed

Both yearly average air temperatures and seasonal air temperatures increased over the period 1911−2014 (Figure 2a). Yearly air temperature increased at a rate of 0.145°C decade$^{-1}$ ($p<0.01$); winter air temperature increased at a rate of 0.225°C decade-1 ($p<0.01$); spring air temperature increased at a rate of 0.165°C decade$^{-1}$ ($p<0.01$); summer air temperature increased at a rate of 0.081°C decade$^{-1}$ ($p<0.05$); and fall air temperature increased at a rate of 0.110°C decade-1 ($p<0.05$). All five sets of data were further analysed for significant changes in slope and for abrupt changes in mean. Yearly average air temperature showed

a significant change in slope from 0.081°C decade$^{-1}$ to 0.334°C decade$^{-1}$ occurring in 1981, but seasonal changes in air temperature showed no significant changes in slope over the study period. Summer air temperatures did show significant abrupt changes in the mean value in 1930 from 18.88°C to 19.96 °C ($p <0.01$); in 1949 from 19.96°C to 19.37 °C ($p<0.05$); and in 2010 from 19.37 °C to 20.54°C ($p<0.05$).

Wind speeds for both yearly and seasonal average exhibited significant decreased in trend over the period 1911−2014 (Figure 2b). Yearly wind speed decreased at a rate of 0.073 m s$^{-1}$ decade$^{-1}$ ($p<0.01$); winter decreased at a rate of 0.083 m s$^{-1}$ decade$^{-1}$ ($p<0.01$); spring decreased at a rate of 0.071 m s$^{-1}$ decade$^{-1}$ ($p<0.01$); summer decreased at a rate of 0.048 m s$^{-1}$ decade$^{-1}$ ($p<0.01$); and fall decreased at a rate of 0.088 m s$^{-1}$ decade$^{-1}$ ($p<0.01$). Additionally, all five sets of wind speed data showed statistically significant abrupt changes in the mean value occurring in the mid-nineties. For yearly average wind speed, a shift

from 4.43 m s$^{-1}$ to 3.74 m s$^{-1}$ ($p<0.01$) occurred after 1994; for winter wind speeds, a shift from 4.72 m s$^{-1}$ to 3.92 m s$^{-1}$ ($p<0.01$) occurred after 1997; for spring wind speeds, a shift from 4.59 m s$^{-1}$ to 3.90 m s$^{-1}$ ($p<0.01$) occurred after 1996; for summer, a shift from 3.70 m s$^{-1}$ to 3.66 m s$^{-1}$ ($p<0.01$) occurred after 1994; and for fall, a shift from 4.64 m s$^{-1}$ to 3.75 m s$^{-1}$ ($p<0.01$) occurred after 1994.

### 3.2 Model evaluation

Model output including epilimnetic (Table 2), hypolimnetic (Table 2), and temperature at 1 m intervals (Figure 3; Table 2) for the three study lakes compared well with observations. The model was validated with all available data for all three lakes



during the period 1911−2014. AME and RMSE (for all variables were low and less than standard deviations for the variables. NS efficiencies were high (>0.85) and most above 0.90, indicating high model accuracy.

### 3.3 Water temperatures

Epilimnion for Lake Mendota (Figure 4a) and Fish Lake (Figure 4b) were defined as 0-10 m depth and 0-5 m depth,
respectively, based on the surface mixed layer depth from observation and model data using LakeAnalyzer analysis (Read et al., 2011). For Lake Wingra (Figure 4c), the whole water column was "epilimnetic" because the lake did not stratify during the summer months. Lake Mendota temperatures ranged from 19.65°C to 26.1°C (mean (M) = 22.8°C, standard deviation (SD) = 1.07°C, range (R) =6.4°C); Fish Lake temperatures ranged from 25.8°C to 19.0°C (M = 22.5°C, SD = 1.3, R = 6.7); Lake Wingra temperatures ranged from 27.5°C to 20.6°C (M = 23.8, SD = 1.27, R = 6.9). Lake Mendota and Lake Wingra had
similar increasing trends of 0.069°C decade$^{-1}$ and 0.079°C decade$^{-1}$, respectively, while Fish Lake had a larger trend increase of 0.138°C decade$^{-1}$ (Table 3). All three lakes have statistically significant ($p$<0.01) abrupt changes in mean values over the study period. For Lake Mendota, there is an abrupt change after 1930 from 22.09 °C to 22.99 °C. For Fish Lake there are three shifts: first after 1934 from 21.68°C to 22.50°C, then after 1995 from 22.50°C to 24.26°C, and finally in 2008 from 24.26°C to 22.14°C. For Lake Wingra, there is an abrupt change after 1930 from 23.13°C to 24.02°C. None of the three lakes show any
significant breakpoint in the trend of epilimnetic temperatures.

Hypolimnetic water temperatures for Lake Mendota (Figure 4d) and Fish Lake (Figure 4e) were defined as 20−25 m and 13−20 m, respectively, based upon the long term bottom of metalimnion depth calculated using LakeAnalyzer (Read et al., 2011). Hypolimnetic water temperatures for Lake Mendota ranged from 8.3°C to 16.7 °C (M = 12.2°C, SD = 1.7 °C, R = 8.4 °C);
Fish Lake temperatures ranged from 5.8°C to 13.8°C (M = 8.6°C, SD = 1.3°C; R = 8.0°C). Opposite to those of the epilimnion, Lake Mendota and Fish Lake both experienced statistically significant decreases in summer-time hypolimnetic water temperatures of 0.131°C decade$^{-1}$ and 0.083°C decade$^{-1}$, respectively (Table 3). The hypolimnetic heating from 15 July to 15 August was also calculated (Table 3), showing a range from 0.04 °C to 2.3°C (M = 0.84°C, SD = 0.37°C, R = 2.2°C) for Lake Mendota and a range from 0.17°C to 0.72°C (M = 0.48, SD = 0.11, R = 0.50) for Fish Lake. Neither lake has a significant
abrupt change in temperature nor a significant breakpoint in linear trend during the study period.

### 3.4 Stratification and stability

In this paper, summer stratification was characterized by 3 variables: stratification onset, fall overturn, and duration of stratification. The dates of onset of stratification and fall turnover were defined as the day when the surface-to-bottom





temperature difference was greater than (for stratification) or less than (for overturn) 2°C (Robertson and Ragotzkie, 1990). Since Lake Wingra did not experience seasonal stratification, only Lake Mendota and Fish Lake are considered here.

For stratification onset, Lake Mendota (Figure 5a) ranged from 15 April to 28 June (M = 20 May; SD = 15 days; R = 74 days)
and Fish Lake (Figure 5b) ranged from 19 March to 14 May (M = 24 April, SD = 8.2 days, R = 56 days). For fall overturn, Lake Mendota (Figure 5a) ranged from 31 July to 17-October (M = 21 September; SD = 11.4 days; R = 78 days) and Fish Lake (Figure 5b) ranged from 9 September to 6 November (M = 15 October, SD = 11.0 days, R = 56 days). Stratification duration for Lake Mendota (Figure 5c) ranged from 52 days to 165 days (M = 124.7, SD = 22.8, R = 113) and for Fish Lake (Figure 5d) ranged from 142 days to 203 days (M = 173.9, SD = 13.7, R = 61). Both lakes experienced earlier stratification
onset, later fall overturn, and longer stratification duration, with Lake Mendota having larger trends in all 3 variables (Table 3). For both lakes, there was a statistically significant ($p<0.01$) change in the long term mean of 13.3 days earlier occurring after 1994 for Lake Mendota and of 15.1 days earlier occurring after 1993 for Fish Lake. Stratification duration in Lake Mendota exhibited a significant change in trend from 0.067 days earlier decade$^{-1}$ to 4.5 days earlier decade$^{-1}$ after 1940. Similarly, stratification duration in Fish Lake exhibited a significant change in trend from 0.19 days later decade$^{-1}$ to 9.6 days
earlier decade$^{-1}$ after 1981.

For lake stability, Lake Wingra had an average Schmidt stability value near 0 (Figure 6), indicating that the lake was easily mixed and polymictic during the period. In contrast, both Lake Mendota and Fish Lake had significantly higher stability values (Figure 6) and both lakes were stratified and more resistant to mixing. While the shallow lake Wingra showed no trend (Table 3),
Lake Mendota and Fish Lake exhibited statistically significant changes in trend. Furthermore, Lake Mendota had a larger number than Fish Lake (Figure 6). A larger trend was also observed in Lake Mendota (Table 3) possibly due to both a larger change in stratification variables and changing hypolimnion temperature, increasing stability. There was no significant abrupt shift or change in trend for any of the three lakes during the study period.

### 3.5 Surface heat fluxes

The modelled surface heat fluxes (Figure 7) including (a) net shortwave radiative flux; (b) net longwave radiative flux; (c) sensible heat flux; (d) latent heat flux; and (e) total heat flux on over the 104-year period on the three study lakes are examined here. While there is no statistically significant trend in shortwave flux, latent flux, or total heat flux, figure 7c shows longwave heat flux exhibits trend toward larger magnitude flux (decreasing absolute value; -5.85 J m$^{-2}$ for Lake Mendota, -5.80 J m$^{-2}$ for Fish Lake, and -4.59 J m$^{-2}$ for Lake Wingra, $p<0.05$ for all three lakes) and sensible heat fluxes displays an increasing trend
(less negative values; 4.10 J m$^{-2}$ for Lake Mendota, 3.65 J m$^{-2}$ for Fish Lake, and 5.65 J m$^{-2}$ for Lake Wingra, $p<0.05$ for all



three lakes). For shortwave radiation, an abrupt change from 1.14 J m$^{-2}$ to -4.92 J m$^{-2}$ occurred in Lake Mendota after 1992($p<0.01$); a change from -1.97 J m$^{-2}$ to 3.34 J m$^{-2}$ after 1945 ($p<0.01$) and from 3.34 J m$^{-2}$ to -4.84 J m$^{-2}$ after 1992 ($p<0.01$) for Fish Lake; and from -4.34 J m$^{-2}$ to 4.07 J m$^{-2}$ after 1937 ($p<0.01$) and from 4.07 J m$^{-2}$ to -5.98 J m$^{-2}$ after 1992 ($p<0.01$) for Lake Wingra. For longwave radiation, multiple abrupt changes occurred in all three lakes. For Lake Mendota: -2.09 J m$^{-2}$ to

2.54 J m$^{-2}$ after 1923 ($p<0.01$), 2.54 J m$^{-2}$ to -0.68 J m$^{-2}$ after 1937 ($p<0.01$), -0.68 J m$^{-2}$ to 2.60 J m$^{-2}$ after 1981 ($p<0.01$), and 2.60 J m$^{-2}$ to -0.86 J m$^{-2}$ after 1998 ($p<0.01$). For Fish Lake: -2.23 J m$^{-2}$ to 2.53 J m$^{-2}$ after 1923 ($p<0.01$), and 2.53 J m$^{-2}$ to 0.10 J m$^{-2}$ after 1937 ($p<0.01$). For Lake Wingra: -1.98 J m$^{-2}$ to 3.09 J m$^{-2}$ after 1924 ($p<0.01$), 3.09 J m$^{-2}$ to -0.81 J m$^{-2}$ after 1937 ($p<0.01$), and -0.81 J m$^{-2}$ to 1.42 J m$^{-2}$ after 1981 ($p<0.01$). Sensible heat flux had an abrupt change after 1926 for Lake Mendota (-1.67 J m$^{-2}$ to 0.26 J m$^{-2}$, $p<0.01$) and after 1921 for both Fish Lake (-2.23 J m$^{-2}$ to 0.26 J m$^{-2}$, $p<0.01$) and Lake

Wingra (-3.68 J m$^{-2}$ to 0.36 J m$^{-2}$, $p<0.01$). While the timing of the abrupt change was similar in all three lakes, the magnitude of the change appears to increase with lake depth. Latent heat flux shows statistically significant ($p<0.01$) changes in mean after 1926 (Lake Mendota 5.78 J m$^{-2}$ to -2.32 J m$^{-2}$; Fish Lake 5.43 J m$^{-2}$ to -2.14 J m$^{-2}$; Lake Wingra 6.42 J m$^{-2}$ to -2.70 J m$^{-2}$) and 1996 (Lake Mendota -2.32 J m$^{-2}$ to 5.22 J m$^{-2}$; Fish Lake -2.14 J m$^{-2}$ to 4.77 J m$^{-2}$; Lake Wingra -2.70 J m$^{-2}$ to 6.20 J m$^{-2}$) for all three lakes. Abrupt changes in latent heat flux appear to be independent of lake morphometry, suggesting that the

timing of change may be primarily driven by climate. Net heat flux shows no significant abrupt change for Lake Mendota, two changes for Fish Lake (-0.33 J m$^{-2}$ to 4.00 J m$^{-2}$ after 1964, $p<0.01$, and 4.00 J m$^{-2}$ to -0.55 J m$^{-2}$ after 1975, $p<0.01$), and one change for Lake Wingra (-1.15 J m$^{-2}$ to 0.28 J m$^{-2}$ after 1930, $p<0.01$). Differences in magnitude and timing of abrupt changes in shortwave, longwave, and net heat fluxes emphasize that morphometry may play a role, it is unclear how or what the specific role may be.

**4 Discussion**

**4.1 Model performance and comparison**

The DYRESM-WQ-ICE model reliably simulated water temperatures over long-term (1911-2014) simulations (Figure 3, Table 3). Deviation between measurement and observed temperatures was attributed to the input averaging, particularly daily averaging of air temperature and wind speed. Discrepancies between modelled and measured values came in part from

differences in location and sampling frequency of observations. Errors in water temperature were attributed to differences between simulated and observed thermocline depth over some years. In general, thermocline depths were within 1 m between observed and simulated, but some years differ by as much as 2.5 m.





Overall, the performance of the DYRESM-WQ-ICE model was similar to that of other studies in the literature. Perroud et al. (2009) performed a comparison of one-dimensional lake models on Lake Geneva, and RMSE for water temperature were as high as 2°C for the Hostetler model (Hostetler and Bartlein, 1990), 1.7°C for DYRESM (Tanentzap et al., 2007), 2°C for SIMSTRAT, and 4°C for Freshwater Lake (FLake) model (Golosov et al., 2007; Kirillin et al., 2012). For all four models, errors were lower in the upper layers and larger in the bottom of the water column (Perroud et al., 2009), similar to errors found in this study. Fang and Stefan (1996) gave standard errors of water temperature of 1.37°C for the open water season and 1.07°C for the total simulation period for Thrush Lake, MN, similar to those found here. Results of Nash-Sutcliff efficiency coefficients for all 3 study lakes were within the ranges found in Yao et al. (2014) for the Simple Lake Model (SIM; Jöhnk et al., 2008), Hostetler (Hostetler and Bartlein, 1990), Minlake (Fang and Stefan, 1996), and General Lake Model (GLM; Hipsey et al., 2014) models for Harp Lake, Ontario, Canada water temperatures.

Limitations in the model and simulations presented here arise from uncertainties in observations and model parameters and the assumption of one-dimensionality in both the model and field data. Quantifying this type of uncertainty is extremely difficulty (Gal et al., 2014; Tebaldi et al., 2005). Generally, small, stratified lakes lack large horizontal temperature gradients (Imberger and Patterson, 1981), allowing the assumption of one-dimensionality to be appropriate. However, short-term deviations in water temperature and thermocline depth may exist due to internal wave activity, especially in larger lakes (Tanentzap et al., 2007); spatial variations in wind stress can produce horizontal variations in temperature profiles (Imberger and Parker, 1985). Neither of which are captured in the one-dimensional model approach nor by the collection of observation data at a single in-lake location. Furthermore, light extinction estimated from Secchi depths can have a large degree of measurement uncertainty (Smith and Hoover, 2000), leading to uncertainty in water temperature simulations (Hocking and Straškraba, 1999). Locations and techniques of meteorological measurements changed at various times throughout the 104 years study period. We have made significant efforts in adjustments to limit uncertainty and errors associated with these changes. While inflow and outflow measurements were assessed by the USGS for quality assurance and control, uncertainty for both quantity and water temperature is unknown, especially in consideration of having to fill in missing data to fully simulate the time period.

Overall, the effects of many uncertainties on simulated temperatures may not be large as the inflow and outflow are small in comparison to lake volume. Model parameters used to characterize the lake hydrodynamics were taken from literature values. These values may be expected to have some small variability between lakes since previous studies have shown that many of the hydrodynamic parameters are insensitive to changes of ±10% (Tanentzap et al., 2007). In this study, the model was validated against an independent dataset for each lake to determine if the model fits measured data and functions adequately,





with errors within the range of those from other studies. The combination of uncertainties in parameters and observed data may be high; however, as all parameters and observational methods were kept consistent among the three lakes, the validity of the model in predicting differences among the three lake types is adequate. We reason that the model accuracy is sufficient to meet the objectives of identifying morphometry-caused differences in lake response for both past and future climate changes.

**4.2 Coherence among lakes**

Temporal coherence, the similarity of lake responses over time, shows that adjacent lakes respond coherently to climate (Magnuson et al., 1990; Thompson et al., 2005). Furthermore, lakes with comparable physical features exhibit higher coherence than lakes with different physical properties (Novikmec et al., 2013). Large correlation coefficients, indicative of high temporal coherence between lakes, are largely due to synchronous patterns in lake variables driven by climate (Magnuson

et al., 1990; Palmer et al., 2014). In this study, the 3 lakes were formed into 3 distinct pairs for comparing the coherence of physical lake variables. Pair 1, Lake Mendota and Fish Lake, have similar depths but different surface areas, illustrating the effects of surface area differences. Pair 2, Lake Wingra and Fish Lake, have similar surface areas, but shallow and deep water depths, addressing the effects of lake depth. Pair 3, Lake Mendota and Lake Wingra, have both differing surface areas and water depths. Pearson correlation coefficients (Table 4) in lake variables were calculated for pairs of the study lakes. This

method allows us to easily identify coherence differences that may be driven by lake surface area or lake depth while simultaneously accounting for differences in climate that may impact the results of similar analysis covering lakes over a broad region.

Epilimnetic temperature exhibited high coherence for the three lake pairs (Table 4), suggesting that inter-annual variability in

epilimnion temperatures was primarily driven by climate drivers such as air temperature and wind speed. Specifically, the Mendota/Wingra pair has the highest correlation and Mendota and Wingra differ significantly in both depth and surface area. Furthermore, comparing the Mendota/Fish pair with similar depth and the Fish/Wingra pair with similar surface area suggests that both surface area and depth impact coherence between lake pairs; and surface area differences may drive asynchronous patterns to a greater extent than does depth differences for epilimnetic water temperature. The lower correlation for the

Mendota/Fish and Fish/Wingra pairs of lakes may be due to the difference in abrupt changes for Fish Lake epilimnion temperature in comparison to the other two lakes. Likely, the large change in lake depth from the period 1966−2001 (Krohelski et al., 2002) may be impacting the coherence between Fish Lake and the other two lakes, which have had relatively little year-to-year variation in water levels over the study period.





Hypolimnion temperature, different from epilimnion temperature, showed only moderate coherence for the Lake Mendota and Fish Lake pair (Table 4), suggesting that inter-annual variability in hypolimnion water temperatures was driven in part by factors other than climate, such as lake morphometry. For example, differences in thermocline depth (~10 m in Lake Mendota and ~6 m in Fish Lake) can play a role in filtering the climate signals into the hypolimnion temperature. This result is consistent

with other studies that show lake morphometry parameters affect the time of climatic signals, especially temperature stored in the lake system (Thompson et al., 2005). Other factors like strength of stratification and fetch differences may drive differences in the timing of stratification, further affecting hypolimnetic temperatures. Moreover, Arvola (2009) showed that hypolimnia temperatures were primarily determined by the conditions that pertained during the previous spring turnover. In our study, the relatively low hypolimnetic coherence (Table 4) suggests that lake morphometry plays a role in hypolimnion temperatures.

Coherence for stratification onset and fall overturn dates were low for the Mendota/Fish Lake pair (Table 4), suggesting that surface area, not air temperature or wind speed, was the main factor driving differences in stratification onset and overturn. Schmidt stability showed high coherence for the Mendota/Fish lake pair, but low coherence between the Wingra/Fish and Mendota/Wingra lake pairs, suggesting that lake depth drives differences in coherence, while surface area has a lesser role. High coherence between the Mendota/Fish pair suggests that climate drives stability when comparing lakes of similar depth.

Low coherence between the other two pairs suggests that lakes with different depths may have asynchronous behavior. Slightly lower coherence for the Mendota/Wingra pair than the Wingra/Fish pair suggests that lake surface area may also play a minor role in asynchronous behavior.

### 4.3 Sensitivity to changes in air temperature and wind speed

To determine the sensitivity of lake water temperature and stratification in response to air temperature and wind speed, we

perturbed these drivers across the range of -10°C to +10°C in 1°C temperature increments and 70% to 130% of the historical value in 5% increments, respectively. For each scenario, meteorological inputs remained the same as for the original simulation and snowfall (rainfall) conversion if the air temperature scenarios increase (decrease) above 0°C. Similarly, the water balance is maintained so that the long-term water levels in both lakes matches the historical record. Results of lake response to all perturbation scenarios will be discussed in the following.

### 4.3.1 Stratification onset

For both Lake Mendota and Fish Lake (Figure 8a and b), increasing (decreasing) air temperature resulted in earlier (later) stratification onset. Lake Mendota exhibited a linear trend of 2.0 days earlier (later) stratification for each degree (C) increase (decrease) in air temperature. Fish Lake, however, shows a nonlinear change in stratification onset with changes in air temperatures of 1.5 days earlier stratification for each degree (C) increase in air temperature but 2.7 days later stratification





for each degree (C) decrease in air temperature from the historical condition. Standard deviations in stratification onset on Lake Mendota remained fairly consistent, ranging from 15.5 to 18 days. In contrast, the standard deviation in stratification onset for Fish Lake decreased from 17.5 days to 12 days as air temperature increased. This may be due to an early limit in stratification onset for Fish Lake, thus reducing the variability of onset dates with increasing air temperatures. The above

results suggest that lake surface area can complicate the response of stratification onset to changes in air temperatures. For both Lake Mendota and Fish Lake (Figure 8c and d), decreased (increased) wind speed results in earlier (later) stratification onset, however the change is nonlinear. For Lake Mendota each 1m s$^{-1}$ decrease in wind speed results in 3.4 days earlier stratification onset and each 1m s$^{-1}$ increase in wind speed results in 10.5 days later stratification onset; meanwhile, Fish Lake shows 3.6 days earlier stratification onset for each 1m s$^{-1}$ decrease in wind speed and 8.1 days later stratification onset for each

1m s$^{-1}$ increase in wind speed. Standard deviations in both lakes see large decreases (increases) with decreasing (increasing) wind speed. Standard deviation changes from 20 days at 130 % of historical wind speed to 12 days at 70% of historical wind speed for Lake Mendota and from 15.6 days at 130 % of historical wind speed to 8.7 days at 70 % of historical wind speed for Fish Lake. As wind speed decreases (increases), the likelihood of the wind-induced kinetic energy being sufficient to mix the lake also decreases (increases). Additionally, the number of higher wind events is decreased (increased) under this scenario,

leading to less (more) kinetic energy available to mix the lake later (earlier) in the season. The change in stratification onset date for both lakes is nonlinear, but Lake Mendota experiences a greater difference between decreasing and increasing wind speeds due to the large surface area of the lake increasing the nonlinear response of thermal structure to wind speed changes. Additionally, standard deviations are much larger for Lake Mendota because the large fetch of the lake causes greater variability in wind stress than for the smaller Fish Lake.

**4.3.2 Fall overturn**

Lake Mendota (Figure 9a) shows a linear change in stratification overturn such that as air temperature increases (decreases) stratification overturn is 0.68 days later (earlier) with each degree (C) increase (decrease) in air temperature. For Fish Lake (Figure 9b), the change is nonlinear, with increases in air temperature causing a 1.81 days later change in stratification overturn for each degree (C) increase in air temperature, but a changes of only 0.77 day per degree (C) for decreases in air temperature

from the historical condition. Standard deviation for Lake Mendota slightly decreased from 14 days to 11 days as air temperature increased from -8°C to +8°C change from the historical condition, and Fish Lake had a consistent standard deviation of 13 days (±0.75 days). Overall, for lakes with different surface areas, it appears air temperature changes have a limited impact on the variability of the stratification overturn dates to changing climate, and a larger impact on the average date of stratification overturn. For wind speed perturbations, both lakes show a nonlinear change for later (earlier) stratification

overturn with decreases (increases) in wind speed. For Lake Mendota (Figure 9c), decreases in wind speed cause a change of



13.9 days later with each 1 m s$^{-1}$ decrease in wind speed and a change of 17.1 days earlier with each 1 m s$^{-1}$ increase in wind speed. For Fish Lake (Figure 9d), decreases in wind speed cause a change of 16.4 days later with each 1 m s$^{-1}$ decrease in wind speed and a change of 8.5 days earlier with each 1 m s$^{-1}$ increase in wind speed. This result suggests that lakes with large surface area, such as Lake Mendota are more sensitive to changing stratification overturn dates as wind speed decreases

(increases) than lakes with smaller surface areas. As with stratification onset, decreasing (increasing) wind speeds decrease (increase) variability in overturn dates (27.6 to 10.6 days for Lake Mendota and 15.1 to 9.2 for Fish Lake). Fish Lake may have a much smaller change in standard deviation than for Lake Mendota because wind speed is a more dominant driver in Mendota than in Fish Lake, due to the difference in surface area between the two lakes.

### 4.3.3 Hypolimnetic water temperature

For Lake Mendota (Figure 10a), each degree (C) increase (decrease) in air temperature resulted in a linear change of 0.18°C increase (decrease) in hypolimnetic temperature. For Fish Lake (Figure 10b), increases in air temperature over the historical result in a water temperature increase of 0.25°C for each degree (C) of air temperature increase, and decreases in air temperature result in a water temperature decrease of 0.18°C for each degree (C) of air temperature decrease. Standard deviations for Lake Mendota and Fish Lake remain consistent with increasing (decreasing) temperature and range from

approximately 2.3°C to 2.7°C for Lake Mendota and from 1.7°C to 2.2°C for Fish Lake. Changes in air temperature alter the mean hypolimnetic temperature in both lakes, but does not affect the variability of hypolimnetic temperatures. For wind speed, Lake Mendota (Figure 10c) experiences a nonlinear change in hypolimnetic temperature such that for decreasing wind speed, the water temperature decreases at a rate of 1.1°C for each m s$^{-1}$ decrease in wind speed and for increasing wind speed, the water temperature increases at a rate of 1.8°C for each m s$^{-1}$ increase in wind speed. For Fish Lake (Figure 10d), the

hypolimnetic temperature also shows a nonlinear change; the water temperature decreases at a rate of 1.2°C for each m s$^{-1}$ decrease in wind speed and for increasing wind speed, the water temperature increases at a rate of 0.8°C for each m s$^{-1}$ increase in wind speed. Standard deviation in Lake Mendota decreased (increased) with decreasing (increasing) wind speeds, changing from 2.6°C at 130 % of historical wind speed to 1.8 °C at 70 % of historical wind speeds, but standard deviation in Fish Lake remained fairly constant over the perturbation scenarios, ranging from 1.3°C to 1.6°C. This indicates that wind speed changes

have a much larger impact on the variability of hypolimnetic temperatures for the larger surface area lake than for smaller surface area lake. Overall, the above results of the increasing temperature perturbation show increasing hypolimnetic water temperature, while decreasing wind speed perturbations show decreasing hypolimnetic water temperatures. Historical climate (Figure 4d and 4e) indicate that hypolimnetic temperatures are decreasing. Combining the effects of air temperature and wind speed, it suggests that wind speed decreases are a larger driver to hypolimnetic water temperature changes than increasing air

temperatures for both lakes. For example, in Lake Mendota, a 5% decrease in wind speed will offset the impacts to



hypolimnetic temperature of a 1°C increase in air temperature, while in Fish Lake, a 12-13% decrease in wind speed is necessary to offset the effects of a 1°C increase in air temperature. In other words, lakes with larger surface areas that also experience decreasing wind speeds may be more resilient to changing hypolimnion temperatures as a result of warmer air temperatures.

## 4.4 Role of morphometry on water temperature and stratification

### 4.4.1 Lake depth

Lakes with different depths (e.g., Lake Wingra and Fish Lake) responded differently to climate change. In this study, Lake Wingra, the shallowest of the three, did not stratify, while the deeper lakes, Lake Mendota and Fish Lake, did. Additionally, results show increased Schmidt Stability over the long term for Fish Lake and Lake Mendota (Table 3), but no trend in Lake Wingra. Indeed, (Kraemer et al., 2015) showed that mean lake depths can explain the most variation in stratification trends and lakes with greater mean depths have larger changes in their stability, consistent with our results for Lake Mendota and Fish Lake (Table 3). Due to lower heat capacity, shallow lakes respond more directly to short-term variations in the weather (Arvola et al., 2009), and heat can be transferred throughout the water column by wind mixing (Nõges et al., 2011). Deep lakes have a higher heat capacity so that greater wind speeds are required to completely mix the lake during the summer months, resulting in more temperature stability and higher Schmidt stability values for deeper Lake Mendota and Fish Lake. For radiative fluxes at the surface of the lake, shallow Lake Wingra had a similar magnitude of shortwave (Figure 7a), longwave (Figure 7b) and sensible heat (Figure 7c) fluxes as Lake Mendota and Fish Lake, but relatively larger magnitude of latent (Figure 7d) and net heat fluxes (Figure 7e). The result indicates that lake depth can play a large role in the magnitude of latent heat fluxes as shallow lakes have larger latent heat flux and thus more evaporation, possibly due to the overall warmer temperatures throughout the water column compared to lakes with cool bottom waters. Additionally, the magnitude of abrupt changes in sensible heat flux appear to be influences by water depth, with increasing depth decreasing the magnitude of shift in mean sensible heat flux after the abrupt change.

### 4.4.2 Surface area

Lake size can alter the effects of climate changes with the increasing air temperature and decreasing wind speeds on increasing epilimnetic water temperatures in Fish Lake and Lake Mendota. Air temperature, responsible for heat transfer between the atmosphere and lake, is the main driver to epilimnetic water temperatures (Boehrer and Schultze, 2008; Palmer et al., 2014). While increasing air temperatures are well documented to increase epilimnetic water temperatures (Livingstone, 2003; Robertson and Ragotzkie, 1990), the exact relationship is nontrivial (Robertson and Ragotzkie, 1990). Wind mixing, a more dominant mechanism of heat transfer (Nõges et al., 2011), can act to dampen the effects of air temperature increase and cool





the epilimnion through increased surface mixed-layer deepening. As a result, decreasing wind speeds increase epilimnion water temperatures (Figure 4 and Table 3). Nevertheless, larger fetch increases mixing and vertical transfer of heat to bottom waters, reducing epilimnion water temperatures (Boehrer and Schultze, 2008) and increasing the rate of lake cooling (Nõges et al., 2011). For this reason, Lake Mendota with the large fetch experiences a smaller increase in epilimnetic water temperature

compared to Fish Lake (Table 5). Trend and variability of Schmidt stability may also be affected by lake size. Compared with Fish Lake, Lake Mendota with a significantly larger fetch experiences greater variability in Schmidt stability that exhibits greater magnitude changes when compared to Fish Lake (Figure 6).

Sensitivity results by perturbation climate drivers indicate that lake surface area plays a role in the nonlinear response and

variability of stratification onset, stratification overturn, and hypolimnetic water temperatures to changes in wind speed. The magnitude of the nonlinear change and change in variability is larger for Lake Mendota than Fish Lake. The larger surface area, and resulting larger fetch, for Lake Mendota causes the increased nonlinearity of response and increased variability. Larger fetch for Lake Mendota causes stronger wind stress on the water surface when compared to Fish Lake, and the change in stress with increases or decreases in wind speed is nonlinear. Larger wind speeds furthermore result in more variability of

wind stress in lakes with larger surface areas and the resulting change in turbulence is also nonlinear. Results in this study indicate that lakes with larger surface areas will have a more nonlinear response to changes in wind speed than lakes with smaller surface areas for stratification onset (Figure 8), fall overturn (Figure 9), and hypolimnetic water temperature (Figure 10).

**5 Conclusion**

Study results show for three lakes with differing morphometry, the combination of increasing air temperatures and decreasing wind speeds yields warmer epilimnion temperatures, lower hypolimnion water temperatures, earlier stratification, later fall overturn, increased stratification duration, decreased hypolimnetic heating, and increased stability. Increased stratification durations and stability may have lasting impacts on fish populations (Gunn, 2002; Jiang et al., 2012; Sharma et al., 2011) and warmer epilimnion temperatures affects the phytoplankton community (Francis et al., 2014; Rice et al., 2015). Results indicate

that over the historical climate, smaller surface area influences wind-mixing, while larger and deeper lakes appear to respond more readily to changes in climate. Additionally, differences in stability between the larger Lake Mendota and smaller Fish Lake suggest that stability in lakes with larger surface areas are more variable than those with small surface areas. Climate perturbations support these historical results and provide additional insight on the individual and combine effects of air temperature increases and changes in wind speed. Increasing air temperature and decreasing wind speeds have a combined



effect toward longer stratification duration. Wind speed specifically plays a more dominant role in stratification onset and overturn and hypolimnetic water temperatures, indicating that air temperature increases are not the only climate variable that managers should plan for any mitigation and adaptation efforts. Previous research has shown uncertainty in the changes in hypolimnion water temperatures for dimictic lakes, however the perturbation scenarios indicate that while increasing air

temperature always increases hypolimnion temperature, wind speed is a larger driving force, and the ultimate hypolimnion temperature response may be primarily determined by whether the lake experiences an increase or decrease in wind speed.

**Acknowledgements**

The authors specifically acknowledge Yi-Fang Hsieh for further developing an ice module into the DYRESM-WQ model that was originally provided by David Hamilton. We would like to thank Dale Robertson at Wisconsin USGS and Richard (Dick)

Lathrop at Center of Limology for providing valuable long-term observation data for the three study lakes. In addition, we thank John Magnuson at Center of Limnology for his insightful suggestions and valuable comments regarding climate change on ice. Research funding was provided in part by the U.S. National Science Foundation Long-Term Ecological Research Program, University of Wisconsin (UW) Water Resources Institute USGS 104(B) Research Project, and UW Office of Sustainability SIRE Award Program. In addition, funding support for the first author by the College of Engineering Grainer

Wisconsin Distinguished Graduate Fellowship is acknowledged.



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



**Table 1: Morphometric and hydrologic characteristics of the three study lakes**

|  | Lake Mendota | Fish Lake | Lake Wingra |
|---|---|---|---|
| **Mean Depth (m)** | 12.8 | 6.6 | 2.7 |
| **Max Depth (m)** | 25.3 | 18.9 | 4.7 |
| **Surface Area (ha)** | 3937.7 | 87.4 | 139.6 |
| **Shoreline Length (km)** | 33.8 | 4.3 | 5.9 |
| **Groundwater** | Groundwater Discharge | Groundwater Flowthrough | Groundwater Flowthrough |
| **Surface Water** | Drainage | Seepage | Drainage |
| **Groundwater Input (%)** | 30 | 6 | 35 |



**Table 2: Absolute mean error (AME), root-mean square error (RMSE), and Nash-Sutcliff efficiency (NS) for water temperature variables on Lake Mendota, Lake Wingra, and Fish Lake. n = number of measurements, N/A represents errors that cannot be determined because Lake Wingra is a polymictic lake and does not have an epilimnion or hypolimnion.**

| Variable | Lake Mendota | | | | Fish Lake | | | | Lake Wingra | | | |
|---|---|---|---|---|---|---|---|---|---|---|---|---|
| | n | AME | RMSE | NS | n | AME | RMSE | NS | n | AME | RMSE | NS |
| Epilimnetic temperature (°C) | 3,239 | 0.69 | 0.3 | 0.99 | 263 | 1.23 | 1.45 | 0.95 | N/A | N/A | N/A | N/A |
| Hypolimnetic temperature (°C) | 3,239 | 1.04 | 0.53 | 0.96 | 263 | 1.63 | 1.94 | 0.92 | N/A | N/A | N/A | N/A |
| temperature at 1m interval (°C) overall range of values for depths | 85,566 | 0.5-1.56 | 0.25-0.75 | 0.95-0.99 | 5,522 | 0.85-1.93 | 1.98-2.42 | 0.85-0.91 | 1,897 | 0.63-0.85 | 0.41-0.96 | 0.99 |



**Table 3: Trends and in lake physical variables for the 3 studied lakes from 1911-2014. Trends are represented as units decade[-1].**

|  | Lake Mendota | Fish Lake | Lake Wingra |
|---|---|---|---|
| **Summer Epilimnetic Temperature** (°C) | + 0. 069$^\Delta$ | + 0.138* | + 0.079* |
| **Summer Hypolimnetic Temperature** (°C) | - 0.131* | - 0.083* | N/A |
| **Stratification Onset (days)** | 1.15 days earlier* | 0.81 days earlier* | N/A |
| **Fall Overturn (days)** | 1.18 days later* | 1.05 days later* | N/A |
| **Stratification Duration (days)** | + 2.68* | + 1.86* | N/A |
| **Hypolimnetic heating** (°C) | - 0.011* | -0.0011* | N/A |
| **Summer Schmidt stability number (J m$^{-2}$)** | +11.7* | +1.44* | no trend |

*indicates significant to $p<0.05$, $^\Delta$ indicates significant to $p<0.1$





**Table 4: Correlation coefficients for lake pairs open water lake variables**

| Lake Variable | Mendota/Fish | Lake Pair Wingra/Fish | Mendota/Wingra |
|---|---|---|---|
| Epilimnion Temperature | 0.605 | 0.742 | 0.804 |
| Hypolimnion Temperature | 0.482 | N/A | N/A |
| Stratification Onset | 0.260 | N/A | N/A |
| Fall Overturn | 0.388 | N/A | N/A |
| Schmidt Stability Number | 0.761 | 0.405 | 0.346 |



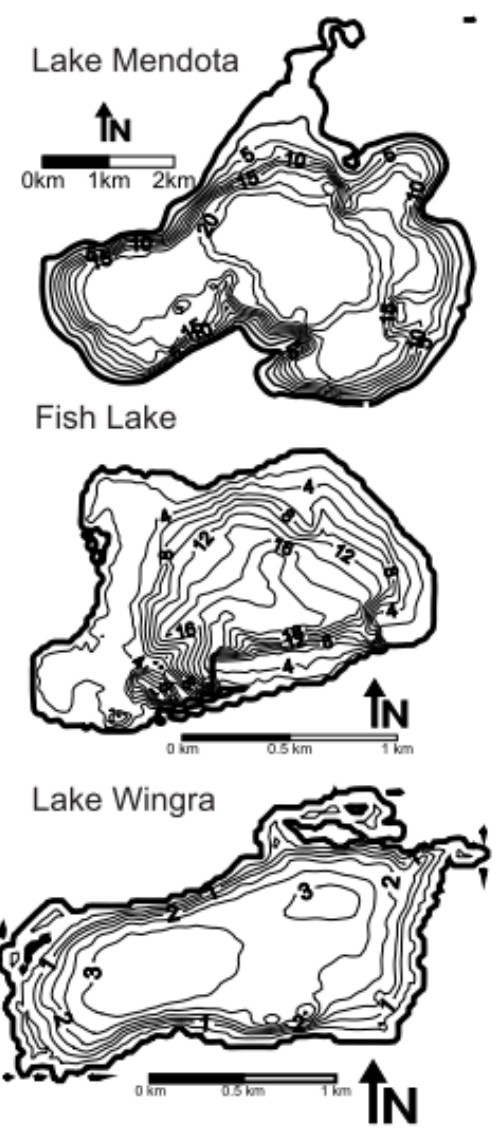

**Figure 1: Bathymetric maps of Lake Mendota, Fish Lake, and Lake Wingra**





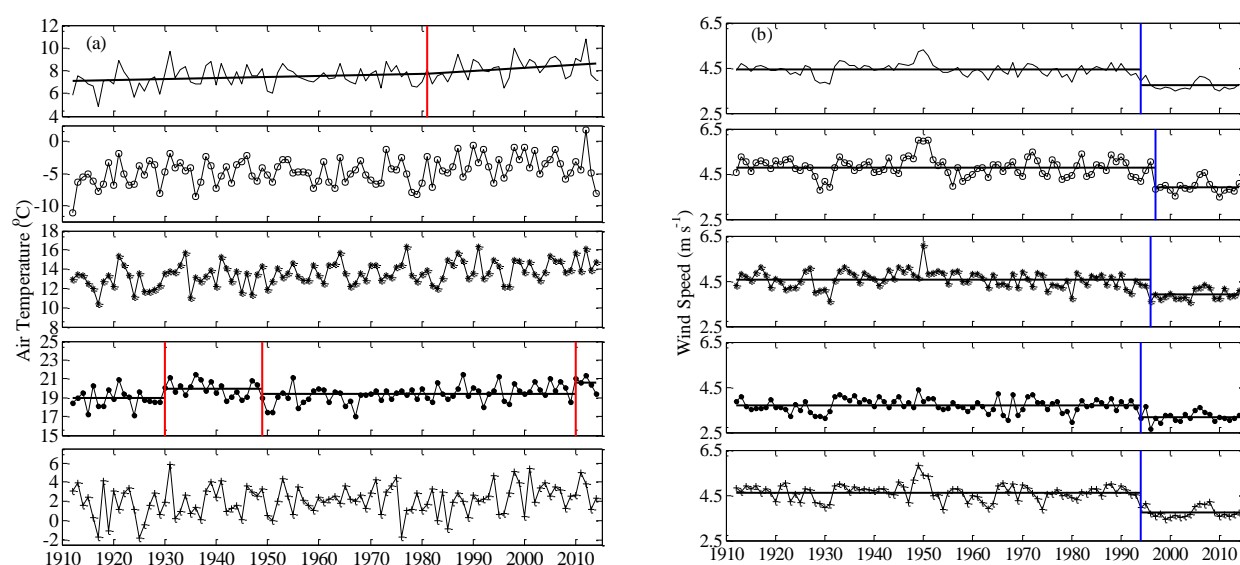

**Figure 2: Yearly (solid line), winter (open circle), spring (asterisk), summer (solid circles), and fall (cross) (a) air temperature and (b) wind speeds for Madison, WI, USA. Red line in yearly air temperature figure represents a breakpoint in the trend of average air temperature increase from 0.081° C decade-1 to 0.334 °C decade-1 occurring in 1981. Red lines in summer air temperature figure represents abrupt changes in average summer air temperature occurring in 1930, 1949, and 2010. Blue lines in wind speed figures represent abrupt changes in average wind speed occurring in each season and in the overall yearly wind speeds. Yearly wind speed change in 1994; winter in 1997; spring in 1996; summer in 1994; and fall in 1994.**





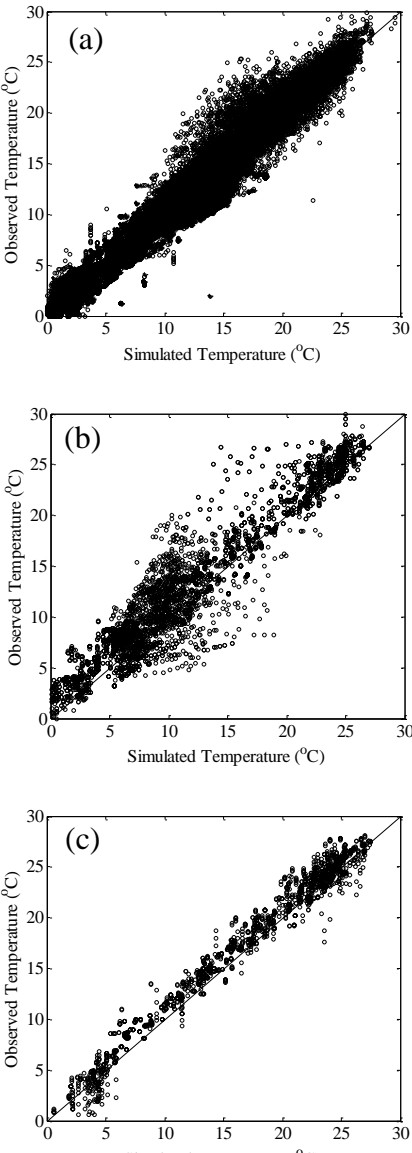

**Figure 3: Comparison of observed and simulated water temperatures for (a) Lake Mendota, (b) Fish Lake, and (c) Lake Wingra. Each point represents one observation vs. simulation pair with unique date and lake depth.**





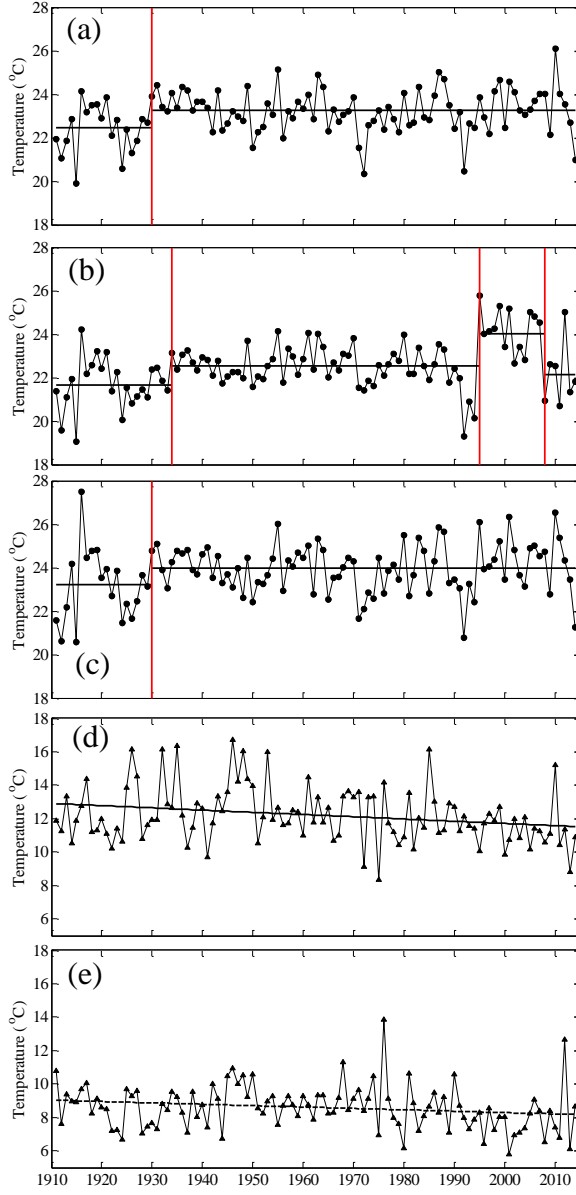

**Figure 4: Mean summertime (July15-August15) epilimnetic temperatures for (a) Lake Mendota, (b) Fish Lake, and (c) Lake Wingra, and mean summertime (July 15-August 15) hypolimnetic temperatures for (d) Lake Mendota and (e) Fish Lake. In (a), (b), and (c), solid red lines represent statistically significant ($p < 0.5$) locations of abrupt changes in epilimnion temperatures and solid lines represent mean temperatures for each period. In (d) and (e) solid lines represent the long-term trend over the period 1911-2014. Hypolimnetic temperatures show no significant abrupt changes. Neither epilimnetic nor hypolimnetic temperatures for any lakes have significant changes in long-term trends.**





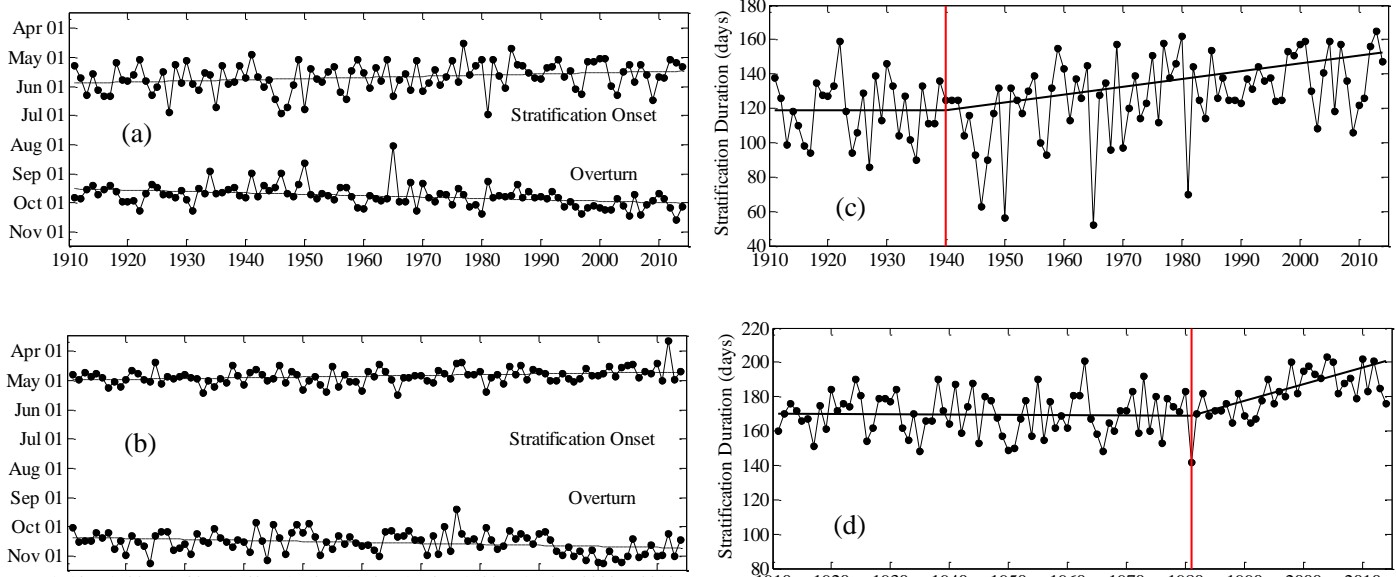

**Figure 5: Stratification onset and overturn dates for (a) Lake Mendota and (b) Fish Lake. Stratification duration for (c) Lake Mendota and (d) Fish Lake. Dark circles are modeled results and dashed lines denote the trendline for the 104-year period. In (a) and (b) solid lines represent the long-term trend in stratification onset and overturn dates. In (c) and (d), solid red lines represent the timing of a statistically significant (*p*<0.01) change in trend and solid black lines represent the trend during the periods.**





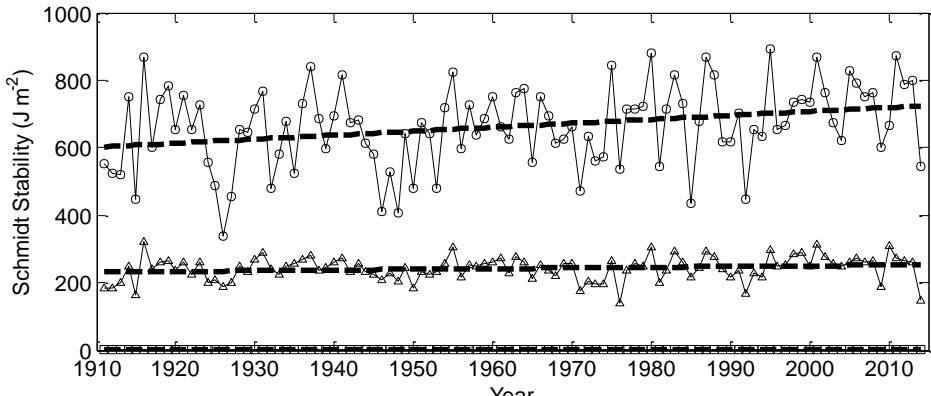

**Figure 6: Yearly average summer-time (15 July - 15 August) Schmidt stability values for Lake Mendota (circle), Fish Lake (triangle), and Lake Wingra (square).**





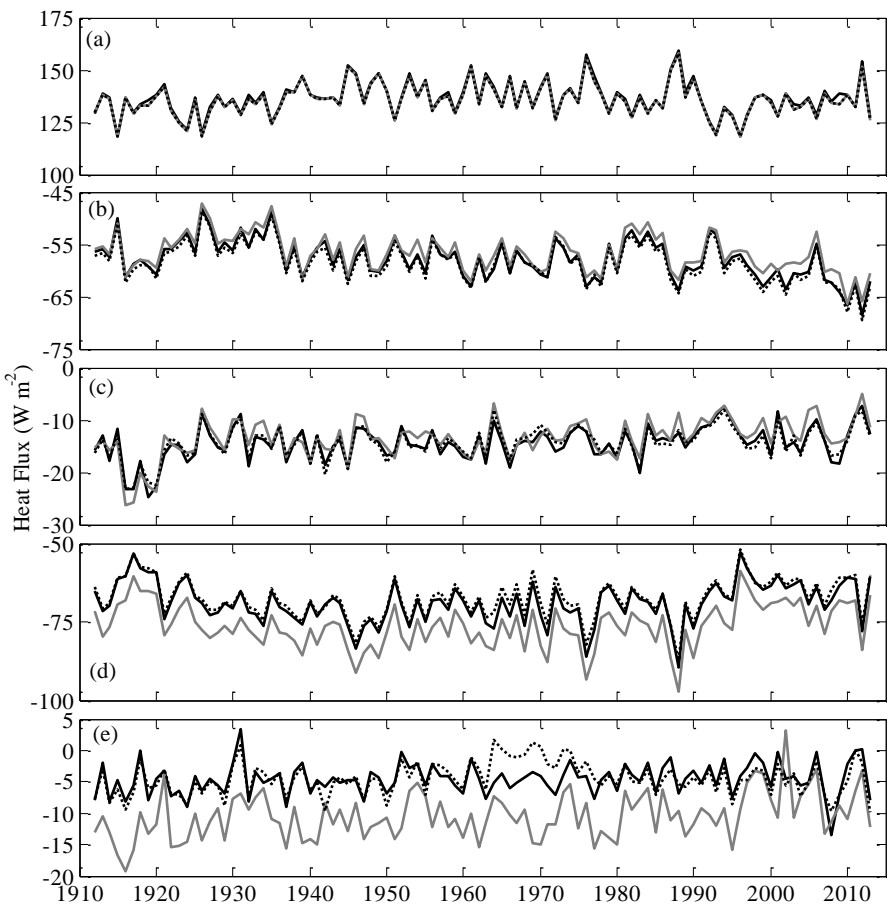

Figure 7: Yearly average (a) solar radiation flux, (b) long wave radiative flux, (c) sensible heat flux, (d) latent heat flux, and (e) total heat flux at the lake surface for Lake Mendota (solid black line), Fish Lake (black dashed line), and Lake Wingra (solid grey line). Trends and abrupt changes for heat fluxes are not shown on the plots for clarity.




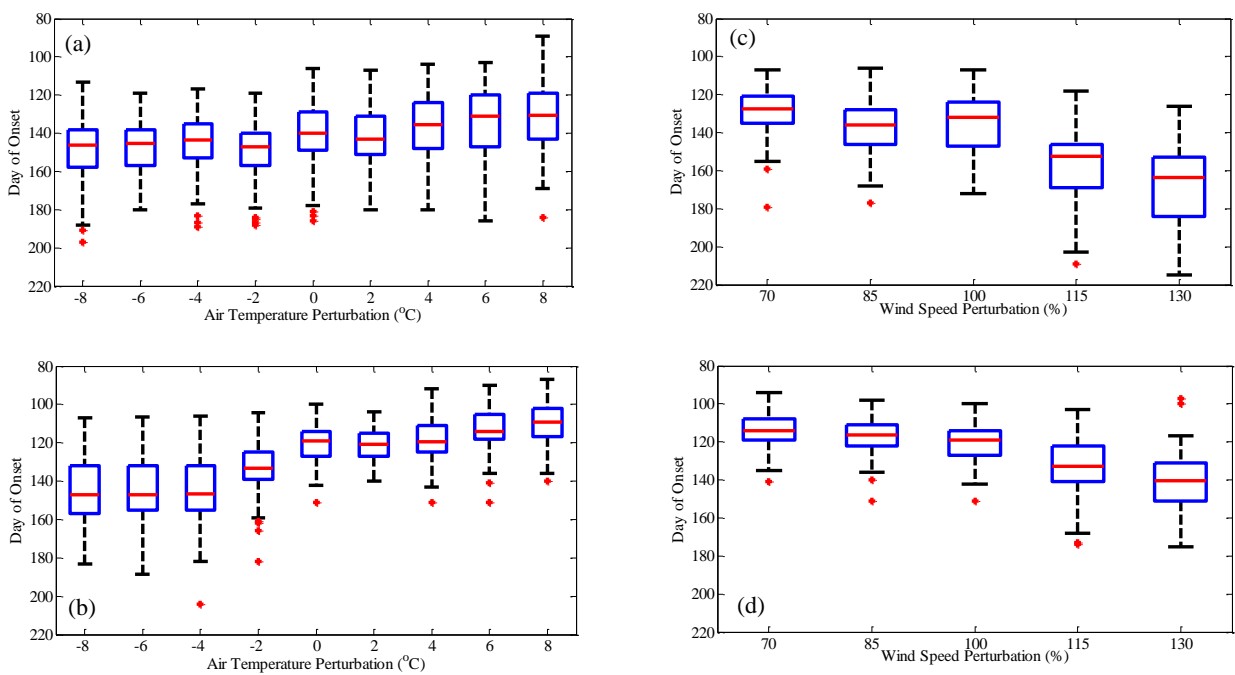

**Figure 8:** Day of stratification onset under select air temperature perturbation scenarios for **(a)** Lake Mendota and **(b)** Fish Lake and day of stratification onset under select wind speed perturbation scenarios for **(c)** Lake Mendota and **(d)** Fish Lake. The box represents the 25th and 75th quartiles and the central line is the median value. The whiskers extend to the minimum and maximum data point in cases where there are no outliers, which are plotted individually.




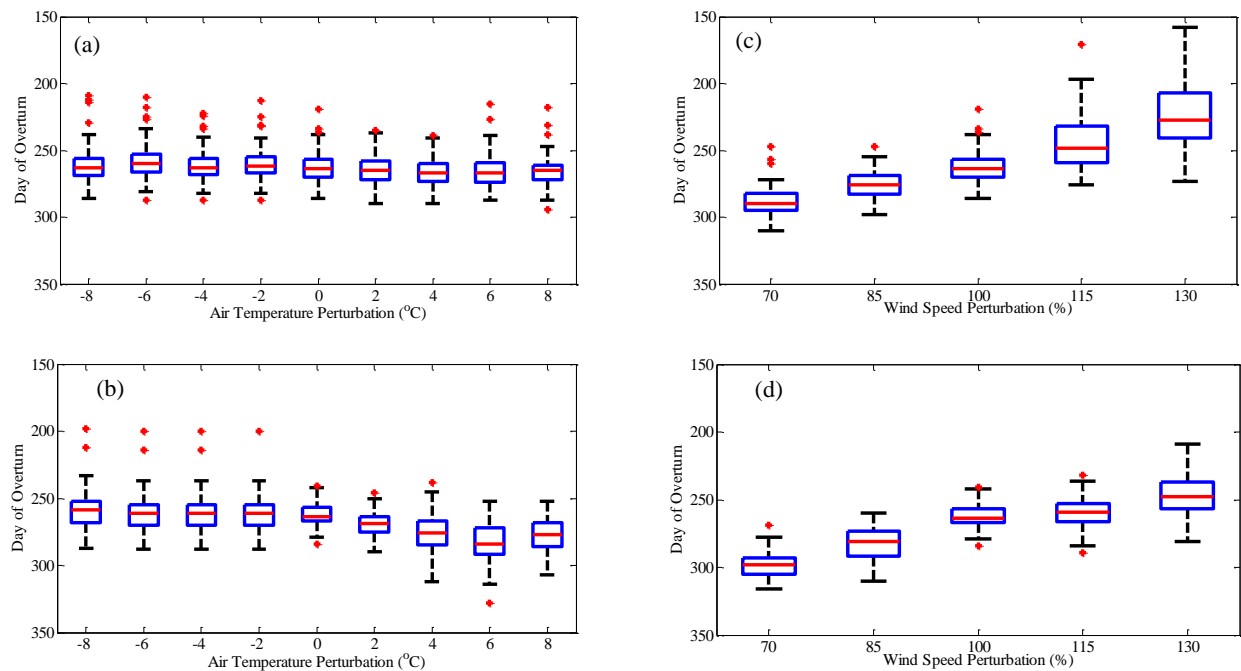

**Figure 9: Day of stratification overturn under select air temperature perturbation scenarios for (a) Lake Mendota and (b) Fish Lake and day of stratification overturn under select wind speed perturbation scenarios for (c) Lake Mendota and (d) Fish Lake. The box represents the 25th and 75th quartiles and the central line is the median value. The whiskers extend to the minimum and maximum data point in cases where there are no outliers, which are plotted individually.**




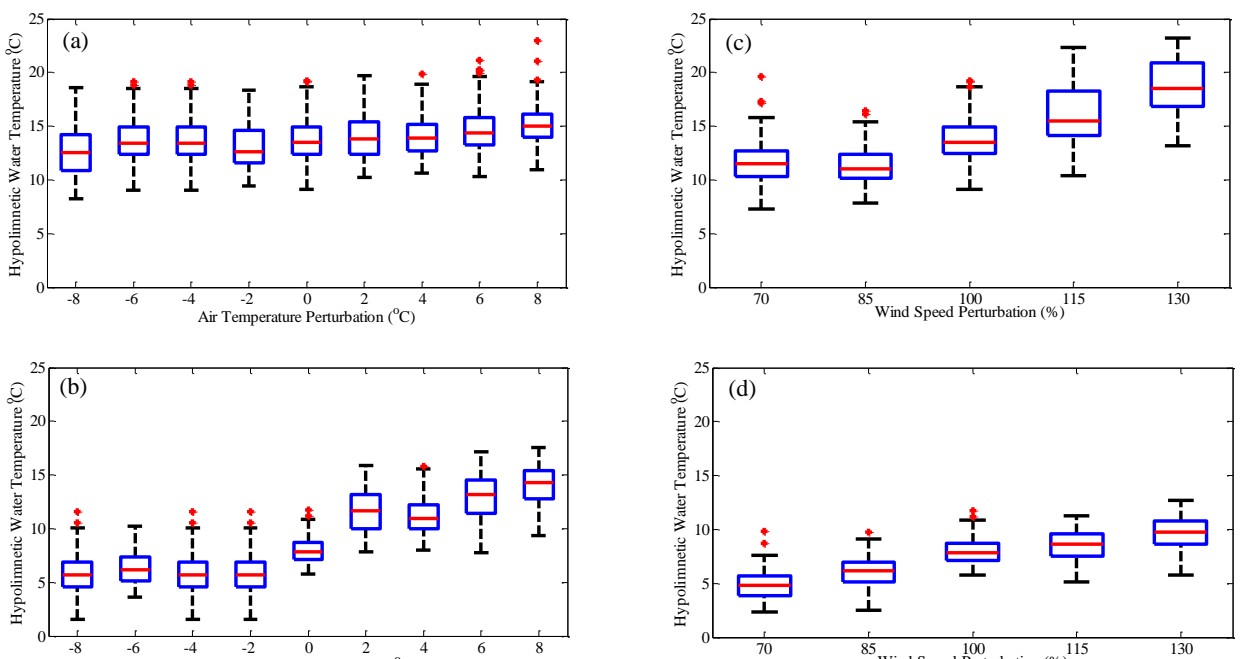

**Figure 10: Hypolimnetic water temperatures under select air temperature perturbation scenarios for (a) Lake Mendota and (b) Fish Lake and hypolimnetic water temperatures under select wind speed perturbation scenarios for (c) Lake Mendota and (d) Fish Lake. The box represents the 25th and 75th quartiles and the central line is the median value. The whiskers extend to the minimum and maximum data point in cases where there are no outliers, which are plotted individually.**