# Peer review of "Response of water temperatures and stratification to changing climate in three lakes with different morphometry"

_Hydrology and Earth System Sciences, 2016_

## Referee Comment (RC1) · Anonymous Referee #1 · 28 Aug 2016

This manuscript by Magee & Wu is based on an extraordinary data set of 104 years and
focuses on effects of changing air temperature and wind speed on water temperature
and stratification patterns of lakes with differing morphometry. The lakes are situated
close to each other which is a great asset in this kind of research. The long data sets
on basic variables and drivers is a good argument for publication and the results based
on these data are fairly convincing. They are also logical and actually so logical that
they very often leave a feeling that 'I already know this'. This may at least partly be due
to simplification of morphometry to lake depth and surface area, but also due to lack
of deep discussion; big part of 'Discussion' actually belong to 'Results' and to certain
extent to 'Methods'. Thus restructuring and extending the real discussion (starting from

4.4.), the paper would certainly improve. Some parts of the paper are also technically challenging for the reader since they are based on listing the numerical results one by one; a good example is section 3.5. The authors should also think about leaving Lake Wingra out completely; I suggest this because this paper has strong focus on lake stratification and Lake Wingra is a polymictic lake. The problem with Lake Wingra becomes obvious in Tables 2, 3 and 4 – lots of N/A markings. I also find it strange that in a paper where models are such an elemental part, they are not properly described; besides the equation for light extinction (eq 1), the authors only use references to published articles. More emphasis should also be given to description of gap filling and calibration data; both are now somewhat superficial. Besides these more general comments, I list here some more detailed ones: 1. I found it a little bit strange that sediment heat fluxes were hardly mentioned in this paper. Although there may have been no data on this or these fluxes were not included in models, they should have been tackled somehow at least in 'Discussion'. 2. The readers would benefit from some more information about the lakes. Especially information on lake clarity (water colour etc; cf. Table 1) would have been useful in a paper with such a strong focus on lake stratification. 3. As a reader I would also appreciate information on fetch for each lake; now the word 'fetch' and importance of fetch is mentioned several times, but the reader is left with the bathymetric maps to figure out the fetch 4. It is said that water level in Fish Lake has raised considerably and this has probably affected some of the results. However, nothing is said about the possible reason behind this phenomenon. Related to climate, human activity or what? 5. The authors state that Fish Lake does not always turn over completely in spring. This is an important piece of information, since in small, dark coloured boreal lakes this is a fairly common observation and it is believed that it is weather/climate driven change. It would be nice if the authors could dig deeper in this observation, especially since they have such a long time series. 6. Fish Lake and Lake Wingra have Secchi-depth results only from 1995 onwards. This appears problematic; could your give more explanation on this. 7. Data on below-ice Secchi-depth were used which I to certain extent understand, but since it is not that

common practice to measure Secchi under the ice, it would be useful to have some more information. 8. Figure 3 shows that in general simulations resulted in slightly lower temperatures in comparison to observations. Did you make this clear also in text? 9. The possible importance of internal waves is mentioned only on general level and not properly discussed in relation to the study lakes 10. Using wording 'increasing (decreasing)' is clumsy for the reader 11. Throughout the text there is repetition, e.g. in 'Results' sentences which belong to 'Material' and are already tackled there. Check the whole manuscript for that 12. Table 1: The meaning of the row 'Groundwater' is not clear to me 13. In Figure 4, the legend contains some description of results 14. In Figure 6, results on Lake Wingra should be left out (= zero line). And in general, the stability index is somehow funny in this context since the lake was known to be polymictic 15. The real discussion starts in 4.4. and all before that should be merged with 'Results'. An indication of that is the fact that for instance in 4.3.1 and 4.3.2 there are no references in the text. 16. There are some spelling mistakes in the text, please check.

---

## Referee Comment (RC2) · Anonymous Referee #2 · 1 Sep 2016

The authors use an extensive dataset on water temperatures from three neighboring lakes to test and validate a one-dimensional lake temperature model. The model is subsequently used for reconstruction of the thermal and stratification regime of the lakes during the last century and for sensitivity studies exploring the lake response to changes in mean annuals of air temperature and wind speed. The idea behind the sensitivity experiments is to elucidate the dissimilarity in the response of lakes with different depths and surface areas subject to identical external atmospheric forcing. The problem statement is clear. The methods are generaly relevant to the questions stated in the study (except the application of a 1d time-depth model to investigation of the effects of horizontal extensions on lake thermics, which requires additional justification,

see below). My major concern is the analysis of the results, which looks superficial, and representation of the outcomes, which is lengthy and poorly structured. The analysis is confined to descriptive presentation of model outcomes without an insight into the physical mechanisms producing the observed effects. Verbal presentation of trends in lake thermal characteristics covering several paragraphs is exhausting and not really informative. The manuscript presents a nice set of data and numerical results, which can serve as a basis for a well-thought study, but has liitle value for the reader in its present form. The manuscript requires a more detailed description of the model and discussion on its uncertainties and relevance to the real lake processes; the discussion should be rethought, moving the accent from the descriptive listing of the model responses to varying inputs to the discussion on the physical mechanisms producing thes responses.

Here are some major critical points:

- Effects of lake surface area on the response to the atmospheric forcing are continiously mentioned throughout the manuscript and are among the main subjects of the model sensitivy runs. However, the entire discussion is based on the ouptuts of a one-dimensional model, i.e. none of the physical processes depending on the horizontal dimensions are modeled directly, but *parameterized* in the model. Hence, the response of the model outcomes to varying surface area does not necessarily coincide with the response of real lakes to the same perturbations. To analyze properly the modeling results the authors need to (i) present the deteils on the model parameterizations related to the effects of horizontal advection, wind fetch, horizontally varying depth, and other horizontal processes, such as mixing by internal waves and upwelling of hypolimnetic waters in near-shore areas of the lake; (ii) when discussing the modeling results state clearly which of them can be extrapolated on the real lakes, which horizontal processes are missed by the model, and how it can affect the real situations; (iii) differentiate between the effects produced by incerase of the wind energy input due to larger aurface area from those produced by increase of the thermal inertia due to larger lake volume,

like, in particular, timing of the stratification onset (Section 4.3.1).

- Do the lakes have ice cover in winter? The ice model is repeatedly mentioned in the ms, but no results on the ice regime are presented/discussed. Duration of the ice-ciovered period directly affects timing of the summer stratification onset and summer hypolimnetic temperatures. Any discussion on these variables is incomplete without considering the ice regime.

- Section 4.3 Sensitivity runs can be shortened, at least, to a half and moved from 'Discussion' to 'Results'. The actual discussion should be added, considering the reasons for the observed dependencies, their relevance to the processes in real lakes and novelty of the results compared to the state-of-the-art in this area of research.

Minor comments:

P3L16 What is 'thermocline shifts'? Please, explain

P6L29 Provide model parameters and simulation specifications here.

P9L7 Add 'summer epilimnetic' to 'temperatures'

P10L13 and other appearances: replace '$0.067$ days earlier decade$^{-1}$' to '$+0.067$ days decade$^{-1}$'

P10L28 onwards: 'J m$^{-2}$' are not correct units for heat flux. Provide flux values in understandable units.

P11L17 How lake morphometry can affect the shortwave flux of solar radiation??

P14L12 and at other places: Schmidt stability is irrelevant to non-stratified lakes and cannot be used for comparison.

P17L9 See above

P17L18 Evaporation depends on surface temperatures, not the deep water temperatures. Explain what do you mean in this sentence, or remove it and find another

explanation for the phenomenon.

P17L2529 Actually, the main driver for epilimnetic temperatures is solar radiation not air temperature. If air temperature is the 'main driver', what do you mean under 'wind. . . a more dominant mechanism'?

P18L14-15 Explain, why stronger winds should produce higher spatial variability in wind stress. How did you estimate changes in turbulence and why do you think they are nonlinear? Table 2, Fig. 3: The model seems to produce consistently a positive bias in lake temperatures. Any explanation for this?

Typos:

P4L12 Capitalize 'Secchi'

P5L29 remove second appearance of 'Lake Mendota'

P8L15 replace 'decreased' with 'decrease'

P12L13 replace 'difficulty' with 'difficult'

---

## Author Comment (AC1) · 19 Oct 2016

**Response to Reviewer 1**

- This manuscript by Magee & Wu is based on an extraordinary data set of 104 years and focuses on effects of changing air temperature and wind speed on water temperature and stratification patterns of lakes with differing morphometry. The lakes are situated close to each other which is a great asset in this kind of research. The long data sets on basic variables and drivers is a good argument for publication and the results based on these data are fairly convincing. They are also logical and actually so logical that they very often leave a feeling that 'I already know this'. This may at least partly be due to simplification of morphometry to lake depth and surface area, but also due to lack of deep discussion; big part of 'Discussion' actually belong to 'Results' and to certain extent to 'Methods'. Thus restructuring and extending the real discussion (starting from 4.4.), the paper would certainly improve.

  The authors thank Reviewer 1 for taking the time to review and providing helpful comments to improve the manuscript. Following the suggestion, the authors are restructuring the paper and extending the discussion section. We have addressed additional comments in a point-by-point reply and carefully address the issues raised in the revised manuscript.

- Some parts of the paper are also technically challenging for the reader since they are based on listing the numerical results one by one; a good example is section 3.5.

  The authors thank the reviewer for this comment. We will address the structure of the writing that become technically challenging for readers. We have reduced the presentation of trends and related thermal characteristics within the text. We avoid listing the numerical results one by one without any meanings and incorporate these numerical results into the figures and tables.

- The authors should also think about leaving Lake Wingra out completely; I suggest this because this paper has strong focus on lake stratification and Lake Wingra is a polymictic lake. The problem with Lake Wingra becomes obvious in Tables 2, 3 and 4 – lots of N/A markings.

  The authors thank the reviewer for this comment. Lake Wingra does stratify on daily or weekly timescales during the summer months (Kimura et al, 2016). Summer Schmidt stability was calculated at daily timescales, and then averaged for each year before comparing coherence among the lake pairs. Higher average stability for one year on Lake Wingra would indicate that the lake experienced more days of stratification during the period. This phenomenon can be coherent with changes in stability for the other two lakes.

  Reference:

  Kimura, N., Wu, C.H., Hoopes, J.A., and Tai, A. 2016, Diurnal thermal dynamic processes in a small and shallow lake under non-uniform wind and weak stratification, Journal of Hydraulic Engineering-ASCE, 142(11), 04016047,

- I also find it strange that in a paper where models are such an elemental part, they are not properly described; besides the equation for light extinction (eq 1), the authors only use references to published articles.

  The authors thank the Reviewer 1 for this comment. Indeed, the description of the model have been described in great detail in the papers (Magee and Wu, 2016 in *Hydrological Processes* doi/10.1002/hyp.10996/full, Magee et al, 2016 in *Hydrology and Earth System Sciences*, DOI:10.5194, 20(5), 1681-1702. As a result, we did not intend to repeat the information and included the detailed description in the manuscript. Based upon the concerns raised by the reviewer, we will add proper description of the model as necessary. The equation for light extinction is included in the manuscript as it is a new updated component. We will provide additional details on the model subroutines that directly affect horizontal processes in the lake. We will also detail parameterizations and describe how they influence the results of this analysis.

- More emphasis should also be given to description of gap filling and calibration data; both are now somewhat superficial.

  More detailed description of gap filling and calibration data can be found in other manuscripts (Magee et al, 2016 in *Hydrology and Earth System Sciences*, Magee and Wu, *Hydrological Processes* doi/10.1002/hyp.10996/full in press). We agree that this information is pertinent to the discussion and results. The authors will provide more details of these analysis in a supplement to the paper, which would address the concern raised by the reviewer.

Besides these more general comments, I list here some more detailed ones:

1. I found it a little bit strange that sediment heat fluxes were hardly mentioned in this paper. Although there may have been no data on this or these fluxes were not included in models, they should have been tackled somehow at least in 'Discussion'.

   Sediment heat flux is included in the model. Details of this information can be found in other published papers (Magee et al, 2016, Magee and Wu, 2016). We agree that sediment heat fluxes are an important component to the overall heat budget of the lake. We will discuss sediment heat fluxes in the revised manuscript. Additionally, we will include more detail on the parameterization and inclusion of sediment heat flux in the revised manuscript.

2. The readers would benefit from some more information about the lakes. Especially information on lake clarity (water colour etc; cf. Table 1) would have been useful in a paper with such a strong focus on lake stratification.

The authors thank the reviewer for this comment. Additional information on the lakes has been added to Table 1, including fetch, shoreline development, landscape position, Secchi depth, surface water chlorophyll concentration, and DOC in each lake. Specific values of lake water color is not collected by the NTL-LTER program as other data were.

3. As a reader I would also appreciate information on fetch for each lake; now the word 'fetch' and importance of fetch is mentioned several times, but the reader is left with the bathymetric maps to figure out the fetch

The authors apologize for neglecting to include this value explicitly in the manuscript. Information on lake fetch for each lake has been added to Table 1.

4. It is said that water level in Fish Lake has raised considerably and this has probably affected some of the results. However, nothing is said about the possible reason behind this phenomenon. Related to climate, human activity or what?

The text has been changed to read "the water level of the lake rose by 2.75 meters due to an increase in regional groundwater recharge causing increased groundwater flow to the lake (Krohelski et al., 2002). Krohelski et al. (2002) hypothesized that the increase in regional groundwater recharge may be the result of increased infiltration from snowmelt after increased snowfall and less frost-covered soil."

5. The authors state that Fish Lake does not always turn over completely in spring. This is an important piece of information, since in small, dark coloured boreal lakes this is a fairly common observation and it is believed that it is weather/climate driven change. It would be nice if the authors could dig deeper in this observation, especially since they have such a long time series.

The lake does mix each spring, however, low water temperatures of only ~5°C in the hypolimnion may be due to shortened spring mixing durations compared to years where the hypolimnion temperature reaches ~8-11°C. The authors hypothesize that this phenomenon may be related to late ice-out in some years. We will investigate the phenomenon further in the revised version of the manuscript. Ice cover does impact the timing of stratification and hypolimnion temperatures. We propose to add a section in the discussion that describes this interaction and its influence on the results presented here while not reproducing content already presented in Magee and Wu (2016). This section would address our hypothesis that colder hypolimnion temperature is related to years with later ice-off.

6. Fish Lake and Lake Wingra have Secchi-depth results only from 1995 onwards. This appears problematic; could you give more explanation on this.

Fish Lake and Lake Wingra became part of the NTL-LTER program in 1995; and regular Secchi depth measurements were taken starting then. The authors agree with Reviewer 1 that it is not ideal to use seasonal averages for the historical period before 1995; however, given the strongly seasonal dynamic of water clarity and light extinction in the lakes, using seasonal averages of Secchi depth to estimate light extinction are preferably to a constant light extinction for the lakes, which is not representative of observed phenomenon in the lakes. The authors will perform additional model analysis to quantify the uncertainty that may be caused by using seasonally-average Secchi depth instead of the measured values.

7. Data on below-ice Secchi-depth were used which I to certain extent understand, but since it is not that common practice to measure Secchi under the ice, it would be useful to have some more information.

Light extinction, which can be estimated from Secchi depth, greatly influences water temperatures and overall temperature profile. Including light extinction in winter more reliably reproduces under-ice water temperatures and as a result, water temperatures at the time of ice-off. Temperature profiles at ice-off impact the timing of stratification and the hypolimnetic water temperature through the summer. Properly characterising and capturing these phenomena in the model will enable accurate reproduction of water temperatures during the historical period. For this study, the authors choose to utilize the available data from previous ecological and water quality studies conducted on the lakes to better inform the model and more accurately reproduce water temperature profiles.

8. Figure 3 shows that in general simulations resulted in slightly lower temperatures in comparison to observations. Did you make this clear also in text?

We will improve the text to make this clearer and expand on why this bias exists.

9. The possible importance of internal waves is mentioned only on general level and not properly discussed in relation to the study lakes

Internal waves are parameterized in the model, however, the reviewer is correct that we did not explicitly explain how the model deals with internal waves nor how internal waves affect hydrodynamics in each of the lakes. Based on other research, the effect of internal waves may be large in Lake Mendota and much smaller in Fish Lake. We will clearly expand upon this issue and discuss the validity of modeling results in relation to the parameterization of internal waves in the model.

10. Using wording 'increasing (decreasing)' is clumsy for the reader

The authors thank the reviewer for this comment. We have revised the text to make the writing clearer for readers.

11. Throughout the text there is repetition, e.g. in 'Results' sentences which belong to 'Material' and are already tackled there. Check the whole manuscript for that

The authors have removed and moved sections of the manuscript to address this issue.

12. Table 1: The meaning of the row 'Groundwater' is not clear to me

The row 'groundwater' describes the groundwater inflow type of the lakes. For example, "discharge" lakes are those which have a net groundwater discharge into the lake. "Flowthrough" lakes are those which have small net inflow or outflow. As for Lake Wingra have high groundwater inflows and outflows which result in small net discharge into the lake.

13. In Figure 4, the legend contains some description of results

The authors thank the reviewer for pointing out this comment. We have removed the description of the results out of the legend.

14. In Figure 6, results on Lake Wingra should be left out (= zero line). And in general, the stability index is somehow funny in this context since the lake was known to be polymictic

As suggested by Reviewer 1, we will leave Lake Wingra out of this figure. To address stability, please see our previous response concerning the inclusion of Lake Wingra.

15. The real discussion starts in 4.4. and all before that should be merged with 'Results'. An indication of that is the fact that for instance in 4.3.1 and 4.3.2 there are no references in the text.

The authors thank the Reviewer 1 for the comment and suggestion. We will restructure the discussion by moving Sections 4.2 and 4.3 to the Results section and emphasize the results in terms of physical mechanisms that are influencing the simulated and observed responses.

16. There are some spelling mistakes in the text, please check.

The typos have been fixed within the manuscript. Thank you to the reviewer for pointing out these mistakes to the authors. We will additionally review the manuscript carefully for typographical errors before resubmitting a revised manuscript.

---

## Author Comment (AC2) · 19 Oct 2016

**Response to Reviewer 2**

- The authors use an extensive dataset on water temperatures from three neighboring lakes to test and validate a one-dimensional lake temperature model. The model is subsequently used for reconstruction of the thermal and stratification regime of the lakes during the last century and for sensitivity studies exploring the lake response to changes in mean annuals of air temperature and wind speed. The idea behind the sensitivity experiments is to elucidate the dissimilarity in the response of lakes with different depths and surface areas subject to identical external atmospheric forcing. The problem statement is clear. The methods are generally relevant to the questions stated in the study (except the application of a 1d time-depth model to investigation of the effects of horizontal extensions on lake thermics, which requires additional justification, see below).

  The authors thank Reviewer 2 for taking the time to review and provide detailed and insightful comments on the manuscript. These point-by-point comments or questions will be carefully addressed in a revised version of the manuscript.

- My major concern is the analysis of the results, which looks superficial, and representation of the outcomes, which is lengthy and poorly structured. The analysis is confined to descriptive presentation of model outcomes without an insight into the physical mechanisms producing the observed effects.

  We appreciate the comment concerning the analysis of results and structure of the paper. We will restructure the manuscript and revise analysis to address the points raised and provide insight of physical mechanisms producing the observed effects.

- Verbal presentation of trends in lake thermal characteristics covering several paragraphs is exhausting and not really informative.

  The authors thank the reviewer for this comment. We have reduced the presentation of trends and related thermal characteristics within the text by removing those that are extraneous to addressing the problem statement. These information have been incorporated into the figures and tables.

- The manuscript presents a nice set of data and numerical results, which can serve as a basis for a well-thought study, but has little value for the reader in its present form.

  The authors thank the reviewer for the positive comments. We restructure the presentation and put careful thoughts into the manuscript to improve the value of the paper for readers.

- The manuscript requires a more detailed description of the model and discussion on its uncertainties and relevance to the real lake processes; the discussion should be rethought, moving the accent from the descriptive listing of the model responses to varying inputs to the discussion on the physical mechanisms producing the responses.

  The authors thank the Reviewer 2 for this suggestion. We restructure the discussion by moving Sections 4.2 and 4.3 to the Results section; emphasize the results in terms of physical mechanisms that are influencing the simulated and observed responses; and discuss the results in context of ecological and chemical processes within the lakes. Detailed description of the model will be provided without repeating in the papers (Magee and Wu, 2016 in *Hydrological Processes* doi/10.1002/hyp.10996/full, Magee et al, 2016 in *Hydrology and Earth System Sciences*, DOI:10.5194, 20(5), 1681-1702. In this manuscript, we will improve description of physical processes and address parameterization of horizontal processes. Furthermore, we will improve discussion on how the model parameterizations affect the results of our study and how they differ from real lake processes.

Here are some major critical points:

- Effects of lake surface area on the response to the atmospheric forcing are continuously mentioned throughout the manuscript and are among the main subjects of the model sensitivity runs. However, the entire discussion is based on the outputs of a one-dimensional model, i.e. none of the physical processes depending on the horizontal dimensions are modeled directly, but parameterized in the model. Hence, the response of the model outcomes to varying surface area does not necessarily coincide with the response of real lakes to the same perturbations. To analyze properly the modeling results the authors need to (i) present the details on the model parameterizations related to the effects of horizontal advection, wind fetch, horizontally varying depth, and other horizontal processes, such as mixing by internal waves and upwelling of hypolimnetic waters in near-shore areas of the lake; (ii) when discussing the modeling results state clearly which of them can be extrapolated on the real lakes, which horizontal processes are missed by the model, and how it can affect the real situations; (iii) differentiate between the effects produced by increase of the wind energy input due to larger surface area from those produced by increase of the thermal inertia due to larger lake volume, like, in particular, timing of the stratification onset (Section 4.3.1).

  The authors thank the reviewer for this comment. We will present additional details of the model, especially concerning horizontal parameterizations in the text to detail how the model may differ from real lakes and how that difference impacts results concerning the response of lakes to perturbations. As suggested by the reviewer, we will clearly state which horizontal processes are missed by the model and which can clearly be extrapolated onto three-dimensional lakes. Additionally, we will discuss the effects of increased wind energy input due to larger surface area

compared to changes caused by increase of thermal inertia as suggested by the reviewer here. This is a particularly important point to investigate when attempting to extrapolate results presented here to a larger variety of lakes.

- Do the lakes have ice cover in winter? The ice model is repeatedly mentioned in the MS, but no results on the ice regime are presented/discussed. Duration of the ice-covered period directly affects timing of the summer stratification onset and summer hypolimnetic temperatures. Any discussion on these variables is incomplete without considering the ice regime.

  These lakes do have ice cover in winter. A paper, Magee and Wu (2016) is in press and details both the ice model and results of changes to the ice regime in the three lakes. As a result, we do not include the same ice cover changes in this manuscript. We agree with the reviewer that timing of ice-off does influence the timing of stratification onset and the duration of mixing in the spring. We will summarize the results on the ice regime in Magee and Wu (2016). We also will add a section in the discussion that describes this interaction and its influence on the results presented here while not reproducing content already presented in Magee and Wu (2016).

  Reference:

  Magee, MR and Wu, CH (2016) Effects of changing climate on ice cover in three morphometrically different lakes. *Hydrological Processes*. DOI: 10.1002/hyp.10996

- Section 4.3 Sensitivity runs can be shortened, at least, to a half and moved from 'Discussion' to 'Results'. The actual discussion should be added, considering the reasons for the observed dependencies, their relevance to the processes in real lakes and novelty of the results compared to the state-of-the-art in this area of research.

  The authors agree with this suggestion from the reviewer. We have moved Section 4.3 to the Results section, shortened the presentation of the results. We will add discussion of the reasons for the observed relationships and their relevance to ecological and chemical processes within the lakes.

  Minor comments:

- P3L16 What is 'thermocline shifts'? Please, explain.

  The authors thank Reviewer 2 for pointing out confusion due to our choice of word. 'Thermocline shifts' refers to changes in thermocline depth in response to a driver such as changes in climate. We have changed the line to read "changes in thermocline depth from warming air temperatures may be dampened…" to remove some of this confusion due to previous word choice.

- P6L29 Provide model parameters and simulation specifications here.

  We will provide addition parameters and simulation specifications as suggested within the text.

- P9L7 Add 'summer epilimnetic' to 'temperatures'

  Done, as requested by the reviewer.

- P10L13 and other appearances: replace '0.067 days earlier decade$^{-1}$ ' to '+0.067 days decade$^{-1}$ '

  Following the suggestion by the reviewer, the authors have made this change to improve readability of the manuscript.

- P10L28 onwards: 'J m$^{-2}$ ' are not correct units for heat flux. Provide flux values in understandable units.

  The authors thank the Reviewer 2 for pointing out this error in units. Indeed, the units should be W m$^{-2}$, and the error occurred by inadvertently carrying over units from the previous sections of text). The units are correct in the corresponding figure. We have addressed the incorrect units in the text.

- P11L17 How lake morphometry can affect the shortwave flux of solar radiation??

  The shortwave flux is the net flux at the surface of each lake. The shortwave flux is controlled in part by albedo of the surface water, by snow ice cover in the lake. Each lake may have slightly different net shortwave radiative flux for each day and average for the year.

- P14L12 and at other places: Schmidt stability is irrelevant to non-stratified lakes and cannot be used for comparison.

  P17L9 See above

  Lake Wingra does stratify on daily or weekly timescales during the summer months (Kimura et al, 2016). Summer Schmidt stability was calculated at daily timescales, and then averaged for each year before comparing coherence among the lake pairs. Higher average stability for one year on Lake Wingra would indicate that the lake experienced more days of stratification during the period. This phenomenon can be coherent with changes in stability for the other two lakes.

  Reference:

  Kimura, N., Wu, C.H., Hoopes, J.A., and Tai, A. 2016, Diurnal thermal dynamic processes in a small and shallow lake under non-uniform wind and weak stratification, Journal of Hydraulic Engineering-ASCE, 142(11), 04016047,

- P17L18 Evaporation depends on surface temperatures, not the deep water temperatures. Explain what do you mean in this sentence, or remove it and find another explanation for the phenomenon.

  Evaporation does depend on surface temperature. Lakes with cooler bottom waters will have more heat transfer from the surface waters to lower levels in the water column compared to lakes that have warmer bottom waters. This heat flux in turn affects the surface temperature of the lake, and consequently affect the evaporation.

- P17L2529 Actually, the main driver for epilimnetic temperatures is solar radiation not air temperature. If air temperature is the 'main driver', what do you mean under 'wind. . . a more dominant mechanism'?

  The authors agree that solar radiation is the main driver for epilimnion temperatures and the driver is not included in this analysis. What we mean is that the main driver of between air temperature and wind speed is air temperature. Wind mixing is a more dominant mechanism to transfer heat from upper layers of the water column to bottom waters, in comparison with molecular diffusion of heat. While air temperature is a main driver to directly influence surface water temperatures, wind speed changes are the main mechanism for dissipating heat to the lower water levels. Both can act to change the response of epilimnion temperatures to air temperature changes. Based on the comments from the reviewer, this section is confusing and unclear for the readers. We will carefully revise the manuscript and also cite references. Overall, we will make it clear the main mechanisms and how they interact with one another..

- P18L14-15 Explain, why stronger winds should produce higher spatial variability in wind stress. How did you estimate changes in turbulence and why do you think they are nonlinear?

  The authors will provide the explanation and address the questions raised by the reviewer. First of all, we do not mean to imply that there is higher spatial variability in wind stress within the lakes themselves. Rather, increases and/or decreases in wind speed in general will result in nonlinear changes in wind stress and turbulence in all lakes. Specifically, wind stress varies with the square of wind speed. As a result, changes in wind speed directly result in non-linear changes in wind stress on the water surface. The DYRESM model parameterizes mixing within the model by estimating the turbulent kinetic energy (TKE) and mixing layers when a potential energy threshold is exceeded. TKE in the model is introduced through convective mixing, wind stirring, and shear mixing using parameterizations that are all non-linear equations and influenced non-linearly either directly or indirectly by wind speed. In other words, linear changes in wind speed yield non-linear changes in the turbulence estimation in the model. This explanation will be made in the revised manuscript both through restructuring and rewording that section and by providing details of model equations and parameterizations. We will make more clear to the readers how the nonlinear response occurs.

- Table 2, Fig. 3: The model seems to produce consistently a positive bias in lake temperatures. Any explanation for this?

  The model results under predict slightly water temperatures. This under prediction is from a combination of averaging meteorological inputs over the day and comparing temperatures output on a daily timestep with observations collected typically during the afternoon when water temperatures are slightly higher than daily averages.

  Typos:

- P4L12 Capitalize 'Secchi'
- P5L29 remove second appearance of 'Lake Mendota'
- P8L15 replace 'decreased' with 'decrease'
- P12L13 replace 'difficulty' with 'difficult'

  All the typos have been fixed within the manuscript. We extend our many thanks to the reviewer for pointing out these mistakes to the authors. Furthermore, we will review and re-review the manuscript carefully for typographical errors before resubmitting a revised manuscript.

---

## Author Response (AR1)

**Response to Reviewer 1**

• This manuscript by Magee & Wu is based on an extraordinary data set of 104 years and focuses on effects of changing air temperature and wind speed on water temperature and stratification patterns of lakes with differing morphometry. The lakes are situated close to each other which is a great asset in this kind of research. The long data sets on basic variables and drivers is a good argument for publication and the results based on these data are fairly convincing. They are also logical and actually so logical that they very often leave a feeling that 'I already know this'. This may at least partly be due to simplification of morphometry to lake depth and surface area, but also due to lack of deep discussion; big part of 'Discussion' actually belong to 'Results' and to certain extent to 'Methods'. Thus restructuring and extending the real discussion (starting from 4.4.), the paper would certainly improve.

The authors thank Reviewer 1 for taking the time to review and providing helpful comments to improve the manuscript. Following the suggestion, the authors are restructuring the paper and extending the discussion section. We have addressed additional comments in a point-by-point reply and carefully address the issues raised in the revised manuscript.

• Some parts of the paper are also technically challenging for the reader since they are based on listing the numerical results one by one; a good example is section 3.5.

The authors thank the reviewer for this comment. We have addressed the structure of the writing in Sec. 3.5 that is challenging for readers. Specifically, we avoid listing the numerical results oneby-one without meaningful interpretation. Instead, we present the results in Figure 7 and Table 4. We summarize the overall and trends and related thermal characteristics in the manuscript (see Page 11, Line 6-12).

• The authors should also think about leaving Lake Wingra out completely; I suggest this because this paper has strong focus on lake stratification and Lake Wingra is a polymictic lake. The problem with Lake Wingra becomes obvious in Tables 2, 3 and 4 – lots of N/A markings.

The authors thank the reviewer for this comment. Lake Wingra does stratify on daily or weekly timescales during the summer months (Kimura et al, 2016). Summer Schmidt stability was calculated at daily timescales, and then averaged for each year before comparing coherence among the lake pairs. Higher average stability for one year on Lake Wingra would indicate that the lake experienced more days of stratification during the period. This phenomenon can be coherent with changes in stability for the other two lakes.

**Reference:**

Kimura, N., Wu, C.H., Hoopes, J.A., and Tai, A. 2016, Diurnal thermal dynamic processes in a small and shallow lake under non-uniform wind and weak stratification, Journal of Hydraulic Engineering-ASCE, 142(11), 04016047.

• I also find it strange that in a paper where models are such an elemental part, they are not properly described; besides the equation for light extinction (eq 1), the authors only use references to published articles.

The authors thank the Reviewer 1 for this comment. Indeed, the description of the model have been described in great detail in the papers (Magee and Wu, 2016 in *Hydrological Processes* doi/10.1002/hyp.10996/full, Magee et al, 2016 in *Hydrology and Earth System Sciences*, DOI:10.5194, 20(5), 1681-1702). As a result, we did not intend to repeat the information in the manuscript and only refer to those published papers. Furthermore, we address the concerns raised by the reviewer. The equation for light extinction is included in the manuscript as it is a new updated component. We have edited and re-written portions of the manuscript to document additional details on the model subroutines that directly affect horizontal processes in the lake. In addition, we detail parameterizations and describe how they influence the results of this analysis. We summarize the change of the manuscript in the following:

• Page 4, Line 12-30: Section 2.2 Model description for hydrodynamics modeling

[revised manuscript text omitted]

• More emphasis should also be given to description of gap filling and calibration data; both are now somewhat superficial.

We have provided more description and re-written the text for additional clarity concerning gapfilling and calibration data. More detailed description of gap filling and calibration data can be found in other manuscripts (Magee et al, 2016; Magee and Wu, 2016).

Besides these more general comments, I list here some more detailed ones:

1. I found it a little bit strange that sediment heat fluxes were hardly mentioned in this paper. Although there may have been no data on this or these fluxes were not included in models, they should have been tackled somehow at least in 'Discussion'.

Sediment heat flux is included in the model. This description can be found in other published papers (Magee et al, 2016, Magee and Wu, 2016). We have added detail about sediment heat flux within the model section in the manuscript (Page 5 Line 1-12) as follows:

"Sediment heat flux is included as a source/sink term for each model layer. A diffusion relation from Rogers et al. (1995) is used to estimate  $q_{sed}$ , heat transfer from the sediments to the water column.

$$q_{sed} = \mathcal{K}_{sed} \frac{dT}{dz} \tag{1}$$

where Ksed represents the sediment conductivity with a value of 1.2  $Wm^{-1}$  °C-1, and dT/dz is estimated as:

$$\frac{dT}{dz} = \frac{T_s - T_w}{z_{sed}} \tag{2}$$

where dT/dz is the temperature gradient across the sediment-water interface,  $T_w$  is the water temperature adjacent to the sediment boundary,  $z_{sed}$  is the distance beneath the water-sediment interface at which the sediment temperature becomes relatively invariant, and is taken to be 5 m (Birge et al., 1927).  $T_s$  derived from Birge et al. (1927) and seasonally variant as follows:

$$T_s = 9.7 + 2.7 \sin\left[\frac{2\pi(D - 151)}{TD}\right]$$
(3)

where D is the number of days from the start of the year and TD is the total number of days within a year."

2. The readers would benefit from some more information about the lakes. Especially information on lake clarity (water colour etc; cf. Table 1) would have been useful in a paper with such a strong focus on lake stratification.

The authors thank the reviewer for this comment. Additional information on the lakes has been added to Table 1, including fetch, shoreline development, landscape position, Secchi depth, surface water chlorophyll concentration, and DOC in each lake. Specific values of lake water color is not collected by the NTL-LTER program as other data were.

3. As a reader I would also appreciate information on fetch for each lake; now the word 'fetch' and importance of fetch is mentioned several times, but the reader is left with the bathymetric maps to figure out the fetch

The authors apologize for neglecting to include this value explicitly in the manuscript. Information on lake fetch for each lake has been added to Table 1.

4. It is said that water level in Fish Lake has raised considerably and this has probably affected some of the results. However, nothing is said about the possible reason behind this phenomenon. Related to climate, human activity or what?

Thank you to Reviewer 2 for pointing out this statement which was not properly described. The text (Page 4 Line 2-4) has been changed to read

"the water level of the lake rose by 2.75 meters due to an increase in regional groundwater recharge causing increased groundwater flow to the lake (Krohelski et al., 2002). The increase in regional groundwater recharge may be the result of increased infiltration from snowmelt after increased snowfall and less frost-covered soil."

5. The authors state that Fish Lake does not always turn over completely in spring. This is an important piece of information, since in small, dark coloured boreal lakes this is a fairly common observation and it is believed that it is weather/climate driven change. It would be nice if the authors could dig deeper in this observation, especially since they have such a long time series.

After analyzing data, Fish lake does always mix each spring. However, low water temperatures of only 5~6°C in the hypolimnion, indicating that long mixing periods do not occur and little heat is added into the hypolimnion during spring mixing. During the historical period, the phenomenon occurred 4 times, and it may be related to lower spring wind speeds. Perturbation analysis suggests that this phenomenon could occur from low wind speeds and low air temperatures (see Page 12 Line 27-Page 13 Line 5).

6. Fish Lake and Lake Wingra have Secchi-depth results only from 1995 onwards. This appears problematic; could you give more explanation on this.

Fish Lake and Lake Wingra became part of the NTL-LTER program in 1995 and regular Secchi depth measurements were taken starting then. The authors agree with Reviewer 1 that it is not perfect to use seasonal averages for the historical period before 1995. However, given the strongly seasonal dynamic of water clarity and light extinction in the lakes, using seasonal averages of Secchi depth to estimate light extinction are preferably to a constant light extinction for the lakes, which is not representative of observed phenomenon in the lakes. To address this concern, we have added comments on the discussion (P14, L19-26): "Light extinction significantly impacts thermal stratification (Hocking and Straškraba, 1999) and light extinction estimated from Secchi depths can have a large degree of measurement uncertainty (Smith and Hoover, 2000, , Magee et al, 2016), which may result in uncertainty in water temperatures. To address this uncertainty, where available, we use measured Secchi depth values, which has been shown to improve estimates of the euphotic zone over fixed coefficients (Luhtala and Tolvanen, 2013). Secchi depths were unavailable for portions of the simulation period, and average values for the season were used. Analysis comparing using the method of known Secchi depths to both seasonally-varying average Secchi depths and constant Secchi depths for the lakes indicates that seasonally-varying averages do not significantly decrease model reliability when compared to year-specific values, but do show improvement over constant Secchi depths."

7. Data on below-ice Secchi-depth were used which I to certain extent understand, but since it is not that common practice to measure Secchi under the ice, it would be useful to have some more information.

Light extinction, which can be estimated from Secchi depth, greatly influences water temperatures and overall temperature profile. Including light extinction in winter more reliably reproduces under-ice water temperatures and as a result, water temperatures at the time of ice-off. Temperature profiles at ice-off impact the timing of stratification and the hypolimnetic water temperature through the summer. Properly characterising and capturing these phenomena in the model enables accurate reproduction of water temperatures during the historical period. For this study, the authors choose to utilize the available data from previous ecological and water quality studies conducted on the lakes to better inform the model for more reliably reproducing water temperature profiles.

8. Figure 3 shows that in general simulations resulted in slightly lower temperatures in comparison to observations. Did you make this clear also in text?

The authors thank the reviewer for pointing out this. We have made this clear in the text of the manuscript (see Page 13, Line 9-13): "Generally, simulated temperatures were lower than observed values. Some may be attributed to timing of observations, which in most instances occur during midday, when water temperatures may be slightly higher than daily averages, as output

from the model. Slight deviation is also expected due to averaging of air temperature and wind speeds. In general, thermocline depths were within 1 m of observed values, but some years differ by as much as 2.5 m, contributing additional error in water temperature comparison for depths near the thermocline."

9. The possible importance of internal waves is mentioned only on general level and not properly discussed in relation to the study lakes

The reviewer is correct that we did not explicitly explain how the model deals with internal waves nor how internal waves affect hydrodynamics in each of the lakes. To address this, we have added sentences on the manuscript (See Page 14, L7-17: "The main limitation in the model and resulting simulations is the assumption of one-dimensionality in both the model and field data. Quantifying the uncertainty from this limitation can be challenging and difficult (Gal et al., 2014; Tebaldi et al., 2005). Small, stratified lakes generally lack large horizontal temperature gradients (Imberger and Patterson, 1981, Kamarainen et al., 2009), allowing the assumption of one-dimensionality to be appropriate. However, short-term deviations in water temperature and thermocline depth may exist due to internal wave activity, especially in larger lakes (Tanentzap et al., 2007, Kamarainen et al., 2009), and spatial variations in wind stress can produce horizontal variations in temperature profiles (Imberger and Parker, 1985, Kimura et al., 2016). To address the role of internal wave activity and benthic boundary layer mixing, the pseudo two-dimensional deep mixing model by Yeates and Imberger (2003) is employed here. This mixing model has been shown to accurately characterize deep mixing that distributes heat from the epilimnion into the hypolimnion, thus weakening stratification, and the rapid distribution of heat entering the top of the hypolimnion from benthic boundary layer mixing, which strengthens stratification (Yeates and Imberger, 2003).

10. Using wording 'increasing (decreasing)' is clumsy for the reader

The authors thank the reviewer for this comment. We have revised the text to make the writing clearer for readers.

11. Throughout the text there is repetition, e.g. in 'Results' sentences which belong to 'Material' and are already tackled there. Check the whole manuscript for that

We thank the reviewer for pointing out this issue in the manuscript. The authors have removed and moved sections of the manuscript to address this issue.

12. Table 1: The meaning of the row 'Groundwater' is not clear to me

The table is revised to show groundwater inflow type of the lakes. For example, "discharge" lakes are those which have a net groundwater discharge into the lake. "Flowthrough" lakes are those which have small net inflow or outflow. Specifically, Lake Wingra have high groundwater inflows and outflows which result in small net inflow into the lake.

13. In Figure 4, the legend contains some description of results

The authors thank the reviewer for pointing out this comment. We have removed the description of the results out of the legend.

14. In Figure 6, results on Lake Wingra should be left out (= zero line). And in general, the stability index is somehow funny in this context since the lake was known to be polymictic

As suggested by Reviewer 1, we remove Lake Wingra out of this figure. To address stability, please see our previous response concerning the inclusion of Lake Wingra.

15. The real discussion starts in 4.4. and all before that should be merged with 'Results'. An indication of that is the fact that for instance in 4.3.1 and 4.3.2 there are no references in the text.

The authors thank the Reviewer 1 for this comment. As suggested, we have restructured the discussion by moving Sections 4.2 and 4.3 to the Results section and emphasized the results in terms of physical mechanisms that influence the simulated and observed responses.

16. There are some spelling mistakes in the text, please check.

The typos have been fixed within the manuscript. Thank you to the reviewer for pointing out these mistakes to the authors. We have re-reviewed the manuscript carefully for any typographical errors.

**Response to Reviewer 2**

• The authors use an extensive dataset on water temperatures from three neighboring lakes to test and validate a one-dimensional lake temperature model. The model is subsequently used for reconstruction of the thermal and stratification regime of the lakes during the last century and for sensitivity studies exploring the lake response to changes in mean annuals of air temperature and wind speed. The idea behind the sensitivity experiments is to elucidate the dissimilarity in the response of lakes with different depths and surface areas subject to identical external atmospheric forcing. The problem statement is clear. The methods are generally relevant to the questions stated in the study (except the application of a 1d time-depth model to investigation of the effects of horizontal extensions on lake thermics, which requires additional justification, see below).

The authors thank the reviewer for the positive comments and insightful comments on the manuscript. We have addressed the comments in a point-by-point reply on the revised manuscript.

• My major concern is the analysis of the results, which looks superficial, and representation of the outcomes, which is lengthy and poorly structured. The analysis is confined to descriptive presentation of model outcomes without an insight into the physical mechanisms producing the observed effects.

We appreciate the comment concerning the analysis of results and structure of the paper. We have restructured the manuscript to address the points raised in both reviews and performed additional analysis and discussion to provide more insight into the physical mechanisms producing the observed effects.

• Verbal presentation of trends in lake thermal characteristics covering several paragraphs is exhausting and not really informative.

The authors thank the reviewer for this comment. We have addressed the verbal presentation of trends in lake thermal characteristics in Sec. 3.5, specifically. For the revised manuscript, we avoid listing the numerical results one-by-one without meaningful interpretation. Instead, we present the results in Figure 7 and Table 4. We summarize the overall and trends and related thermal characteristics in the manuscript (see Page 11, Line 6-12).

• The manuscript presents a nice set of data and numerical results, which can serve as a basis for a well-thought study, but has little value for the reader in its present form.

Following the previous section, we have greatly revised throughout the manuscript (see the changes in Result Sections and Discussion Sections.

• The manuscript requires a more detailed description of the model and discussion on its uncertainties and relevance to the real lake processes; the discussion should be rethought, moving the accent from the descriptive listing of the model responses to varying inputs to the discussion on the physical mechanisms producing the responses.

The authors thank the Reviewer 2 for this comment. As suggested, we restructured the discussion by moving Sections 4.2 and 4.3 to the "Results" section and emphasizing results in terms of physical mechanisms that are influencing the simulated and observed responses. The model was first developed by other researchers, and detailed descriptions of the model are presented elsewhere, which can be found in references. For the revised manuscript, we have added more detailed description of processes and described the parameterization of horizontal processes to improve the discussion, specifically on effects of non-linear surface momentum and the role that fetch differences play on lake thermal structures. Additionally, we restructured the "Methods" section to provide details on parameterization, calibration, and gap-filling of data in the manuscript.

Here are some major critical points:

• Effects of lake surface area on the response to the atmospheric forcing are continuously mentioned throughout the manuscript and are among the main subjects of the model sensitivity runs. However, the entire discussion is based on the outputs of a one-dimensional model, i.e. none of the physical processes depending on the horizontal dimensions are modeled directly, but parameterized in the model. Hence, the response of the model outcomes to varying surface area does not necessarily coincide with the response of real lakes to the same perturbations. To analyze properly the modeling results the authors need to (i) present the details on the model parameterizations related to the effects of horizontal advection, wind fetch, horizontally varying depth, and other horizontal processes, such as mixing by internal waves and upwelling of hypolimnetic waters in near-shore areas of the lake; (ii) when discussing the modeling results state clearly which of them can be extrapolated on the real lakes, which horizontal processes are missed by the model, and how it can affect the real situations; (iii) differentiate between the effects produced by increase of the wind energy input due to larger surface area from those produced by increase of the thermal inertia due to larger lake volume, like, in particular, timing of the stratification onset (Section 4.3.1).

The authors thank the reviewer for this comment. Since the model was developed by previous studies. We do not repeat the description of model equations. Instead, we have listed specific equation numbers in references to be clear about which equations the model uses and how horizontal processes are parameterized in the model. Specifically, we address the reviewer's three comments as follows:

• Page 4, Line 12-30: Section 2.2 Model description for hydrodynamics modeling and parameterizations

"To hindcast water temperature and stratification in the three study lakes, we use the DYRESM-WQ (DYnamic REservoir Simulation Model-Water Quality; Hamilton and Schladow, 1997), which employs discrete horizontal Lagrangian layers to simulate vertical water temperature, salinity, and density with input including inflows, outflows, and mixing (Imberger et al., 1978). The model has been previously used on a variety of lake types and is accepted as a standard for hydrodynamic lake modelling (Gal et al., 2003; Hetherington et al., 2015; Imberger and Patterson, 1981; Kara et al., 2012; Tanentzap et al., 2007). DYRESM-WQ adopts a one-dimensional layer structure based on the importance of vertical density stratification over horizontal density variations. A one-dimensional assumption is based on observations that the density stratification found in lakes inhibits vertical motions while horizontal variations in density relax due to horizontal advection and convection (Antenucci and Imerito, 2003; Imerito, 2010). Surface exchanges include heating due to shortwave radiation penetration into the lake and surface fluxes of evaporation, sensible heat, long wave radiation, and wind stress (Imerito, 2010). Surface layer mixing is based on potential energy required for mixing, and introduction of turbulent kinetic energy through convective mixing, wind stirring, and shear mixing (Imerito, 2010; Yeates and Imberger, 2003). Yeates and Imberger (2003) improved performance of the surface mixed layer routine within the model by including an effective surface area algorithm (see Eq 32 in Yeates and Imberger, 2003) that reduced surface mixing in smaller, more sheltered lakes. Details of the surface mixed layer algorithm are not reproduced here, but can be found in Eq 27-34 of Yeates and Imberger (2003). Hypolimnetic mixing is parameterized through a vertical eddy diffusion coefficient, which accounts for turbulence created by the damping of basin-scale internal waves on the bottom boundary and lake interior (Yeates and Imberger, 2003). Detailed equations on the simulation of water temperature and mixing can be found in Imberger and Patterson (1981), and Yeates and Imberger (2003)."

• Page 14, Line 7-17 in Discussion for model limitation and uncertainty

The main limitation in the model and resulting simulations is the assumption of one-dimensionality in both the model and field data. Quantifying the uncertainty from this limitation can be challenging and difficult (Gal et al., 2014; Tebaldi et al., 2005) Small, stratified lakes generally lack large horizontal temperature gradients (Imberger and Patterson, 198, Kamarainen et al., 2009), allowing the assumption of one-dimensionality to be appropriate. However, short-term deviations in water temperature and thermocline depth may exist due to internal wave activity, especially in larger lakes (Tanentzap et al., 2007, Kamarainen et al., 2009), and spatial variations in wind stress can produce horizontal variations in temperature profiles (Imberger and Parker, 1985, Kimura et al., 2016). To address the role of internal wave activity and benthic boundary layer mixing, the pseudo two-dimensional deep mixing model by Yeates and Imberger (2003) is employed here. This mixing model has been shown to accurately characterize deep mixing that distributes heat from the epilimnion into the hypolimnion, thus weakening stratification, and the rapid distribution of heat entering the top of the hypolimnion from benthic boundary layer mixing, which strengthens stratification (Yeates and Imberger, 2003).

• Page 14, Line 27-Page 15 Line 16 by adding the importance of wind speeds in Discussion

While many have addressed the importance of changing air temperatures on water temperatures and water quality (e.g. Adrian et al., 2009; Arhonditsis et al., 2004; O'Reilly et al., 2015; Shimoda et al., 2011), fewer have investigated wind speed as a specific driver of changes to lakes (Magee et al., 2016; Snortheim et al., 2017). However, results here show that correlations between wind speeds and lake temperature variables are as high as, or higher than, correlations between air temperature and lake temperature variables (Fig. 7), highlighting the importance of wind speeds as drivers of lake temperature and stratification changes. For many variables (e.g. stratification dates, epilimnetic temperatures, stability), correlation is opposite for air temperature and wind speed variables, indicating that wind speed increases can offset the effects of air temperature increases, while locations with decreasing wind speeds may experience a greater impact on water temperature and stratification than with air temperature increases alone. This statement is further supported through sensitivity analysis on stratification onset and overturn (Fig. 8 and 9), which show that for Madison-area lakes, increasing air temperatures and decreasing wind speeds have a cumulative effect toward earlier stratification onset and later overturn. However, for hypolimnetic temperatures, correlations and sensitivity indicate that decreasing wind speeds cool hypolimnetic temperatures, while increasing air temperatures warm hypolimnetic temperatures. Arvola (2009) showed that hypolimnion temperatures were primarily determined by the conditions that pertained during the previous spring turnover, which is consistent with our results showing significant (p<0.01) correlation between hypolimnion temperatures and wind speed (Fig. 8), but no significant correlation with air temperature or summer conditions. This could explain the conflicting results of previous research showing both warming and cooling trends in different lakes (Gerten and Adrian, 2001). Hindcasted hypolimnion temperatures (Fig. 4) show decreasing trends for Lake Mendota and Fish Lake. Combining the effects of air temperature and wind speed suggests that decreasing wind speeds, instead of increasing air temperatures, plays a more important role to change hypolimnetic water temperature for both lakes.

- Finally, we also revise the section 4.3.1 Lake Depth and 4.3.2 Surface Lake Area that address the reviewer's comment (iii).
- Do the lakes have ice cover in winter? The ice model is repeatedly mentioned in the MS, but no results on the ice regime are presented/discussed. Duration of the ice-covered period directly affects timing of the summer stratification onset and summer hypolimnetic temperatures. Any discussion on these variables is incomplete without considering the ice regime.

The authors thank the reviewer for pointing out this critical point. These lakes do have ice cover in winter. A recent paper by Magee and Wu (2016) in *Hydrological Processes* details both the ice model and the impact of air temperature changes on the three study lakes. As a result, we do not repeat the ice model and specific results of ice cover changes. Instead, we add text and make it clear that the lakes are in fact ice covered. Furthermore, we included analysis of the impact of ice cover on stratification onset and hypolimnetic temperatures in new Fig. 8 and in the results and discussion sections.

**Reference:**

Magee, MR and Wu, CH (2016) Effects of changing climate on ice cover in three morphometrically different lakes. *Hydrological Processes*. DOI: 10.1002/hyp.10996.

• Section 4.3 Sensitivity runs can be shortened, at least, to a half and moved from 'Discussion' to 'Results'. The actual discussion should be added, considering the reasons for the observed dependencies, their relevance to the processes in real lakes and novelty of the results compared to the state-of-the-art in this area of research.

The authors agree with this suggestion from the reviewer, and we have moved Section 4.3 to the Results section, shortened the presentation of the results, and have added discussion on the reasons for the observed relationships and add "4.2 Importance of wind-speed" as follows:

"While many have addressed the importance of changing air temperatures on water temperatures and water quality (e.g. Adrian et al., 2009; Arhonditsis et al., 2004; O'Reilly et al., 2015; Rimmer et al., 2011, Shimoda et al., 2011), fewer have investigated wind speed as a specific driver of changes to lakes (Magee et al., 2016; Snortheim et al., 2017). However, results here show that correlations between wind speeds and lake temperature variables are as high as, or higher than, correlations between air temperature and lake temperature variables (Fig. 7), highlighting the importance of wind speeds as drivers of lake temperature and stratification changes. For many variables (e.g. stratification dates, epilimnetic temperatures, stability), correlation is opposite for air temperature and wind speed variables, indicating that wind speed increases can offset the effects of air temperature increases, while locations with decreasing wind speeds may experience a greater impact on water temperature and stratification than with air temperature increases alone. This statement is further supported through sensitivity analysis on stratification onset and overturn (Fig. 8 and 9), which show that for Madison-area lakes, increasing air temperatures and decreasing wind speeds have a cumulative effect toward earlier stratification onset and later overturn. However, for hypolimnetic temperatures, correlations and sensitivity indicate that decreasing wind speeds cool hypolimnetic temperatures, while increasing air temperatures warm hypolimnetic temperatures. Arvola (2009) showed that hypolimnion temperatures were primarily determined by the conditions that pertained during the previous spring turnover, which is consistent with our results showing significant (p < 0.01) correlation between hypolimnion temperatures and wind

speed (Fig. 8), but no significant correlation with air temperature or summer conditions. This could explain the conflicting results of previous research showing both warming and cooling trends in different lakes (Gerten and Adrian, 2001). Hindcasted hypolimnion temperatures (Fig. 4) show decreasing trends for Lake Mendota and Fish Lake. Combining the effects of air temperature and wind speed suggests that decreasing wind speeds, instead of increasing air temperatures, plays a more important role to change hypolimnetic water temperature for both lakes.

Minor comments:

• P3L16 What is 'thermocline shifts'? Please, explain.

The authors thank Reviewer 2 for pointing out confusion due to our choice of word. 'Thermocline shifts' refers to changes in thermocline depth in response to a driver such as changes in climate. We have changed the line to read "changes in thermocline depth from warming air temperatures may be dampened..." to remove some of this confusion due to previous word choice.

• P6L29 Provide model parameters and simulation specifications here.

We have provided addition parameters and simulation specifications as suggested within the text (see previous reply or Model description Page 4-Line 13-30, Page 5-Line 11-Page 6 Line 3)

• P9L7 Add 'summer epilimnetic' to 'temperatures'

We did add 'summer epilimnetic' to temperature, as requested by the reviewer.

• P10L13 and other appearances: replace '0.067 days earlier decade-1 ' to '+0.067 days decade-1 '

Following the suggestion by the reviewer, the authors have made this change to improve readability of the manuscript.

• P10L28 onwards: 'J m-2' are not correct units for heat flux. Provide flux values in understandable units.

The authors thank the Reviewer 2 for pointing out this error in units. Indeed, the units should be  $W m^{-2}$ , and the error occurred by inadvertently carrying over units from the previous sections of text). The units are correct in the corresponding figure. We have addressed the incorrect units in the text.

• P11L17 How lake morphometry can affect the shortwave flux of solar radiation??

The shortwave flux is the net flux at the surface of each lake. The shortwave flux is controlled in part by albedo of the surface water, by snow ice cover in the lake. Each lake may have slightly different net shortwave radiative flux for each day and average for the year.

• P14L12 and at other places: Schmidt stability is irrelevant to non-stratified lakes and cannot be used for comparison.

**P17L9 See above**

Lake Wingra does stratify on daily or weekly timescales during the summer months (Kimura et al, 2016). Summer Schmidt stability was calculated at daily timescales, and then averaged for each year before comparing coherence among the lake pairs. Higher average stability for one year on Lake Wingra would indicate that the lake experienced more days of stratification during the period. This phenomenon can be coherent with changes in stability for the other two lakes.

• P17L18 Evaporation depends on surface temperatures, not the deep water temperatures. Explain what do you mean in this sentence, or remove it and find another explanation for the phenomenon.

We have re-written this section as follows (Page15 Line 30-Page 16 Line7): "Overall, Lake Wingra had a larger magnitude of latent and net heat fluxes than the deeper lakes. Diurnal variability in surface temperatures is larger for shallow lakes, promoting increased latent heat fluxes in these lakes (Woo, 2007). This increased response may also explain the larger change in trend for sensible heat flux since Lake Wingra responds more quickly to changes in air temperature, thus, have a larger change in sensible heat flux during each day. Interestingly, net heat flux of Lake Wingra is less coherent with the deeper lakes than the deep lakes are with each other. This may be due to the combination of more extreme temperature variability, increasing sensible and latent heat fluxes during the open water season and the lower sensitivity of ice cover duration in Lake Wingra compared to the deeper lakes (Magee and Wu, 2016)."

• P17L2529 Actually, the main driver for epilimnetic temperatures is solar radiation not air temperature. If air temperature is the 'main driver', what do you mean under 'wind. . . a more dominant mechanism'?

The authors agree that solar radiation is the main driver for epilimnion temperatures. Nevertheless, Air-temperature increase is a natural candidate to explain the increase in the average epilimnion temperature at both short (monthly) and long (annual) timescale (Livingstone, 2003, Rimmer et al, 2011, Magee et al., 2016). What we mean is that we examine epilimnetic temperature change by running sensitivity analysis through changing air temperature and wind speed scenarios. What we find is that wind mixing is a more dominant mechanism to transfer heat from upper layers of the water column to bottom waters than is molecular diffusion of heat. While air temperature can directly influence surface water temperatures, wind speed changes can dissipate heat to the lower water levels and can act to change the response of epilimnion temperatures to air temperature changes. To clarify this confusion, we have re-written the section as follows (Page16, Line 12-17): "Increasing air temperatures are well documented to increase epilimnetic water temperatures (Livingstone, 2003; Robertson and Ragotzkie, 1990), since air temperature drives heat transfer

between the atmosphere and lake (Boehrer and Schultze, 2008; Palmer et al., 2014). However, wind mixing can act as a mechanism of heat transfer (Nõges et al., 2011), and cool the epilimnion through increased surface mixed-layer deepening. Decreasing wind speeds may increase epilimnion temperatures above that from air temperature increases alone (Fig. 8)."

• P18L14-15 Explain, why stronger winds should produce higher spatial variability in wind stress. How did you estimate changes in turbulence and why do you think they are nonlinear?

We address this in the following. In the text, we do not imply that there is higher spatial variability in wind stress within the lakes themselves. Rather, increases and/or decreases in wind speed in general will result in nonlinear changes in wind stress and turbulence in all lakes. Wind stress varies with the square of wind speed, so changes in wind speed directly result in non-linear changes in wind stress on the water surface. The DYRESM model parameterizes mixing within the model by estimating the turbulent kinetic energy (TKE) and mixing layers when a potential energy threshold is exceeded. TKE in the model is introduced through convective mixing, wind stirring, and shear mixing using parameterizations that are all non-linear equations and influenced nonlinearly either directly or indirectly by wind speed. So linear changes in wind speed yield nonlinear changes in the turbulence estimation in the model. To clarify this, we have added the text in the manuscript (Page 16-Line 17- Page 17 Line 2) as follows:

"Surface area plays a role in lake-wide average vertical heat fluxes from boundary processes (Wüest and Lorke, 2003), and the model accounts for this by including an effective surface area algorithm to scale transfer of momentum from surface stress based on lake surface area (Yeates and Imberger, 2003). This increases transfer momentum from surface stress and vertical heat transfer for lakes with larger fetch. Accounting for this larger fetch increases mixing and vertical transfer of heat to bottom waters, reducing epilimnion water temperatures (Boehrer and Schultze, 2008) and increasing the rate of lake cooling (Nõges et al., 2011). For this reason, Lake Mendota with the large fetch experiences a smaller increase in epilimnetic water temperature compared to Fish Lake (Table 5). Additionally, momentum from surface stress scales linearly with lake area and non-linearly with wind speed (Yeates and Imberger, 2003, see Eq. 31 and 33), making momentum from surface stress, and thus, mixing, stratification, and hypolimnion temperatures more variable for lakes with larger fetch and even more variable when wind speed is increased (see Fig. 8-10). Greater variability in momentum and mixing corresponds to larger variability of Schmidt stability for Lake Mendota, with the larger surface area. Greater transfer of momentum in Lake Mendota results in the slightly deeper thermocline for the larger surface area lake (~10 m in Lake Mendota and ~6 m in Fish Lake), which may play a role in filtering the climate signals into hypolimnion temperatures. Low hypolimnetic temperature coherence between Mendota and Fish suggest that lake morphometry plays a role. This result is consistent with other studies that show lake morphometry parameter affects the way temperature is stored in the lake system (Thompson et al., 2005)."

• Table 2, Fig. 3: The model seems to produce consistently a positive bias in lake temperatures. Any explanation for this?

The model results under predict slightly water temperatures. This under prediction is from a combination of averaging meteorological inputs over the day and comparing temperatures output on a daily timestep with observations collected typically during the afternoon when water temperatures are slightly higher than daily averages.

Typos:

- P4L12 Capitalize 'Secchi'
- P5L29 remove second appearance of 'Lake Mendota'
- P8L15 replace 'decreased' with 'decrease'
- P12L13 replace 'difficulty' with 'difficult'

All the typos have been fixed within the manuscript. We extend our many thanks to the reviewer for pointing out these mistakes to the authors. Furthermore, we have reviewed the manuscript carefully for typographical errors.

[revised manuscript text omitted]

**Commented [M1]: What is 'thermocline shifts'? Please, explain**

**Commented [MRM2R1]:** The authors thank Reviewer 2 for pointing out confusion due to our word choice, 'thermoeline shifts' refers to changes in thermoeline depth in response to a driver such a changes in climate. We have changed the line to read "changes in thermoeline depth may be dampened..." to remove some of this confusion due to previous word choice.

**2 Methods**

**2.1 Study sites**

Three morphometrically different lakes, Lake Mendota, Fish Lake, and Lake Wingra, located near Madison, Wisconsin, United States of America (USA), were selected for this study. These lakes are chosen for (i) their morphometry differences, (ii) their elose proximity proximity to one another, and (iii) the availability of long-term limnological recordsdata for model input and calibration.

Lake Mendota (43°6' N; 89°24'W; Figure 1a; Table 1), is a dimictic, eutrophic, drainage lake in an urbanizing agricultural watershed (Carpenter and Lathrop, 2008). The lake stratifies during the summer, and typical stratification periods lasts from

- 10 May to September. During the summer monthsSummer (1 June 31 August), the mean surface water temperature is 22.4 °C, and hypolimnetic temperatures range in value vary between from 11°C to 15 °C. Normal sSecchi depth during the summer is 3.0 meters (Lathrop et al., 1996). Fish Lake (43°17'N; 89°39'W; Fig.ure 1b; Table 1) is a dimictic, eutrophic, shallow seepage lake located in northwestern Dane County. From 1966 to 2001, lake level the water level of the lake rose by 2.75 meters the to increased groundwater flow from higher than normal regional groundwater recharge (Krohelski et al., 2002). Krohelski et al., 2002).
- 15 al. (2002) hypothesized that the increase in recharge may be the result of increased infiltration from snowmelt after increased snowfall and less frost-covered soil. Summer stratification lasts The lake experiences summer stratification lasting from the beginning of May to mid-September. Mean surface water temperature 23.9°C and hypolimnetic temperatures are normally near 8°C during summer months; however, some years reach temperatures of only 5-6 °C in the hypolimnion due to shortened spring mixing durations. however, some years do not experience complete mixing in the spring and reach temperatures of
- 20 only 4-5°C in the bottom waters by the end of the summer. The aAverage Secchi depth during the summer months is 2.4 m. Lake Wingra (43°3' N; 89°26' W; Fig.ure 1c; Table 1) is a very-shallow, eutrophic, drainage lake. It stratifies on short timescales of hours to weeks (Kimura et al., 2016), but does not experienced sustained thermal stratification. Due to its shallow depth, Lake Wingra does not experience thermal stratification in the summer. During the summerSummer, the mean water temperature is 23.9°C3 and mean Ssecchi depth is 0.7 meters. All three lakes have ice cover during winter months, and a description of ice on the lakes can be found in Magee and Wu (2016)

2.2 Data

Meteorological data used in the model input consisted of daily solar radiation, air temperature, vapor pressure, wind speed, eloud cover, rainfall, and snowfall over a period of 104 years from 1911 to 2014. Air temperature, wind speed, vapor pressure, and cloud cover were computed as an average of the whole day, while solar radiation, rainfall, and snowfall were the daily

**Commented [M3]:** The readers would benefit from some more information about the lakes. Especially information on lake clarity (water colour etc; cf. Table 1) would have been useful in a paper with such a strong focus on lake stratification.

As a reader, I would also appreciate information on fetch for each lake; now the word 'fetch' and importance of fetch is mentioned several times, but the reader is left with the bathymetric maps to figure out the fetch.

**Commented [MRM4R3]:** Information on lake fetch has been added to Table 1 to provide that detail to the readers. The authors apologize for neglecting to include this value explicitly in the original submission of the manuscript.

Additional information on the lakes has been added to Table 1, including fetch, shoreline development, landscape position, Secchi depth, surface water chlorophyll concentration, and DOC in each lake. Specific values of lake water color is not collected by the NTL LTER program as other data was, however,

**Commented [M5]:** It is said that water level in Fish Lake has raised considerably and this has probably affected some of the results. However, nothing is said about the possible reason behind this phenomenon. Related to climate, human activity or what?

**Commented [MRM6R5]:** The text has been changed to read "the water level of the lake rose by 2.75 meters due to an increase in regional groundwater recharge causing increased groundwater flow to the lake (Krohelski et al., 2002). Krohelski et al. (2002) hypothesized that the increase in regional groundwater recharge may be the result of increased infiltration from snowmelt after increased snowfall and less frost-covered soil."

**Field Code Changed**

**Commented [M7]:** The authors state that Fish Lake does not always turn over completely in spring. This is an important piece of information, since in small, dark coloured boreal lakes this is a fairly common observation and it is believed that it is weather/climate driven change. It would be nice if the authors could dig deeper in this observation, especially since they have such a long time series.

**Commented [M8]:** Do the lakes have ice cover in winter? The ice model is repeatedly mentioned in the ms, but no results on the ice regime are presented/discussed. Duration of the ice-covered period directly affects timing of the summer stratification onset and summer hypolimnetic temperatures. Any discussion on these variables is incomplete without considering the ice regime.

5

totals. Meteorological data was gathered from Robertson (1989), who compiled a continuous daily meteorological dataset for Madison Wisconsin from 1884 to 1988 by adjusting for changes in site location. Appended to this dataset is data from the National Climate Data Center weather station at the Dane County Regional Airport. All data other than solar radiation can be obtained from http://www.ncdc.noaa.gov/, for Madison (MSN), and solar radiation can be obtained from http://www.sws.uiuc.edu/warm/weather/. Adjustments to wind speed were made based on changes in observational techniques

- occurring in 1996 (McKee et al., 2000) by comparing data from Dane County Airport with that collected from the Atmospheric and Oceanic Science Building instrumentation tower at the University of Wisconsin Madison (http://ginsea.aos.wisc.edu/labs/mendota/index.htm). Detail of this adjustment can be found in Magee et al. (2016).
- 10 Seasonal Secchi depths were used to determine the light extinction coefficients. Lathrop et al. (1996) compiled Secchi depth data for Lake Mendota between 1900 and 1993 (1701 daily Secchi depth readings from 70 calendar years), and summarized the data for six seasonal periods: winter (ice on to ice out), spring turnover (ice out to 10 May), early stratification (11 May to 29 June), summer (30 June to 2 September), destratification (3 September to 12 October), and fall turnover (13 October to ice-on). After 1993, Seechi depths are obtained from the North Temperate Lake Long Term Ecological Research (NTL-LTER) program (). For Fish Lake and Lake Wingra, Secchi depths were compiled for 1995 to the present from the NTL LTER 15
- program, For years with no Seechi data, the long-term mean seasonal Seechi depths were used. Light extinction coefficients were estimated from Secchi depth using the equation from Williams et al., (1980):

(1)

where k is the light extinction coefficient and  $z_{*}$  is the Secchi depth (m).

20

k = 1.1/

5

Inflow and outflow measurements were collected from gauging stations (http://waterdata.usgs.gov/wi/nwis/sw/), and complied to calculate daily totals. In cases where inflow and outflow measurements were not available, inflow and outflow were estimated as the residual unknown term of the water budget balancing precipitation, evaporation, and lake level. The residual term was distributed evenly across the number of days between water level measurements. For Lake Mendota, water level was recorded since 1916 (http://waterdata.usgs.gov/wi/nwis/dv). Water level at Fish Lake was recorded almost daily from 1966-25 2003 (http://waterdata.usgs.gov/wi/nwis/dv/?site\_no=05406050&agency\_cd=USGS&referred\_module=sw). For Lake Wingra, water level was recorded sporadically during the period of interest. When lake level information was unavailable, the long-term mean lake level was assumed for water budget calculations. Only Lake Mendota has inflowing surface water streams. Inflow temperatures were estimated following the method in (Magee et al., 2016). Groundwater temperature measurements near each lake were used to estimate the temperature of groundwater fluxes.

6

30

Commented [M9]: Data on below-ice Secchi-depth were used

Commented [MRM10R9]: Light extinction, which can be

Commented [M11]: Fish Lake and Lake Wingra have Secchidepth results only from 1995 onwards. This appears problematic; could you give more explanation on this?

Commented [MRM12R11]: Fish Lake and Lake Wingra became part of the NTL-LTER program in 1995 and regular Secchi depth measurements were taken starting then. The authors agree with Reviewer 1 that it is not ideal to use seasonal averages for the historical period before 1995; however, given the strongly seasonal dynamic of water clarity and light extinction in the lakes, using seasonal averages of Secchi depth to estimate light extinction are preferably to a constant light extinction for the lakes, which is not representative of observed phenomenon in the lakes. The authors have added comments about the uncertainty and errors caused by this assumption in the discussion.

Observation data used for model calibration came from a variety of sources. For Lake Mendota, long term water temperature records for Lake Mendota were collected from Robertson (1989) and the NTL-LTER (2012b). Ice thickness data were gathered from E. Birge, University of Wisconsin (unpublished); D. Lathrop, Wisconsin Department of Natural Resources (unpublished); Stewart (1965); and the NTL-LTER program (2012a). Frequency of temperature data varied from one or two profiles per year to several profiles for a given week. Additionally, the vertical resolution of the water profiles varied greatly. For Fish Lake

and Lake Wingra, water temperature data were collected from NTL-LTER only from 1996-2014 (2012b).

**2.23 Model description**

5

To hindcast water temperature and stratification in the three study lakes we use Thethe vertical heat transfer model, DYRESM-

- 10 WQ (DYnamic REservoir Simulation Model-Water Quality; Hamilton and Schladow, 1997), model (Hamilton and Schladow, 1997), which employs discrete horizontal Lagrangian layers to simulate vertical water temperature, salinity, and density with input including inflows, outflows, and mixing (Imberger et al., 1978). The model has been previously used on a variety of lake types and is accepted as a standard for hydrodynamic lake modelling (Gal et al., 2003; Hetherington et al., 2015; Imberger and Patterson, 1981; Kara et al., 2012; Tanentzap et al., 2007). DYRESM-WQ adopts a one-dimensional layer structure based on
- 15 the importance of vertical density stratification over horizontal density variations. A one-dimensional assumption is based on observations that the density stratification found in lakes inhibits vertical motions while horizontal variations in density relax due to horizontal advection and convection (Antenucci and Imerito, 2003; Imerito, 2010). Surface exchanges include heating due to shortwave radiation penetration into the lake and surface fluxes of evaporation, sensible heat, long wave radiation, and wind stress (Imerito, 2010). A one dimensional layer structure is adopted based on the vertical density stratification over
- 20 horizontal density variations and destabilizing forces such as wind stress and surface cooling abbreviated to ensure a one dimensional structure (Antenucci and Imerito, 2003). Surface layer mixing is based on potential energy required for mixing, and introduction of turbulent kinetic energy through convective mixing, wind stirring, and shear mixing (Imerito, 2010; Yeates and Imberger, 2003). Mixing and surface layer dynamics depend on a turbulent kinetic energy budget and potential energy required for mixing (Hamilton and Schladow, 1997; Sherman et al., 1978). Yeates and Imberger (2003) improved performance
- 25 of the surface mixed layer routine within the model by including an effective surface area algorithm (see Eq 32 in Yeates and Imberger, 2003) that reduced surface mixing in smaller, more sheltered lakes. Details of the surface mixed layer algorithm are not reproduced here, but can be found in Eq 27-34 of Yeates and Imberger (2003). Hypolimnetic mixing is parameterized through a vertical eddy diffusion coefficient, which accounts for turbulence created by the damping of basin-scale internal waves on the bottom boundary and lake interior (Yeates and Imberger, 2003). Detailed equations More information on the

7

**Commented [M13]:** I find it strange that in a paper where models are such an elemental part, they are not properly described; besides the equation of light extinction, the authors only use references to published articles. More emphasis should also be given to description of gap filling and calibration data; both are now somewhat superficial.

**Commented [M14]:** The manuscript requires a more detailed description of the model and discussion on its uncertainties and relevance to the real lake processes.

[revised manuscript text omitted]

**Commented [M15]:** Provide model parameters and simulation specifications here

**Commented [M16]:** Do the lakes have ice cover in winter? The ice model is repeatedly mentioned in the ms, but no results on the ice regime are presented/discussed. Duration of the ice-covered period directly affects timing of the summer stratification onset and summer hypolimmetic temperatures. Any discussion on these variables is incomplete without considering the ice regime.

**Formatted: Heading 3**

**Commented [M17]:** Data on below-ice Secchi-depth were used which I to a certain extent understand, but since it is not that common practice to measure Secchi under the ice, it would be useful to have some more information.

**Commented [MRM18R17]:** Light extinction, which can be estimated from Secchi depth, greatly influences water temperatures and overall temperature profile. Including light extinction in winter more reliably reproduces under-ice water temperatures and as a result, water temperatures at the time of ice-off. Temperature profiles at ice-off impact the timing of stratification and the hypolimnetic water temperature through the summer, so properly characterising and capturing these phenomena in the model enable a more accurate reproduction of water temperatures during the historical period. For this study, the authors choose to utilize the available data from previous ecological and water quality studies conducted on the lakes to better inform the model and more accurately reproduce water temperature profiles. 1995 
[revised manuscript text omitted]
:  $\pm 0.225^{\circ}$ C decade-1; spring  $\pm 0.165^{\circ}$ C decade-1; summer  $\pm 0.081^{\circ}$ C decade-1; fall  $\pm 0.110^{\circ}$ C decade-1; p<0.05) increased over the periodfrom 1911–2014 (Figure 2a). Yearly air temperature increased at a rate of  $0.145^{\circ}$ C decade+1 (p<0.01); winter air temperature

- 15 increased at a rate of 0.225°C decade 1 (p<0.01); spring air temperature increased at a rate of 0.165°C decade+ (p<0.01); summer air temperature increased at a rate of 0.081°C decade-1 (p<0.05); and fall air temperature increased at a rate of 0.110°C decade-1 (p<0.05). Additionally, All five sets of data were further analysed for significant changes in slope and for abrupt changes in mean. Yyearly average air temperature, but not seasonal temperatures, showed a significant change in slope from 0.081°C decade-1 temperature increased at a rate of 0.081°C decade-1 temperature increased at a rate of 0.110°C decade-1 (p<0.05). Additionally, All five sets of data were further analysed for significant changes in slope and for abrupt changes in mean. Yyearly average air temperature, but not seasonal temperatures, showed a significant change in slope from 0.081°C decade-1 temperature in 1981, and summer air temperatures showed three significant abrupt changes
- 20 in mean value (Fig. 2a). Yearly (-0.073 m s-1 decade-1; p<0.01) and seasonal average (winter: -0.083 m s-1 decade-1; spring = 0.071 m s-1 decade-1; summer: -0.048 m s-1 decade-1; fall: -0.088 m s-1 decade-1; p<0.01) wind speeds decreased from 1911-2014 (Fig. 2b). Wind speeds for both yearly and seasonal average exhibited significant decreased in trend over the period 1911–2014 (Figure 2b). Yearly wind speed decreased at a rate of 0.073 m s-1 decade-1 (p<0.01); winter decreased at a rate of 0.083 m s-1 decade-1 (p<0.01); spring decreased at a rate of 0.071 m s-1 decade-1 (p<0.01); summer decreased at a rate of 0.084 m s-1 decade-1 (p<0.01); summer decreased at a rate of 0.084 m s-1 decade-1 (p<0.01); summer decreased at a rate of 0.084 m s-1 decade-1 (p<0.01); summer decreased at a rate of 0.084 m s-1 decade-1 (p<0.01); summer decreased at a rate of 0.084 m s-1 decade-1 (p<0.01); summer decreased at a rate of 0.084 m s-1 decade-1 (p<0.01); summer decreased at a rate of 0.084 m s-1 decade-1 (p<0.01); summer decreased at a rate of 0.084 m s-1 decade-1 (p<0.01); summer decreased at a rate of 0.084 m s-1 decade-1 (p<0.01); summer decreased at a rate of 0.084 m s-1 decade-1 (p<0.01); summer decreased at a rate of 0.084 m s-1 decade-1 (p<0.01); summer decreased at a rate of 0.084 m s-1 decade-1 (p<0.01); summer decreased at a rate of 0.084 m s-1 decade-1 (p<0.01); summer decreased at a rate of 0.084 m s-1 decade-1 (p<0.01); summer decreased at a rate of 0.084 m s-1 decade-1 (p<0.01); summer decreased at a rate of 0.084 m s-1 decade-1 (p<0.01); summer decreased at a rate of 0.084 m s-1 decade-1 (p<0.01); summer decreased at a rate of 0.084 m s-1 decade-1 (p<0.01); summer decreased at a rate of 0.084 m s-1 decade-1 (p<0.01); summer decreased at a rate of 0.084 m s-1 decade-1 (p<0.01); summer decreased at a rate of 0.084 m s-1 decade-1 (p<0.01); summer decreased at a rate of 0.084 m s-1 decade-1 (
- 25 m s+ decade+ (p<0.01); and fall decreased at a rate of 0.088 m s+-decade+ (p<0.01), Significant shifts (p<0.01) in the mean occurred in the mid-nineties for all seasons, but there were no changes in rate of wind speed decreases. Additionally, all five sets of wind speed data showed statistically significant abrupt changes in the mean value occurring in the mid-nineties. For yearly average wind speed, a shift from 4.43 m s+-to 3.74 m s+ (p<0.01) occurred after 1994; for winter wind speeds, a shift

13

**Commented [M19]:** I find it a bit strange that sediment heat fluxes were hardly mentioned in this paper. Although there may have been no data on this or these fluxes were not included in models, they should have been tackled somehow at least in "discussion"

**Commented [M20]:** My major concern is with analysis of the results, which looks superficial, and representation of the outcomes, which is lengthy and poorly structured. The analysis is confined to descriptive presentation of model outcomes without any insight into the physical mechanisms producing the observed effects. Verbal presentation of trends in lake thermal characteristics covering several paragraphs is exhausting and not really informative.

from 4.72 m s4-to 3.92 m s4 (p<0.01) occurred after 1997; for spring wind speeds, a shift from 4.59 m s4 to 3.90 m s4 (p<0.01) occurred after 1996; for summer, a shift from 3.70 m s4 to 3.66 m s4 (p<0.01) occurred after 1994; and for fall, a shift from 4.64 m s4 to 3.75 m s4 (p<0.01) occurred after 1994.

**3.2 Model evaluation**

[revised manuscript text omitted]

- 20 sensible heat flux; (d) latent heat flux; and (e) total heat flux on over the 104-year period on the three study lakes are examined here. Magnitude of shortwave, longwave, and sensible heat fluxes are similar for all three lakes, but Lake Wingra has a larger magnitude of both latent and net heat fluxes. Net longwave is negative for all three lakes and increased in magnitude (Table 4), and sensible heat flux decreased in magnitude (became less negative; Table 4). There is no significant trend in other surface heat flux variables. Lake Wingra has a much smaller change in trend for longwave radiation than Mendota or Fish, but a larger
- 25 change in trend for sensible heat flux, indicating that depth likely influences the response of those heat fluxes to air temperature and wind speed changes.

While there is no statistically significant trend in shortwave flux, latent flux, or total heat flux, figure 7c shows longwave heat flux exhibits trend toward larger magnitude flux (decreasing absolute value; -5.85 J m-2 for Lake Mendota, -5.80 J m-2 for Fish Lake, and -4.59 J m-2 for Lake Wingra, *p*<0.05 for all three lakes) and sensible heat fluxes displays an increasing trend (less negative values; 4.10 J m-2 for Lake Mendota, 3.65 J m-2 for Fish Lake, and 5.65 J m-2 for Lake Wingra, *p*<0.05 for all three

16

**Commented [M23]:** Make this change elsewhere throughout the manuscript

**Commented [MRM24R23]:** The authors thank the reviewer for this comment. We have made the appropriate changes throughout the manuscript as requested.

**Commented [M25]:** Some parts of the paper are also technically challenging for the reader since they are based on listing the numerical results one by one: a good example is section 3.5

lakes). For shortwave radiation, an abrupt change from  $1.14 \text{ Jm}^2$  to  $4.92 \text{ Jm}^2$  occurred in Lake Mendota after 1992(p < 0.01); a change from  $1.97 \text{ Jm}^2$  to  $3.34 \text{ Jm}^2$  after 1945 (p < 0.01) and from  $3.34 \text{ Jm}^2$  to  $4.84 \text{ Jm}^2$  after 1992 (p < 0.01) for Fish Lake; and from  $4.34 \text{ Jm}^2$  to  $4.07 \text{ Jm}^2$  after 1937 (p < 0.01) and from  $4.07 \text{ Jm}^2$  to  $-5.98 \text{ Jm}^2$  after 1992 (p < 0.01) for Lake Wingra. Net heat flux shows no significant abrupt change for Lake Mendota, two changes for Fish Lake ( $0.33 \text{ Jm}^2$  to  $4.00 \text{ Jm}^2$

- $\frac{1}{1000} \frac{1}{1000} \frac{1}{100$
- 10 Jm2 after 1937 (p<0.01), and -0.81 J m2 to 1.42 J m2 after 1981 (p<0.01). Sensible heat flux had an abrupt change after 1926 for Lake Mendota (-1.67 J m2 to 0.26 J m2, p<0.01) and after 1921 for both Fish Lake (-2.23 J m2 to 0.26 J m2, p<0.01) and Lake Wingra (-3.68 J m2 to 0.36 J m2, p<0.01). While the timing of the abrupt change was similar in all three lakes, the magnitude of the change appears to increase with lake depth. Latent heat flux shows statistically significant (p<0.01) changes in mean after 1926 (Lake Mendota 5.78 J m2 to -2.32 J m2; Fish Lake 5.43 J m2 to -2.14 J m2; Lake Wingra 6.42 J m2 to -2.32 J m2 to -2.32 J m2; Fish Lake 5.43 J m2 to -2.14 J m2; Lake Wingra 6.42 J m2 to -2.32 J m2 to -2.32 J m2; Fish Lake 5.43 J m2 to -2.14 J m2; Lake Wingra 6.42 J m2 to -2.32 J m2 to -2.32 J m2; Fish Lake 5.43 J m2 to -2.14 J m2; Lake Wingra 6.42 J m2 to -2.32 J m2 to -2.32 J m2 to -2.32 J m2 to -2.32 J m2 to -2.34 J m2 to -2.34
- 2.70 J m2) and 1996 (Lake Mendota -2.32 J m2 to 5.22 J m2; Fish Lake -2.14 J m2 to 4.77 J m2; Lake Wingra -2.70 J m2 to 6.20 J m2) for all three lakes. Abrupt changes in latent heat flux appear to be independent of lake morphometry, suggesting that the timing of change may be primarily driven by climate. Net heat flux shows no significant abrupt change for Lake Mendota, two changes for Fish Lake (-0.33 J m2 to 4.00 J m2 after 1964, *p* <0.01, and 4.00 J m2 to -0.55 J m2 after 1975, *p* <0.01), and one change for Lake Wingra (-1.15 J m2 to 0.28 J m2 after 1930, *p* <0.01). Differences in magnitude and timing of abrupt changes in shortwave, longwave, and net heat fluxes emphasize that morphometry may play a role, it is unclear how</li>
- or what the specific role may be.

**3.64.2 Coherence between lake pairs among lakes**

Pearson correlations for all variables and lake pairs are significant (Table 5). Shortwave, longwave, sensible, and latent heat fluxes show high correlation for lake pairs, suggesting that morphometry has little impact on variability among lakes. Similarly,
epilimnion temperatures have high temporal coherence. However, Fish Lake pairs have lower correlations, which may be a result of changes to lake depth (Krohelski et al. 2002) compared to stable water levels in Mendota and Wingra. Low coherence between the Mendota/Fish pair for hypolimnion temperature and stratification dates suggest that fetch differences impact variability. Stability, however, is lower for pairs with Lake Wingra, indicating that lake depth plays a role in temporal coherence of stability. Similarly, Lake Wingra pairs have lower coherence of net heat flux although the coherence of heat flux

17

**Commented [M26]:** How lake morphometry can affect the shortwave flux of solar radiation??

**Commented [MRM27R26]:** The shortwave flux is the net flux at the surface of each lake, which is controlled in part by albedo of the surface water and whether the lake is ice or snow covered, so each lake may have slightly different net shortwave radiative flux for each day and average for the year. components is relatively high. Depth may be influencing a non-linear response of net heat flux that is not present in the components of the flux.

Epilimnetic temperature exhibited high coherence for the three lake pairs (Table 4), suggesting that inter-annual variability in epilimnion temperatures was primarily driven by climate drivers such as air temperature and wind speed. Specifically, the

- 5 Mendota/Wingra pair has the highest correlation and Mendota and Wingra differ significantly in both depth and surface area. Furthermore, comparing the Mendota/Fish pair with similar depth and the Fish/Wingra pair with similar surface area suggests that both surface area and depth impact coherence between lake pairs; and surface area differences may drive asynchronous patterns to a greater extent than does depth differences for epilimnetic water temperature. The lower correlation for the Mendota/Fish and Fish/Wingra pairs of lakes may be due to the difference in abrupt changes for Fish Lake epilimnion
- 10 temperature in comparison to the other two lakes. Likely, the large change in lake depth from the period 1966–2001 (Krohelski et al., 2002) may be impacting the coherence between Fish Lake and the other two lakes, which have had relatively little yearto year variation in water levels over the study period.

Hypolimnion temperature, different from epilimnion temperature, showed only moderate coherence for the Lake Mendota and

- 15 Fish Lake pair (Table 4), suggesting that inter-annual variability in hypolimnion water temperatures was driven in part by factors other than climate, such as lake morphometry. For example, differences in thermocline depth (~10 m in Lake Mendota and ~6 m in Fish Lake) can play a role in filtering the climate signals into the hypolimnion temperature. This result is consistent with other studies that show lake morphometry parameters affect the time of climatic signals, especially temperature stored in the lake system (Thompson et al., 2005). Other factors like strength of stratification and fetch differences may drive differences
- 20 in the timing of stratification, further affecting hypolimnetic temperatures. Moreover, Arvola (2009) showed that hypolimnia temperatures were primarily determined by the conditions that pertained during the previous spring turnover. In our study, the relatively low hypolimnetic coherence (Table 4) suggests that lake morphometry plays a role in hypolimnion temperatures. Coherence for stratification onset and fall overturn dates were low for the Mendota/Fish Lake pair (Table 4), suggesting that surface area, not air temperature or wind speed, was the main factor driving differences in stratification onset and overturn.
- 25 Schmidt stability showed high coherence for the Mendota/Fish lake pair, but low coherence between the Wingra/Fish and Mendota/Wingra lake pairs, suggesting that lake depth drives differences in coherence, while surface area has a lesser role. High coherence between the Mendota/Fish pair suggests that climate drives stability when comparing lakes of similar depth. Low coherence between the other two pairs suggests that lakes with different depths may have asynchronous behavior. Slightly lower coherence for the Mendota/Wingra pair than the Wingra/Fish pair suggests that lake surface area may also play a minor
- 30 role in asynchronous behavior.

**Commented [M28]:** Schmidt stability is irrelevant to nonstratified lakes and cannot be used for comparison

**Commented [MRM29R28]:** Lake Wingra does stratify on daily or weekly timescales during the summer months (Kimura et al, 2016). Summer Schmidt stability was calculated at daily timescales, and then averaged for each year before comparing coherence among the lake pairs. Higher average stability for one year on Lake Wingra would indicate that the lake experienced more days of stratification during the period

18

**3.7 Correlations between lake variables**

Generally, direction and magnitude of Pearson correlation between lake variables are similar for each of the three lakes, however, there are some notable exceptions (Fig. 8). Ice off dates are significantly correlated with stratification onset dates and hypolimnetic temperature on Fish Lake, but those correlations do not exist for Lake Mendota. Stratification onset is

5 significantly correlated with hypolimnetic temperature and stability in Lake Mendota, but not significantly correlated on Fish Lake. Summer air temperatures are more highly correlated with stability than summer wind speed for Lake Mendota and Fish Lake, but the opposite is true for Lake Wingra, where summer air temperature is not significantly correlated. Additionally, hypolimnion temperature is more highly correlated with stability in Lake Mendota, whereas epilimnion temperature is more highly correlated with stability in Fish Lake.

**10 4.33.8 Sensitivity to changes in air temperature and wind speed**

To determine the sensitivity of lake water temperature and stratification in response to air temperature and wind speed, we perturbed these drivers across the range of -10°C to +10°C in 1°C temperature increments and 70% to 130% of the historical value in 5% increments, respectively. For each scenario, meteorological inputs remained the same as for the original simulation and snowfall (rainfall) conversion if the air temperature scenarios increase (decrease) above 0°C. Similarly, the water balance

15 is maintained so that the long-term water levels in both lakes matches the historical record. Responseults of stratification onset, fall overturn, and hypolimnetic temperature to air temperature and wind speed perturbation scenarios for Lake Mendota and Fish Lake are discussed in the following. Other variables are omitted for brevity and Lake Wingra did not experience prolonged stratification under any sensitivity scenarios, so are excluded from the analysis. lake response to all perturbation scenarios will be discussed in the following.

**20 4.3.1 Stratification onset**

Stratification onset generally occurs earlier on Fish Lake than Lake Mendota for all scenarios (Fig. 9). Simulations show that the response of median onset dates to changes in air temperature is linear (-2.0 days °C-1) for Lake Mendota, but for Fish Lake, the change is nonlinear (-1.5 days °C-1 for temperature increases and +2.7 days °C-1 for temperature decreases). Variability in Lake Mendota onset remains consistent, but decreases for Fish Lake as air temperatures increase. This may be from interaction
between ice cover and stratification onset on Fish Lake but not on Lake Mendota. Both lakes have a nonlinear decrease in stratification onset date with decreasing wind speed. For Lake Mendota, the change is -3.4 days (m s-1)-1 for decreases and +10.5 days (m s-1)-1 for wind speed increases. For Fish Lake, the change is -3.6 days (m s-1)-1 for decreases and +8.1 days (m s-1)-1 for wind speed increases. Variability in onset dates decreases with lower wind speeds and increases with higher wind speeds.

19

**Commented [M30]:** Sensitivity runs can be shortened, at least, to a half and moved from 'Discussion' to 'Results'. The actual discussion should be added, considering the reasons for the observed dependencies, their relevance to the processes in real lakes and novelty of the results compared to the state-of-the-art in this area of research.
Fall overturn typically occurs slightly early on Lake Mendota than Fish Lake for all scenarios (Fig. 10). For Lake Mendota, stratification overturn dates change at a rate of +0.68 days  $^{\circ}C^{-1}$  with changes in temperature, while Fish Lake changes nonlinearly at a rate of +1.81 days  $^{\circ}C^{-1}$  for temperature increases and -0.77 days  $^{\circ}C^{-1}$  for temperature decreases from the

- 5 historical condition. Standard deviation in overturn dates decreased slightly for Lake Mendota as air temperature increase, but remains consistent for Fish Lake. Both lakes have nonlinear increases in fall overturn dates with decreasing wind speed. For Lake Mendota, the change is +13.9 days (m s-1)-1 for decreases and -17.1 days (m s-1)-1 for wind speed increases. For Fish Lake, the change is +16.4 days (m s-1)-1 for decreases and -8.5 days (m s-1)-1 for wind speed increases. Like onset dates, variability in overturn dates decreases with lower wind speeds and increases with higher wind speeds.
- 10

For both lakes, increases in air temperature increase hypolimnetic temperatures, while decreases in wind speed decrease temperatures (Fig. 11). Simulations show that the response of median hypolimnetic temperatures to changes in air temperatures is linear for Lake Mendota (+ $0.18^{\circ}C_{hypolimnion} C_{air temperature}^{-1}$ ), but nonlinear for Fish Lake (+ $0.25^{\circ}C_{hypolimnion} C_{air temperature}^{-1}$  for air temperature decreases). Standard deviations under varying air

- 15 temperature scenarios remain consistent for both lakes. Hypolimnion temperatures change non-linearly with wind speed perturbations for both lakes. For Lake Mendota, the change is -1.1°C (m s-1)-1 for decreases and +1.8°C (m s-1)-1 for wind speed increases. For Fish Lake, the change is -1.2°C (m s-1)-1 for decreases and +0.8°C (m s-1)-1 for wind speed increases. Variability decreases for lower wind speeds in Lake Mendota, but remains constant for Fish Lake.
- For both Lake Mendota and Fish Lake (Figure 8a and b), increasing (decreasing) air temperature resulted in carlier (later)
   stratification onset. Lake Mendota exhibited a linear trend of 2.0 days earlier (later) stratification for each degree (C) increase (decrease) in air temperature. Fish Lake, however, shows a nonlinear change in stratification onset with changes in air temperatures of 1.5 days earlier stratification for each degree (C) increase in air temperature but 2.7 days later stratification for each degree (C) each degree (C) decrease in air temperature from the historical condition. Standard deviations in stratification onset on Lake Mendota remained fairly consistent, ranging from 15.5 to 18 days. In contrast, the standard deviation in stratification
- 25 onset for Fish Lake decreased from 17.5 days to 12 days as air temperature increased. This may be due to an early limit in stratification onset for Fish Lake, thus reducing the variability of onset dates with increasing air temperatures. The above results suggest that lake surface area can complicate the response of stratification onset to changes in air temperatures. For both Lake Mendota and Fish Lake (Figure 8c and d), decreased (increased) wind speed results in earlier (later) stratification onset, however the change is nonlinear. For Lake Mendota each 1m s-1-decrease in wind speed results in 3.4 days earlier
- 30 stratification onset and each 1m s+-increase in wind speed results in 10.5 days later stratification onset; meanwhile, Fish Lake shows 3.6 days earlier stratification onset for each 1m s+ decrease in wind speed and 8.1 days later stratification onset for each

**Commented [M31]:** Using wording 'increasing (decreasing)' is clumsy for the reader

**Commented [MRM32R31]:** The authors thank the reviewer for this comment, and we have edited the text to make the writing clearer for readers.

[revised manuscript text omitted]

**Commented [M33]:** See above? Schmidt stability is irrelevant to non-stratified lakes and cannot be used for comparison.

**4 Discussion**

**4.1 Model performance and comparison**

The DYRESM-WQ-ICE model reliably simulated water temperatures over long-term (1911-2014) simulations (Figure 3, Table 3, Table 4). Generally, simulated temperatures were lower than observed values. Some may be attributed to timing of

- 5 observations, which in most instances occur during midday, when water temperatures may be slightly higher than daily averages, as output from the model. Slight deviation is also expected due to averaging of air temperature and wind speeds. Deviation between measurement and observed temperatures was attributed to the input averaging, particularly daily averaging of air temperature and wind speed. In general, thermocline depths were within 1 m of observed values, but some years differ by as much as 2.5 m, contributing additional error in water temperature comparison for depths near the thermocline.
- 10 Discrepancies between modelled and measured values came in part from differences in location and sampling frequency of observations. Errors in water temperature were attributed to differences between simulated and observed thermocline depth over some years. In general, thermocline depths were within 1 m between observed and simulated, but some years differ by as much as 2.5 m.
- 15 Toverall, the performance of the DYRESM-WQ-ICE model was similar towithin those of that of other studies in the literature. Perroud et al. (2009) performed a comparison of one-dimensional lake models on Lake Geneva, and RMSE for water temperatures were as high as 2°C for the Hostetler model (Hostetler and Bartlein, 1990), 1.7°C for DYRESM (Tanentzap et al., 2007), 2°C for SIMSTRAT, and 4°C for Freshwater Lake (FLake) model (Golosov et al., 2007; Kirillin et al., 2012). Similar to this study, errors in the upper layers were lower than those in the bottom of the water column For all four models.
- 20 errors were lower in the upper layers and larger in the bottom of the water column (Perroud et al., 2009)., similar to errors found in this study. Fang and Stefan (1996) gave standard errors of water temperature of 1.37°C for the open water season and 1.07°C for the total simulation period for Thrush Lake, MN, similar to those found hereto those here. Results of Nash-Sutcliff efficiency coefficients for all 3 study lakes were within the ranges found in Yao et al. (2014) for the Simple Lake Model (SIM; Jöhnk et al., 2008), Hostetler (Hostetler and Bartlein, 1990), Minlake (Fang and Stefan, 1996), and General Lake Model (GLM; Hipsey et al., 2014) models for Harp Lake, Ontario, Canada water temperatures.

Model parameters used to characterize the lake hydrodynamics were taken from literature values. These values may be expected to have small variability between lakes; however, previous studies have shown that many of the hydrodynamic parameters are insensitive to changes of ±10% (Tanentzap et al., 2007). Here the model was validated against an independent

30 dataset for each lake to determine if the model fits measured data and functions adequately, with errors within the range of

**Commented [M34]: Lack of deep discussion**

**Commented [M35]:** Effects of lake surface area on the response to the atmospheric forcing are continuously mentioned throughout the manuscript and are among the main subjects of the model sensitivity runs. However, the entire discussion is based on the outputs of a one-dimensional model, i.e. none of the physical processes depending on the horizontal dimensions are modelled directly, but parameterized in the model. Hence, the response of the model outcomes to varying surface area does not necessarily coincide with the response of real lakes to the same perturbations. To analyse properly the modelling results the authors need to (i) present the details on the model parameterizations related to the effects of horizontal advection, wind fetch, horizontally varying depth, and other horizontal processes, such as mixing by internal waves and uwwelling of hynolimmetic.

waters in near-shore areas of the lake; (ii) when discussing the modelling results state clearly which of them can be extrapolated on the real lakes, which horizontal processes

are missed by the model, and how it can affect the real situations; (iii) differentiate be-tween the effects produced by increase of the wind energy input due to larger surface

area from those produced by increase of the thermal inertia due to larger lake volume, like, in particular, timing of the stratification onset (Section 4.3.1)

**Commented [M36]:** The discussion should be rethought, moving the accent from the descriptive listing of the model responses to varying inputs to the discussion on the physical mechanisms producing these responses.

Field Code Changed

**Field Code Changed**

[revised manuscript text omitted]

---

## Referee Report (RR1)

Overall, this paper does not present any breakthrough insights or new ideas; however, it does present a useful analysis of a fairly unique data set – three paired lakes in the same ecoregion for which over 100 years of data are available, and for which weather forcing is similar but differences in lake morphometry are clear. The modeling approach itself is not particularly novel, but, to my knowledge, it has not previously been applied to this rich dataset.  The principal study question regarding the relative influence of morphometry vs. climate forcing is of active interest and is elucidated by application to lakes with such rich data sets.  Accordingly, I recommend publication with minor revisions.

The paper is generally well written and is a logical follow-on to an earlier paper in HESS (Mage et al., 2016, HESS 20:1681, doi:10.5194/hess-20-1681-2016).  There are a few items that may require further attention, as described below; however, the paper is generally acceptable.  Therefore, I recommend publishing subject to minor revisions as described below.

The submitted paper appears to have gone through an extensive and chequered review process.  In the latest iteration, one reviewer was generally favorable and the other rather negative as to publication. The negative review (Reviewer #3) was largely based on critique of the selection of a 1-D model for the comparison.  I do not find this to be a compelling reason to reject the paper.  Indeed, 1-D (vertical) models may be preferable when looking to isolate the general impacts of external forcing on temperature trends.  The potential problems with 1-D models are (1) influence of tributary inflows and (2) representation of mixing due to wind fetch.  Item (1) is not a big issue for these lakes, which are natural lakes with large groundwater inputs.  Item (2) is a potential concern, but it all depends on how well the 1-D model represents wind driven eddy diffusivity. In my opinion, this aspect of the paper would be fully acceptable with the addition of a few lines that address the details of how wind-driven mixing is addressed in the 1-D representation.  The authors' response to Reviewer #3 that 3-D models are computationally expensive is correct, but not sufficient to answer this criticisim.

The three lakes included in this study have somewhat similar perimeter to area ratios, so the major obstacles to the 1-D approach of comparing lakes with very different wind fetch to area ratios will not be encountered here.  The lakes in question are natural, seepage-dominated lakes, so enhanced diffusion due to inflow temperatures should not be a major issue.  The authors should be clear that the results may not apply to individual lakes elsewhere depending on their specific configuration.

Wind-driven mixing is a bigger issue for 1-D models.  Because DYRESM documentation is not readily available (see below) the authors should expand a small amount on this issue.  Specifically, some notes on how wind-driven eddy diffusivity is represented should be supplied, along with any validation data to confirm the 1-D representation.  I am more familiar with Hostetler-based 1-D lake models[12] in which wind-driven eddy diffusivity is estimated as a function of 2 m windspeed, the Brunt-Väsäilä frequency implied by the lake density-gradient, and the Ekman decay as a function of latitude[1]. These 1-D formulations generally require an expression of "enhanced" diffusivity to account for sources of turbulence not represented in the base 1-D formulation.  Some additional explanation of why the 1-D

[1] Hostetler SW, Bartlein PJ.  (1990)  Simulation of lake evaporation with application to modeling lake level variations of Harney-Malheur Lake, Oregon.  *Water Resour Res* 26: 2603-2612, doi:10.1029/WR026i010p02603.
[2] Subin ZM, Riley WJ, Mironov D.  (2012)  An improved lake model for climate simulations: Model structure, evaluation, and sensitivity analyses in CESM1*.  J Adv Model Earth Syst* 4: M02001, doi:10.1029/2011MS000072, 2012

formulation is appropriate would be useful here (for instance, on p. 5 of the current draft). (See also Fang and Stefan[3] for arguments in favor of the 1-D approach.)

On the other hand, 3-D lake models are indeed computationally expensive, and the extra precision does not necessarily lead to greater accuracy. A 3-D model requires data at multiple points in space and time for calibration. When these data are lacking calibration of a 3-D model is often not well constrained and subject to over-fitting. Thus, a 1-D model can be preferable for answering questions about long-term trends and forcing factors.

Additional comments that should be addressed include the following:

- Reviewer #2 correctly noted that humidity, cloud cover, solar radiation all influence lake response, in addition to surface area and air temperature. To this list, precipitation regime could also be added, as large direct precipitation inputs can have a significant impact on stratification stability. The authors added a reasonable discussion of these issues. However, I think the main point that should be added is that the study addresses three lakes in the same ecoregion with similar climate forcing, so differences in responses relate primarily to morphometry or general climate perturbations. Another potentially important factor is water clarity, which determines how solar radiation is vertically partitioned. The authors mention some of these issues in their revision, but should present more discussion. In particular, Table 1 presents Secchi depth as a constant for each of the lakes, but is there any evidence on how this may have changed over time?
- The DYRESM model is a useful formulation. However, it is also somewhat problematic as changes in the Australian scientific establishment have resulted in the deletion of most all links to the DYRESM code and documentation. It is not immediately clear how to obtain the code today. Authors should include a note about the availability of DYREMS – and, if possible, provide a link for access to the DYRESM code as adapted for ice cover.

Line-by-line specific comments:

10. (abstract)    This posits an effect from "decreasing wind speed" – but is decreasing wind speed really a known for these lakes? If it is, it needs to be stated in the abstract.

13 (abstract)    Make clear what is inferred from data vs. from modeling

17. (abstract)    "larger lakes have more variability": This needs to be qualified as to what is mean by "larger lakes". You are comparing three relatively small lakes. The conclusions likely do not apply to Lake Superior.

p. 1, 24. Text states that land and ocean surface temperature anomaliies increased from 1850 to 2012. An anomaly is a departure from an expectation, so you need to state what basis is used for the anomaly assessment.

p.3, 26: "Large surface areas increase the effects of vertical wind mixing…" Isn't it a ratio of wind fetch to depth that is more important here?
* * *
[3] Fang X, Stefan HG. (1996) Development and validation of the water quality model MINLAKE96 with winter data, Project Report 390, St. Anthony Falls Laboratory, Univ. of Minnesota.

p.5,line 28:  Provide a reference for the assumed value of Ksed.

p. 7, line 16:  Text implies that meteorological data are entered into the model on a daily basis.  Is this true?  If so, explain how daily cycles of heating and cooling of the epilimnion and their effect on vertical stability are incorporated into the model.

p. 8, line 24: I agree with the prior reviewers in having some discomfort here about the discussion of adjusting "the minimum water level thickness" as a calibration parameter.  This may be warranted in terms of finding an appropriate minimum thickness that correctly resolves the thermocline position, but needs to be better explained.  If is also unclear why it is appropriate to choose "one minimum layer thickness" for all three lakes.

p. 9, line 25-29: Perturbation tests examine response to changes in air temperature and wind speed.  These tests assume "the water balance is maintained."  This seems unlikely if increased air temperature leads to increase ET.  However, that is the nature of one-parameter perturbation tests – but, the limitation should be acknowledged.

p. 10, line 5: Air temperature "showed a significant change in slope" – based on what test at what significance level?

p. 10, line 15: For NS efficiencies, state the time basis (daily?)

p. 10, line 23: "lake 1980s" should presumably be "late 1980s."

Section 3.5: Presents surface heat fluxes.  Are there any direct measurements to validate these estimates?

Section 3.6: Presents correlations between "lake pairs" for the energy balance.  These are simulated results, right?  If so, how much of the apparent correlation is due to model formulation vs. reality?

p.13, line 1: Discussion of stratification onset: Is this based on observations or modeled results?  If modeled, how does the model compare to observations?

p. 15, line 24: Missing word in "correlations air temperature".

p. 16, line 3: Discusses results of Woolway et al. indicating that influence of air temperature increases was minimal relative to wind speed in determining the length of stratification period.  Isn't this result dependent on the surface area-to-volume ratio of the specific lake in question?  Also, the length of stratification may not be the key result as the onset of stratification may determine the relative temperatures in the epilimnion and hypolimnion over the summer.

p. 17, line 15-20: Discussion of heat fluxes: Is this entirely based on model results?  Is there any observational evidence to confirm magnitude of heat fluxes?

Table 4: Estimated trends need a time dimension – e.g., express in per year metrics.  Also, clarify what test is used to define significance of trend.

Table 5: Clarify that correlation is based on modeled results (?).  If so, how do the modeled results compare to observations?

Figure 3: Results appear "cleaner" for Lake Wingra because it is shallow and more closely tied to air temperature.  For Mendota and Fish Lake is there further insight to be gained by plotting simulated vs. observed temperature by depth range or by season?  Are larger apparent discrepancies caused by mis-estimation of the depth of the thermocline?  Also, put correlation coefficients on the X:Y plots.

---

## Author Response (AR2)

**Response to Reviewer 3**

The authors thank the reviewer for comments regarding this manuscript, and we appreciate the time that the reviewer has taken to carefully review the manuscript. We have addressed the Reviewer's comments and criticisms below.

This paper consists of a sensitivity study of three lakes in the Madison area to changes in meteorological forcing, and compares the responses to these changes among these three lakes utilizing some fairly standard metrics like phenology, max/mean temperatures and stability metrics. I think it could be published, but I'm not entirely convinced of its originality and I have some comments about methodology as well. If the paper is to be revised, they need to do a better job of distinguishing this from the many other papers out there that discuss this sort of sensitivity analysis.

This paper overall looks at two factors (i) the relative influence of wind speeds vs air temperature and (ii) the role of lake size and depth in response of water temperature variables to climate changes. Research investigating the relative importance of other climatic variables in comparison to air temperature are few. Furthermore, to the best of our knowledge, studies that look at changes in non-air temperature climate variables in addition to changes in air temperature do so in either single-lake studies or studies that may encompass a large geographic area. In our study, we investigate lakes that do not undergo identical changes in meteorological variables. We aim to understand how lakes of different size and depth respond to the same changes over a long term. From a management perspective, understanding how lake size and shape alter the lake temperature response is the first step into forecasting changes in chemical and biological components of the lake ecosystem. The key contribution of this manuscript is the investigation into lake morphometry differences under wind speeds and air temperature. To the best of our knowledge, this type of studied for a long historical time period (104 years) have yet reported before. We have edited the introduction, discussion, and conclusion sections of the paper to make this contribution and significance more clear to the readers.

My main methodological concern with this paper is that it attempts to look at the role that lake area has on the response to changes in meteorological conditions, but uses a 1-dimensional model to do so. They include an empirical adjustment to account for lake area, but it seems to me that at this stage, you're not looking at the response dependence on lake area, you're looking at the response dependence on an empirical lake area adjustment. Why not just use a 3D model? Computing is cheap these days.

The authors thank the reviewer for this critical comment. We agree, that there are drawbacks to using a one-dimensional approach, as stated as an empirical lake area adjustment. We respond the comment from the following two perspectives:

From the computing perspective, using a comprehensive 3D models for 3 different lakes over a period of 104 years is still not a cheap task. Based on a paper from Reimer and Wu (2016, Water Resources Management), running a 3D model on Lake Mendota has been developed under a high performance computing (HPC) server based upon Rocks Cluster 6.0, an open-source Linux cluster distribution capable of scalable and parallel computing. The server with AMD Opteron 3.2 GHz Processor and 64 Central Processing Units (CPUs). Model input data is temporarily stored on the server for model simulation. Afterwards, each model simulation runs in parallel mode with 8 CPUs. In the modeling configuration, the time ratio of run time to modeled time is 1:72." Based on this time ratio, running a 104 year simulation on Lake Mendota would take 1.4 years of run time using their 64 CPU set-up. Fish Lake and Lake Wingra

are smaller lakes. So in theory it would take less time to run. Our modeling experience (Reimer and Wu, 2016) suggested that modeling run time of ~3 years would for all three lakes would take tremendous amount of time. Overall, the additional information provided from a 3D model doesn't justify such a significant computational effort.

From data availability perspective, any 3D model simulations requires observation data only collected at the deep hole, but also spatially distributed data across the surface of the lake (Kamarainen et al., 2007, Zhang et al., 2015, and Kimura et al., 2016). In reality, we do not have 3D data for three lakes over for the period 1911-2014. If we decide to conduct this 3D modeling, we should have at least wind direction information. Based upon the best of the authors' knowledge, such data for the duration of that time period is not available. As a result, the authors structured the research using a one-dimensional model to include an empirical adjustment (which has been used on other studies of lakes with varying surface areas and shown to be a valid adjustment) for a 100+ year time period than from using a three-dimensional model for only a 35-year time period. Indeed, this type of long-term study using one-dimensional model is not an easy task. Nevertheless, the outcome of the 1-D modeling provide insights to address the response of water temperatures and stratification in lakes with different morphometry (water depth and surface area) to changing air temperature and wind speed over the period 1911-2014.

Kamarainen, A., Yuan, H.L, Wu, C.H., Carpenter, S.R., 2009. Estimates of phosphorus entrainment in Lake Mendota: A comparison of one-dimensional and three-dimensional approaches, *Limnology and Oceanography: Methods*. 7, 553-567

Kimura, N., Wu, C.H., Hoopes, J.A., and Tai, A. 2016, Diurnal thermal dynamic processes in a small and shallow lake under non-uniform wind and weak stratification, *Journal of Hydraulic Engineering-ASCE*, 142(11), 04016047,

Reimer J.R. and Wu, C.H., 2016. Development and application of a Nowcast and Forecast system tool for planning and managing a river chain of lakes, *Water Resources Management*, 30(4), 1375-1393.

Zhang, Y.J., Ateljevich, E., Yu, H.C., Wu, C.H., Yu, J.C.S., 2015. A new vertical cooridnate system for a 3D unstructured-grid system, *Ocean Modelling*, 85(1), 16-31.

One result I found very strange is their description of the "nonlinear" response of some of the lakes to various perturbations. They state that the response depends on whether they are perturbing, for instance, air temperature in the positive direction or negative direction (p12. Lines 10-15, again lines 20-25). There's nothing special about the base case, so the division they point out is artificial.

Thank you to the reviewer for pointing out that this wording is strange. We reference from the base case since that case encompasses the historical observations of meteorological conditions and the air temperature and wind speed were perturbed from that historical condition. We have rewritten the sections to address, instead of saying "nonlinear", the change caused from an increase or decrease in air temperature or wind speed from this base case.

If they are going to discuss step changes in various parameters, they should probably reference the van Cleave and Lenters paper that does this for Lake Superior.

The Van Cleave et al paper uses a different method for determining step changes, and focuses significantly on ice cover. However, we have referenced the difference in step changes between Lake Superior and our lakes for surface water temperatures as suggested by the reviewer in the following: "These increases in epilimnetic temperatures are similar to those found for European lakes (Woolway et al., 2017b) in response to regional climate changes, although Woolway et al. (2017b) demonstrated a substantial increase in annually averaged lake surface water temperatures in the lake 1980s in response to an abrupt shift in the climate, which is not apparent in epilimnion water temperatures for our study lakes. Additionally, Van Cleave et al. (2014) showed a regime shift in July – September Lake Superior surface water temperatures after 1997, driven by El Niño in 1997-1998; however we do not find a similar regime shift in our study lakes, which may be due to geographical differences in meteorology or morphometric differences from the larger Lake Superior."

On a more minor note, they should specify what bulk turbulence method they are using in the model.

Thank you to the reviewer for pointing out that this wasn't clear in the manuscript. We have revised the model description as "Surface layer mixing is based on potential energy required for mixing, and introduction of turbulent kinetic energy through convective mixing, wind stirring, and shear mixing (Imerito, 2010; Yeates and Imberger, 2003). Layer mixing occurs when the turbulent kinetic energy, stored in the topmost layers, exceeds a potential energy threshold (Yeates and Imberger, 2003)."

I was confused by the statement "all parameters and coefficients are kept constant" (p. 5 line 26). What parameters are these?

Thank the reviewer for pointing out this confusion. The parameters referred to are the model parameters and coefficients – they are constant throughout the simulation rather than being time-varying. Some confusion may be from this sentences placement within the paragraph. We have moved the sentence later in the paragraph. It now follows the sentences which specify the model parameters. "…Parameters relevant to the open water period are provided in Table 2. Ice cover model parameters can be found in Hamilton et al. (in review), Magee and Wu (2016), and Magee et al., (2016). During the entire simulation period, all model parameters and coefficients are kept constant. …"

Do they use the same sediment temperature model for all three lakes? Is this justified?

Yes, we use the same model to estimate heat transfer from the sediments to the water column (eq 1 and 2), and the same equation to estimate sediment temperature (eq 3) for all three lakes. Given the distance between lakes, it is likely justified to assume that over the long-term modeling period, sediment temperature changes at all three lakes follow a very similar sinusoidal pattern during the year. Based on data from Hennings and Connelly (Hennings, R. G. and Connelly, J. P.: Average ground-water temperature

map, Wisconsin, Wisconsin Geological and Natural History Survey., 2008.), groundwater temperatures at all three lakes have the same average and vary similarly throughout the year. At these depths, groundwater temperatures are an appropriate proxy for sediment temperatures. Since groundwater temperatures have the same average and vary similarly throughout the year at all three locations, we feel that although the equation for sediment temperature was developed for Lake Mendota, it is justified to use it for all three of the lakes in this study.

**Response to Reviewer 4**

The authors thank Reviewer for their helpful and constructive comments. We have addressed the comments to improve the quality of the manuscript. We have addressed comments in the reply provided below.

**General comments**

This is a well written paper describing an interesting topic in climatology and limnology: 'How has lake temperature and stratification responded to a changing climate, and how can we expect these responses to differ among lakes'. Magee and Wu use an exceptional dataset of >100 years and focus on the effects of changing air temperature and wind speed on water temperature and stratification patterns in three lakes situated near Madison, Wisconsin (US), which are characterized by different morphometric features. Similar results have been presented in the literature for lakes elsewhere (see specific details below), but the strengths of this paper is the across-lake comparison and the use of a one-dimensional lake model to disentangle the different factors which contribute to a lakes response to climate change. The authors use the one-dimensional lake model (DYRESM-WQ), which has been used frequently by the limnological community during the past 20 or so years. I must admit, I am not a fan of using models that are not openly available (this is my understanding of DYRESM), especially as many other one-dimensional lake temperature models are openly available to the community. For example, the General Lake Model (GLM), Freshwater Lake model (FLake), and MyLake all have similar capabilities to DYRESM but are open-access. If my understanding is incorrect and DYRESM is openly available, please state this in the methods. The authors use DYRESM to reconstruct the thermal and stratification regime of the lakes during the last century and for sensitivity studies exploring the lake responses to changes in mean annual air temperature and wind speed.

I found the aim of this paper clearly described, the results very convincing and presented in a reasonable manner, and I think the paper will be well received by the limnological community. Specifically, this paper investigates what, in my opinion, is currently lacking in the scientific literature. Specifically, many of the previous studies that have examined the response of lake surface water temperature to climate change have focused on the past few decades. Thus, having a study focusing on lake temperature responses from the start of the 20th century is an important contribution. I do believe this is an important topic, and one that deserves some attention - particularly given the recent emphasis on the rapid warming of lakes from around the world, and the numerous ecological and socioeconomic consequences of increasing lake surface water temperature, in addition to their interactions within the climate system.

Overall, I think this paper is well written, the data is well chosen and the analysis is reasonable. I can see this paper being a benchmark for future studies and an important contribution to the literature. I think there is great potential for this paper to be valuable to the community.

We thank the reviewer for the kind comments. We appreciate that the reviewer thinks the paper would be useful or valuable for the limnology community. Regarding the accessibility of DYRESM, the authors agree that this is valid concern. While this paper deals specifically with open water temperature comparisons across the three lakes, the authors have been working for a number of years on additional analysis in the three study lakes dealing with also ice cover (see Magee and Wu, Hydrological Processes, 2016), water clarity (see Magee et al, HESS, 2016), and dissolved oxygen and fish habitat (ongoing research). Because of the long-term goal, we select one model that was capable of simulating not only

temperature and dissolved oxygen, but also the biogeochemical processes in the lakes. When the time modeling work was originally initiated, such open source models were not available. Flake doesn't incorporate biogeochemical processes, and MyLake has some issues in regard to the number of algal groups incorporated and biogeochemical processes incorporated into the model when compared to those in DYRESM-WQ. For those reasons, FLake and MyLake were not chosen. Additionally, at the onset of the study, the GLM model, while available for use, was not at the time able to simulate ice cover accurately (see Yao et al, Hydrological Processes, 2014). At the time of project initiation, DYRESM-WQ was superior to the available open source models in terms of meeting all the goals of our overall research agenda with only one model development. However, the authors do appreciate the reviewer's comments regarding open source modeling and we will, of course, keep these comments in mind when moving forward with future research questions.

**Specific Comments**

Introduction

In general, the introduction is well written. However, the literature review seems rather limited and many of the recent studies covering this topic have been overlooked. For example, see the O'Reilly et al. (2015) paper and references therein for an update of some recent paper in this topic. Also, in the opening paragraph, the authors refer to decreasing wind speeds but not referred to some of the important papers in this topic, such as Vautard et al. (2010) who demonstrated that wind speeds globally have been decreasing. In addition, of great relevance to the current study is the paper by Woolway et al. (2017a) who followed a very similar approach to that described in this paper when reconstructing the thermal dynamics of Vortsjarv (Estonia) - they also investigated the response of lake temperature dynamics to changes in air temperature and wind speed.

The authors thank the reviewer for this comment, and we agree, that some of the most recently published literature was left out of the introduction and discussion of the manuscript. The original manuscript was submitted to Hydrology and Earth Systems Sciences in May 2016, which left a large bulk of the most recent literature (published in 2016 and 2017) outside of the paper. Taking the reviewer's comment into consideration, we have added some of the most pertinent newly published literature within the introduction and discussion where appropriate and necessary to place this research within the broader context of lake temperature investigations. To reduce length and give appropriate credit to a broad variety of research and groups, we have not included the exhaustive list of most recent literature, but rather those we feel adds the most value to our manuscript.

P2L15 - Lake water temperature is closely related to the... Equally important is humidity, cloud cover, solar radiation - in particular the surface energy fluxes. For example, the study of Schmid and Köster (2016) demonstrated that 60% of lake surface water temperature warming in Lake Zurich was caused by air temperature and 40% by increased solar radiation. Also, a paper by Wilhelm et al. (2006) demonstrated that daily extreme water temperatures (although they used the equilibrium temperature) responded to shifts in air temperature, wind speed, relative humidity, and cloud cover. Thus, one could argue that to understand fully how lake surface water temperatures will respond to climate change, one would require each of these variables to be included. This has been shown to be important for some lakes (Schmid and

Köster 2016). I realize that the authors have likely considered all of the points I've raised here, but I think they should be mentioned in the paper.

Also, important is local features. For example, the study Tanentzap et al. (2008) showed that some lakes might cool as air temperatures increase, as a response of the complicated interactions between lakes and their environment and/or internal processes.

The authors thank the reviewer for the comment. We have added a section in the discussion to address these points concerning changes in other climate variables and local lake-specific interactions, as follows: "Ultimately, lake warming or cooling may depend on the magnitudes and directions of changes of air temperature, wind speed, and other variables as climatic variables humidity, cloud cover, and solar radiation and water clarity variables are important in determining lake water temperatures. Schmid and Köster (2016) demonstrated that 40% of surface water warming in Lake Zurich was caused by increased solar radiation. Wilhelm et al. (2006) showed that daily extrema of surface equilibrium temperature responded to shifts in wind speed, relative humidity, and cloud cover in addition to changes in air temperature. However, neither study looked at lakes with seasonal ice cover, which may not account for changes in ice sheet formation and the resulting influence on lake water temperatures (Austin and Colman, 2007). Changes in underwater light conditions from increased dissolved organic carbon concentrations combined with reduction in surface wind speeds can result in cooling whole-lake average temperatures despite substantial air temperature increases, as was the case for Clearwater Lake, Canada (Tanentzap et al., 2008). Water clarity has seen both increases and decreases since the early 1990's (Rose et al., 2017), with precipitation playing a critical role in year-to-year variability (Rose et al., 2017). Further investigation into the combined effects of these climatic and lake-specific variables is warranted."

…found that decreasing wind speeds resulted in increased stratification… - This is a great example of the Woolway et al. (2017a) study.

As stated above, this paper was not published when our manuscript was originally submitted, but the authors thank the Reviewer for pointing out its relationship to our study and we have added this reference as suggested.

In terms of the influence of lake surface area altering the response of lakes to atmospheric forcing, I think more context is needed here. Specifically, numerous papers have demonstrated the importance of lake size, some of which you already cited. Some examples:

• Winslow et al. (2015), which you cite, demonstrated that small lakes demonstrate a muted response of deep water temperature to climate change.
• Read et al. (2012) demonstrated that lake size influences the relative contribution of wind and convective mixing to the gas transfer coefficient in lakes.
• Woolway et al. (2016) demonstrated that lake size can influence the magnitude of diurnal heating and cooling in lakes which has important consequences for gas transfer (Holgerson et al. 2017).
• Torbick et al. (2016) demonstrates that smaller lakes in northeast United States have been warming more rapidly than larger lakes in terms of surface water temperature.
Note I only list a few above but there are many others, which I trust the authors to find.

The authors thank the Reviewer for this list of suggested literatures, we have added additional context where suggested in the introduction and discussion. Specifically, the last two bullet points address research that was published after our manuscript was submitted to Hydrology and Earth Systems Sciences. We were aware of the new research but those paper were published after the submission of the original manuscript. We have added these new studies as suggested by the reviewer,

Methods

Is DYRESM-WQ open-access? If so, I think details of where the reader can find the source code would be useful. Similar to GLM (General Lake Model), which is openly available, I would hope DYRESM-WQ is available to the community, as all results should be reproducible by the reader.

DYRESM-WQ is not open access. As discussed above, GLM was only in the early stages of development when the modeling development and setup was first started. It was not an option for us to use in our study. Only recently has GLM successfully reproduced ice and snow cover, which was an important component of the continuous modeling and other parts of the overall study (see Magee and Wu, Hydrological Processes, 2016). Additionally, other available open-source models were not chosen because of deficiencies in simulating biogeochemical processes and dissolved oxygen, as described previously.

I would show the light attenuation coefficient as Kd (not k), as it is commonly referred in the limnological community.

The authors thank the reviewer for this comment. We have shown light attenuation coefficient at Kd, as suggested by the reviewer.

Discussion

This paper has some similarities with a paper very recently published in Climatic Change by Woolway et al. (2017b). In particular, they also analyze a > 100 year lake temperature time series from two lakes in Austria (Mondsee and Worthersee), and investigate how 20 lakes with >50 years of observations have responded to climatic warming. I would strongly encourage the authors to discuss these results and compare to their findings.

In addition, how do the abrupt shifts, which you discuss (evaluated via the Rodionov method), compare with Woolway et al. (2017b)?

The authors thank the Reviewer for these two comments. While epilimnion water temperatures are a part of this manuscript, they are not the focal point as in Woolway et al. To address the comments, we have added some sentences in the results placing our epilimnion temperature changes in the context of this work by Woolway et al in European lakes: " These increases in epilimnetic temperatures are similar to those found for European lakes (Woolway et al., 2017b) in response to regional climate changes, although Woolway et al. (2017b) demonstrated a substantial increase in annually averaged lake surface water temperatures in the lake 1980s in response to an abrupt shift in the climate, which is not apparent in epilimnion water temperatures for our study lakes."

Figures and Tables

Figure 1: Can the authors add a local map to illustrate how close these lakes are to one-another? I know this is mentioned in the text, but including a map would be useful. Also, it is rather difficult to see some of the contour values. If the contours can be re-drawn in grey, that would make the writing more understandable – or potentially redraw these as colour plot, if you are generating figures in colour.

The authors thank the Reviewer for this comment. We have added a local map with scale, as suggested to aid the reader in interpreting the distance between all three lakes. Additionally, we have re-drawn contours in grey to make the figures easier to interpret for the reader.

Figure 2. Interestingly, the step in annual air temperature seems to occur at the same time as reported by Woolway et al. (2017b) for Central Europe, a result of a global shift perhaps? Also, the wind speed change in ~1995 is consistent with the results of Woolway et al. (2017a) for Estonia, although they do not report on the exact year, but a shift can be seen from their figures.

The authors thank the Reviewer for this comment, we have added a sentence addressing the shift in annual air temperatures "A change in air temperatures also occurred in the 1980s in Central Europe (Woolway et al., 2017b), which may indicate that change in air temperature were a global phenomenon rather than local occurrence."

The authors thank the Reviewer for this extensive list of additional references, some of which were published after the submission of the manuscript to Hydrology and Earth Systems Sciences. We have elected to include the most pertinent new references and a few others as appropriate for enhancing introduction and discussion of the manuscript.

[revised manuscript text omitted]

---

## Author Response (AR3)

**Response to the Reviewer**

Overall, this paper does not present any breakthrough insights or new ideas; however, it does present a useful analysis of a fairly unique data set – three paired lakes in the same ecoregion for which over 100 years of data are available, and for which weather forcing is similar but differences in lake morphometry are clear. The modeling approach itself is not particularly novel, but, to my knowledge, it has not previously been applied to this rich dataset. The principal study question regarding the relative influence of morphometry vs. climate forcing is of active interest and is elucidated by application to lakes with such rich data sets. Accordingly, I recommend publication with minor revisions.

The paper is generally well written and is a logical follow-on to an earlier paper in HESS (Mage et al., 2016, HESS 20:1681, doi:10.5194/hess-20-1681-2016). There are a few items that may require further attention, as described below; however, the paper is generally acceptable. Therefore, I recommend publishing subject to minor revisions as described below.

Thank the Reviewer for taking valuable time to provide constructive and thoughtful comments on the manuscript. We have addressed the comments in a point-by-point reply below.

The submitted paper appears to have gone through an extensive and chequered review process. In the latest iteration, one reviewer was generally favorable and the other rather negative as to publication. The negative review (Reviewer #3) was largely based on critique of the selection of a 1-D model for the comparison. I do not find this to be a compelling reason to reject the paper. Indeed, 1-D (vertical) models may be preferable when looking to isolate the general impacts of external forcing on temperature trends. The potential problems with 1-D models are (1) influence of tributary inflows and (2) representation of mixing due to wind fetch. Item (1) is not a big issue for these lakes, which are natural lakes with large groundwater inputs. Item (2) is a potential concern, but it all depends on how well the 1-D model represents wind driven eddy diffusivity. In my opinion, this aspect of the paper would be fully acceptable with the addition of a few lines that address the details of how wind-driven mixing is addressed in the 1-D representation. The authors' response to Reviewer #3 that 3-D models are computationally expensive is correct, but not sufficient to answer this criticism.

We agree with the reviewer that the influence of tributary inflows is minimal for these lakes. Lake Mendota is the only lake with inflow water volume that is small compared to the overall lake volume. We have added some additional text on page 5 to further detail how the model represents mixing due to wind. Since all equations within the model are presented elsewhere (see Imerito, 2010; Yeates and Imberger, 2003), these equations are not reproduced and re-written on our manuscript.

The three lakes included in this study have somewhat similar perimeter to area ratios, so the major obstacles to the 1-D approach of comparing lakes with very different wind fetch to area ratios will not be encountered here. The lakes in question are natural, seepage-dominated lakes, so enhanced diffusion due to inflow temperatures should not be a major issue. The authors should be clear that the results may not apply to individual lakes elsewhere depending on their specific configuration.

Thank you to the reviewer for identifying these important points. We have added text to address this in the discussion section of the manuscript on page 16.

Wind-driven mixing is a bigger issue for 1-D models.  Because DYRESM documentation is not readily available (see below) the authors should expand a small amount on this issue.  Specifically, some notes on how wind-driven eddy diffusivity is represented should be supplied, along with any validation data to confirm the 1-D representation.  I am more familiar with Hostetler-based 1-D lake models[1,2] in which wind-driven eddy diffusivity is estimated as a function of 2 m windspeed, the Brunt-Väsäilä frequency implied by the lake density-gradient, and the Ekman decay as a function of latitude[1]. These 1-D formulations generally require an expression of "enhanced" diffusivity to account for sources of turbulence not represented in the base 1-D formulation.   Some additional explanation of why the 1-D Formulation is appropriate would be useful here (for instance, on p. 5 of the current draft).  (See also Fang and Stefan[3] for arguments in favor of the 1-D approach.)

In response to the reviewer's suggestion, we have added additional text to further describe how the model deals with wind-driven mixing. (Yeates and Imberger (2003) previously detailed the description of the model equations used in this version of DYRESM as well as validation data from five different lakes of varying size and wind forcing. As a result, we will refer to the excellent paper and other references and not reproduce this work on our manuscript.  Additionally, we have addressed why the one-dimensional model is appropriate with the discussion section of the manuscript (see pages 15 and 16).

On the other hand, 3-D lake models are indeed computationally expensive, and the extra precision does not necessarily lead to greater accuracy.  A 3-D model requires data at multiple points in space and time for calibration.  When these data are lacking calibration of a 3-D model is often not well constrained and subject to over-fitting.  Thus, a 1-D model can be preferable for answering questions about long-term trends and forcing factors.

We agree with the Reviewer that a one-dimensional lake model for our study is preferable to answer our science questions of interest. Thanks for the Reviewer's comment and support.

Additional comments that should be addressed include the following:
• Reviewer #2 correctly noted that humidity, cloud cover, solar radiation all influence lake response, in addition to surface area and air temperature.  To this list, precipitation regime could also be added, as large direct precipitation inputs can have a significant impact on stratification stability.  The authors added a reasonable discussion of these issues.  However, I think the main point that should be added is that the study addresses three lakes in the same ecoregion with similar climate forcing, so differences in responses relate primarily to morphometry or general climate perturbations.  Another potentially important factor is water clarity, which determines how solar radiation is vertically partitioned.  The authors mention some of these issues in their revision, but should present more discussion.  In particular, Table 1 presents Secchi depth as a constant for each of the lakes, but is there any evidence on how this may have changed over time?

[1] Hostetler SW, Bartlein PJ.  (1990)  Simulation of lake evaporation with application to modeling lake level variations of Harney-Malheur Lake, Oregon.  Water Resour Res 26: 2603-2612, doi:10.1029/WR026i010p02603.
[2] Subin ZM, Riley WJ, Mironov D.  (2012)  An improved lake model for climate simulations: Model structure, evaluation, and sensitivity analyses in CESM1.  J Adv Model Earth Syst 4: M02001, doi:10.1029/2011MS000072, 2012
[3] Fang X, Stefan HG.  (1996)  Development and validation of the water quality model MINLAKE96 with winter data, Project Report 390, St. Anthony Falls Laboratory, Univ. of Minnesota

We have added a sentence on page 3 to clearly state that we are looking at lakes in the same region with the same climate forcing to differentiate responses that are primarily a result of morphometry differences. In the model, we use seasonal Secchi depth averages for each year. Therefore, Secchi depths in the model do change over time. Table 1 lists the lake-average value of the study period is presented in Table 1. Data from NTL-LTER (NTL LTER, 2012a) do not show a significant long-term trend in yearly averaged open-water Secchi depth values for any of the three lakes. We have added text within Section 2.3.3 to address these minor concerns about Secchi depth changes with time.

• The DYRESM model is a useful formulation. However, it is also somewhat problematic as changes in the Australian scientific establishment have resulted in the deletion of most all links to the DYRESM code and documentation. It is not immediately clear how to obtain the code today. Authors should include a note about the availability of DYREMS – and, if possible, provide a link for access to the DYRESM code as adapted for ice cover.

To address the Reviewer's question regarding the availability to access the DYRESM code, we add the statement at the acknowledgement section. Readers who are interested in accessing the DRYESM code should contact the authors: Chin Wu (Email: chinwu@engr.wisc.edu) or Madeline Magee (Email: mrmagee@wisc.edu).

**Line-by-line specific comments:**

10. (abstract) This posits an effect from "decreasing wind speed" – but is decreasing wind speed really a known for these lakes? If it is, it needs to be stated in the abstract.

Wind speed in Madison, WI, has been decreasing over the last 100 years. We have edited the text to note that increases in air temperature and decreases in wind speed have been observed.

13 (abstract) Make clear what is inferred from data vs. from modeling

We have edited the sentence to make clear that results of increases and decreases are from modeling results.

17. (abstract) "larger lakes have more variability": This needs to be qualified as to what is mean by "larger lakes". You are comparing three relatively small lakes. The conclusions likely do not apply to Lake Superior.

We have edited the text to clarify that the results apply to the lakes within this study.

p. 1, 24. Text states that land and ocean surface temperature anomalies increased from 1850 to 2012. An anomaly is a departure from an expectation, so you need to state what basis is used for the anomaly assessment.

The basis is 1961-1990, and we have added this to the text.

p.3, 6: "Large surface areas increase the effects of vertical wind mixing…" Isn't it a ratio of wind fetch to depth that is more important here?

Rueda and Schladow (2009) used simulations and scaling arguments to investigate the effects of a decrease in the surface area of the Salton Sea. Mixing in general is stronger in larger basins due to greater

mechanical energy fluxes across the free surface, regardless of the depth of the lake. Addition of greater mixing fluxes across the free surface will increase as surface area increases regardless of the depth of the lake(s). We make no claims as to whether the ratio of wind fetch to depth or surface area itself is more important to determining long term water temperature differences. Only that lakes with larger surface areas have greater mechanical energy fluxes than lakes with smaller surface areas.

p.5,line 28:  Provide a reference for the assumed value of Ksed.
The reference has been added.

p. 7, line 16:  Text implies that meteorological data are entered into the model on a daily basis.  Is this true?  If so, explain how daily cycles of heating and cooling of the epilimnion and their effect on vertical stability are incorporated into the model.
As described in section 2.2, water temperature, surface heat fluxes, water budget, and ice thickness are all calculated at 1 hour timesteps, so daily cycles of heating and cooling are incorporated into the model at these time steps. However, inputs of cloud cover, air pressure, wind speed, and air temperature are assumed to be constant values throughout the day. We are limited to the daily constant values by the resolution of meteorological observations, which are not available at an hourly resolution for the full duration of the 100-year study period. Similarly, precipitation is also assumed to be uniformly distributed throughout the day. However, shortwave radiation, which is a critical driver for determining diurnal changes in epilimnion temperatures are assumed to have a sinusoidal distribution throughout the day. This distribution is computed based on lake latitude and the Julian day.

p. 8, line 24: I agree with the prior reviewers in having some discomfort here about the discussion of adjusting "the minimum water level thickness" as a calibration parameter.  This may be warranted in terms of finding an appropriate minimum thickness that correctly resolves the thermocline position, but needs to be better explained.  If is also unclear why it is appropriate to choose "one minimum layer thickness" for all three lakes.
Model predictions are driven in part by the thickness of the horizontal layers to detect changes in vertical density stratification (Imberger et al., 1978; Tanentzap et al., 2007), which needs to be sufficiently small to properly resolve the changes in density and water temperature that occur at the thermocline, as the reviewer has correctly stated. If a layer becomes smaller than the minimum water layer thickness during any time step, then it is merged with the smaller of the layer above or below, taking on the characteristics of that layer. If the minimum layer thickness is too large, then the resolution around the thermocline is not enough to appropriately calculate changes in density and temperature that occur over a very small vertical distance. We have added text in the model calibration section to make this clearer to the readers. We choose to use one layer thickness for all three lakes so that differences in water temperature, heat fluxes, and stratification among the three lakes are limited to differences in morphometry of the lakes rather than differences in model implementation.

p. 9, line 25-29: Perturbation tests examine response to changes in air temperature and wind speed. These tests assume "the water balance is maintained."  This seems unlikely if increased air temperature leads to increase ET.  However, that is the nature of one-parameter perturbation tests – but, the limitation should be acknowledged.

Thank you for this comment. We have added a sentence to point out (acknowledge) this limitation of the study. Our approach would exclude changes in water level (and as a result water temperature and stratification) based on ET changes with altered air temperatures and wind speeds as well as differences in precipitation under future climates. It also neglects effects of climate changes on water clarity, which may be observed as changes in climate drivers will alter turbidity of the lakes and algal population characteristics and overall biomass.

p. 10, line 5: Air temperature "showed a significant change in slope" – based on what test at what significance level?
This was based on breakpoint analysis with an *F* test to show significance at a threshold of 0.05, as in (Magee et al., 2016). Detail of this and the appropriate reference has been added to the text.

p. 10, line 15: For NS efficiencies, state the time basis (daily?)
Yes, we used a daily timescale for NS efficiencies. This has been noted in the text.

p. 10, line 23: "lake 1980s" should presumably be "late 1980s."
This has been changed in the text. We thank the reviewer for pointing out this typo which went unnoticed.

Section 3.5: Presents surface heat fluxes. Are there any direct measurements to validate these estimates?
No, there are no direct measurements to validate the surface heat flux estimates. However, equations used to derive the model surface heat fluxes are empirical equations based on direct measurements. Heat fluxes are included in this manuscript at the request of a reviewer during a previous iteration to use the full capability of the model to present useful and worthwhile results. We feel that, while the model is not suitable to put a specific value on hindcast or forecast heat fluxes for any one year with a high degree of certainty, it is suitable for informing general differences among the three lakes over the long term. This is how we have presented surface heat flux results within the text of this manuscript.

Section 3.6: Presents correlations between "lake pairs" for the energy balance. These are simulated results, right? If so, how much of the apparent correlation is due to model formulation vs. reality?
Yes, the correlations for the energy balance is for simulated results. Since we have no direct measurements of heat fluxes on these three lakes to calculate non-modeled correlations, we are unable to determine how much of the correlation is due to model formulation vs. reality. Nevertheless, calculations for the heat fluxes within the model employ the same equations for each lake and meteorological drivers for all three lakes are identical. This approach likely yields correlations that are higher than would be observed because we are excluding local differences among the lakes that may impact surface heat fluxes such as differences in wind sheltering and shading; however, differences in heat flux caused by differential water temperatures among the three lakes as a result of the different lake shapes is included.

p.13, line 1: Discussion of stratification onset: Is this based on observations or modeled results? If modeled, how does the model compare to observations?
This section is based on modeled results of the two lakes under temperature perturbation scenarios. We have added the word "simulated" to the first sentence of section 3.8 to make it clearer that the results

from this section are for simulated results after perturbing air temperature and wind speed drivers of the model. Ideally, we would compare historical stratification onset dates on years that have similar air temperature and wind speed as those of the perturbed scenarios. Air temperature and wind speed perturbations outside of +/- 2°C and +/- 10% of the historical condition are not captured extensively in the historical dataset. This limits us to only a minor set with which to compare how well the model might perform under the perturbed scenarios. Unfortunately, there are no known observations of stratification onset for any of the three lakes as water temperatures observations are generally taken at the period of 2-week intervals over the period. We can say that in every year under the historical simulation, the model hindcasts stratification onset within the appropriate time window based on observed water temperatures. We feel this indicates the model performs well in terms of correctly hindcasting stratification onset during the historical condition and likely does perform well in forecasting changes to stratification onset as air temperature and wind speed drivers are perturbed within the model.

p. 15, line 24: Missing word in "correlations air temperature".
Thank you, the line has been changed to say "…correlations between air temperature and lake temperature variables…"

p. 16, line 3: Discusses results of Woolway et al. indicating that influence of air temperature increases was minimal relative to wind speed in determining the length of stratification period.  Isn't this result dependent on the surface area-to-volume ratio of the specific lake in question?  Also, the length of stratification may not be the key result as the onset of stratification may determine the relative temperatures in the epilimnion and hypolimnion over the summer.
Yes, the effect of air temperature vs wind speed to determine stratification and water temperature are likely to be influenced by the size and shape of the lake(s) in question. We include this reference as well as others (Kerimoglu and Rinke, 2013, Hadley et al., 2014), suggested by the previous reviewer to point out other studies in which wind speed was found to be at least as important as air temperature to changes in stratification and water temperature. The lake studied by Woolway et al. (2017) is a polymictic lake. Specifically, their study looked at the number of stratification days rather than the onset of stratification, where they found that wind speed was an important driver for number of days of stratification. On page 16, lines 10-13, we discuss how temperature in the hypolimnion over the summer are driven by conditions during spring turnover and timing of stratification onset. We have revised texts in this section to cite (Woolway et al., 2017) in the context of results of our manuscript.

p. 17, line 15-20: Discussion of heat fluxes: Is this entirely based on model results?  Is there any observational evidence to confirm magnitude of heat fluxes?
Yes, the heat fluxes are based on model results. Unfortunately, there is no observational evidence regarding magnitude of the heat fluxes on these three lakes that we are aware of.

Table 4: Estimated trends need a time dimension – e.g., express in per year metrics.  Also, clarify what test is used to define significance of trend.
As stated in the table heading, the time dimension for trends are in decades. i.e. each trend in in units per decade. We used a t-test to define significance of trend, and we have added an annotation to indicate this.

Table 5: Clarify that correlation is based on modeled results (?). If so, how do the modeled results compare to observations?
We have edited the table heading to indicate that the correlations are based on modeled results.

Figure 3: Results appear "cleaner" for Lake Wingra because it is shallow and more closely tied to air temperature. For Mendota and Fish Lake is there further insight to be gained by plotting simulated vs. observed temperature by depth range or by season? Are larger apparent discrepancies caused by mis-estimation of the depth of the thermocline? Also, put correlation coefficients on the X:Y plots.
We have added correlation coefficients on the XY plots in Figure 3. As you suggested, the larger errors in temperature simulations are caused in large part by the model not perfectly capturing the depth of the thermocline. In Fish Lake, especially, observations show that temperatures at the thermocline consistently changes by as much as 4°C per meter during the summer months (NTL LTER, 2012b). A difference in thermocline depth between observations and model by as little as 1 meter can yield high discrepancies between simulated and observed temperatures at those depths.

[revised manuscript text omitted]
 century period (1911-2014). By examining three lakes in the same ecoregion with similar climate forcing, we aim to elucidate differences in responses that relate to
25   morphometry rather than climatic variables. Long-term changes in water temperature, stratification, heat fluxes, and stability are used to investigate how lake depth and surface area alter the response of thermal structure to air temperature and wind speed changes for the three study lakes.

**2 Methods**

**2.1 Study sites**

Three morphometrically different lakes, Lake Mendota, Fish Lake, and Lake Wingra, located near Madison, Wisconsin, United States of America (USA), were selected for this study. These lakes are chosen for (i) their morphometry differences, (ii) their proximity to one another, and (iii) the availability of long-term data for model input and calibration.

Lake Mendota (43°6' N; 89°24'W; Fig. 1a; Table 1), is a dimictic, eutrophic, drainage lake in an urbanizing agricultural watershed (Carpenter and Lathrop, 2008). The lake stratifies during the summer, and typical stratification periods lasts from May to September. Summer (1 June - 31 August) mean surface water temperature is 22.4 °C, and hypolimnetic temperatures vary between 11°C to 15 °C. Normal Secchi depth during the summer is 3.0 meters (Lathrop et al., 1996). Fish Lake (43°17'N; 89°39'W; Fig. 1b; Table 1) is a dimictic, eutrophic, shallow seepage lake located in northwestern Dane County. From 1966 to 2001, lake level rose by 2.75 meters due to increased groundwater flow from higher than normal regional groundwater recharge (Krohelski et al., 2002). Krohelski et al. (2002) hypothesized that the increase in recharge may be the result of increased infiltration from snowmelt after increased snowfall and less frost-covered soil. Summer stratification lasts from the beginning of May to mid-September. Mean surface water temperature 23.9°C and hypolimnetic temperatures are normally near 8°C during summer months; however, some years reach temperatures of only 5-6 °C in the hypolimnion due to shortened spring mixing durations. Average Secchi depth during the summer months is 2.4 m. Lake Wingra (43°3' N; 89°26' W; Fig. 1c; Table 1) is a shallow, eutrophic, drainage lake. It stratifies on short timescales of hours to weeks (Kimura et al., 2016), but does not experienced sustained thermal stratification. Summer mean water temperature is 23.9°C, and mean Secchi depth is 0.7 meters. All three lakes have ice cover during winter months, and a description of ice on the lakes can be found in Magee and Wu (2017).

**2.2 Model description**

To hindcast water temperature and stratification in the three study lakes we use the vertical heat transfer model, DYRESM-WQ (DYnamic REservoir Simulation Model-Water Quality; Hamilton and Schladow, 1997), which employs discrete horizontal Lagrangian layers to simulate vertical water temperature, salinity, and density with input including inflows, outflows, and mixing (Imberger et al., 1978). The model has been previously used on a variety of lake types and is accepted as a standard for hydrodynamic lake modelling (Gal et al., 2003; Hetherington et al., 2015; Imberger and Patterson, 1981; Kara et al., 2012; Tanentzap et al., 2007). DYRESM-WQ adopts a one-dimensional layer structure based on the importance of vertical density stratification over horizontal density variations. A one-dimensional assumption is based on observations that

the density stratification found in lakes inhibits vertical motions while horizontal variations in density relax due to horizontal advection and convection (Antenucci and Imerito, 2003; Imerito, 2010). Surface exchanges include heating due to shortwave radiation penetration into the lake and surface fluxes of evaporation, sensible heat, long wave radiation, and wind stress (Imerito, 2010). Surface layer mixing is based on potential energy required for mixing, and introduction of turbulent kinetic

5 energy through convective mixing, wind stirring, and shear mixing (Imerito, 2010; Yeates and Imberger, 2003). Layer mixing occurs when the turbulent kinetic energy (TKE), stored in the topmost layers, exceeds a potential energy threshold (Yeates and Imberger, 2003). To represent convective overturn, layers are checked for instabilities resulting from surface cooling, and if they exist, layers are merged and a fraction of the potential energy released becomes available as TKE. To represent wind stirring, wind stress is calculated from the wind speed and TKE is produced in the uppermost layer. Upper layers will mix with

10 lower layers if TKE is greater than the energy required for mixing. Mixing by shear flow is determined by calculating the mean horizontal velocity of the uppermost layer, which is dependent on the critical wind speed and the shear period. Yeates and Imberger (2003) improved performance of the surface mixed layer routine within the model by including an effective surface area algorithm based on observations in five lakes of different size, shape, and wind forcing characteristics (see Eq 32 in Yeates and Imberger, 2003) that reduced surface mixing in smaller, more sheltered lakes. The effective area is used to modify the

15 transfer of momentum from surface stress, as described in detail in Yeates and Imberger (2003) and not reproduced here. Their analysis developed a strong inverse relationship between the Lake number and lake-wide average vertical eddy diffusion coefficient, which configures a pseudo two-dimensional deep-mixing within the code, found to significantly improve the simulation of thermal structures observed in lakes that experienced strong wind forcing (Yeates and Imberger, 2003). Details of the surface mixed layer algorithm are not reproduced here, but can be found in Eq 27-34 of Yeates and Imberger (2003).

20 Hypolimnetic mixing is parameterized through a vertical eddy diffusion coefficient, which accounts for turbulence created by the damping of basin-scale internal waves on the bottom boundary and lake interior (Yeates and Imberger, 2003). Detailed equations on the simulation of water temperature and mixing can be found in Imberger and Patterson (1981), Imerito (2010), and Yeates and Imberger (2003).

25 Sediment heat flux is included as a source/sink term for each model layer. A diffusion relation from Rogers et al. (1995) is used to estimate $q_{sed}$, heat transfer from the sediments to the water column.

$$q_{sed} = K_{sed} \frac{dT}{dz} \tag{1}$$

where Ksed represents the sediment conductivity with a value of 1.2 Wm$^{-1}$ °C$^{-1}$ (Rogers et al., 1995), and dT/dz is estimated as:

30 $$\frac{dT}{dz} = \frac{T_S - T_w}{z_{sed}} \tag{2}$$

where dT/dz is the temperature gradient across the sediment-water interface, $T_w$ is the water temperature adjacent to the sediment boundary, $z_{sed}$ is the distance beneath the water-sediment interface at which the sediment temperature becomes relatively invariant, and is taken to be 5 m (Birge et al., 1927). $T_s$ derived from Birge et al. (1927) and seasonally variant as follows:

5    $$T_s = 9.7 + 2.7 \sin\left[\frac{2\pi(D-151)}{TD}\right]$$    (3)

where D is the number of days from the start of the year and TD is the total number of days within a year.

The ice component of the model, DYRESM-WQ-I, is based on the three-component MLI model of Rogers et al., (1995), with the additions of two-way coupling of the hydrodynamic and ice models and time-dependent sediment heat flux for all
10   horizontal layers. The model assumes that the time scale for heat conduction through the ice is short relative to the time scale of meteorological forcing (Patterson and Hamblin, 1988; Rogers et al., 1995), an assumption which is valid with a Stefan number less than 0.1 (Hill and Kucera, 1983). The three-component ice model simulates blue ice, white ice, and snow thickness (see Eq. 1 and Fig. 5 of Rogers et al., 1995). Further description of the ice model can be found in Magee et al. (2016) and Hamilton et al. (in review). Details on ice cover simulations in response to changing climate for the three lakes can be found
15   in Magee and Wu (2017).

[revised manuscript text omitted]

To determine the sensitivity of lake water temperature and stratification in response to air temperature and wind speed, we

15   perturbed these drivers across the range of -10°C to +10°C in 1°C temperature increments and 70% to 130% of the historical value in 5% increments, respectively. For each scenario, meteorological inputs remained the same as for the original simulation and snowfall (rainfall) conversion if the air temperature scenarios increase (decrease) above 0°C. Similarly, the water balance and water clarity are maintained so that the long-term values in both lakes matches the historical record. This limits our analysis as it may exclude changes in water temperatures as a result of increased evapotranspiration, increased precipitation, or altered

[revised manuscript text omitted]
, 2003). Tributary inflows may also contribute to enhanced diffusion due to inflow momentum; however, lakes used in this study are seepage-dominated lakes with little to no tributary contribution, so this phenomenon does not prevent the one-dimensional model from being applicable in this study. This one-dimensional assumption here and corresponding model results may not apply elsewhere for lakes with large inflow volumes.

Additionally, light extinction significantly impacts thermal stratification (Hocking and Straškraba, 1999) and light extinction estimated from Secchi depths can have a large degree of measurement uncertainty (Smith and Hoover, 2000), which may result in uncertainty in water temperatures. To address this uncertainty, where available, we use measured Secchi depth values, which has been shown to improve estimates of the euphotic zone over fixed coefficients (Luhtala and Tolvanen, 2013). Secchi depths were unavailable for portions of the simulation period, and average values for the season were used. Analysis comparing using the method of known Secchi depths to both seasonally-varying average Secchi depths and constant Secchi depths for the lakes indicates that seasonally-varying averages do not significantly decrease model reliability when compared to year-specific values, but do show improvement over constant Secchi depths.

**4.2 Importance of wind speed and other variables**

While many have addressed the importance of changing air temperatures on water temperatures and water quality (e.g. Adrian et al., 2009; Arhonditsis et al., 2004; O'Reilly et al., 2015; Shimoda et al., 2011), fewer have investigated wind speed as a specific driver of changes to lakes (Magee et al., 2016; Kimura et al., 2016; Snortheim et al., 2017). However, results here show that correlations between wind speeds and lake temperature variables are as high as, or higher than, correlations between air temperature and lake temperature variables (Fig. 8), highlighting the importance of considering wind speeds as drivers of lake temperature and stratification changes. For many variables (e.g. stratification dates, epilimnetic temperatures, stability), correlation is opposite for air temperature and wind speed variables, indicating that wind speed increases may offset the effect of air temperature increases, while locations with decreasing wind speeds may experience a greater impact on water temperature and stratification than with air temperature increases alone. This is further supported through sensitivity analysis on stratification onset and overturn (Fig. 9 and 10), which show that for Madison-area lakes, increasing air temperatures and decreasing wind speeds have a cumulative effect toward earlier stratification onset and later overturn. Other studies have also found that wind speed can be as important or even more important than air temperature for influencing lake stratification and water temperatures. For example, Woolway et al. (2017a) 
[revised manuscript text omitted]